# A Statistical Benchmark for Diffusion-Posterior-Sampling Algorithms

**Martin Zach[1,2], Youssef Haouchat[1], and Michael Unser[1]**

[1]Biomedical Imaging Group, École Polytechnique Fédérale de Lausanne, Lausanne, Switzerland
[2]CIBM Center for Biomedical Imaging, Lausanne, Switzerland
{martin.zach, youssef.haouchat, michael.unser}@epfl.ch

## Abstract

We propose a statistical benchmark for diffusion-posterior-sampling (DPS) algorithms in linear inverse problems. Our test signals are discretized Lévy processes whose posteriors admit efficient Gibbs methods. These Gibbs methods provide gold-standard posterior samples for direct, distribution-level comparisons with DPS algorithms. They can also sample the denoising posteriors in the reverse diffusion, which enables the arbitrary-precision Monte Carlo estimation of various objects that may be needed in the DPS algorithms, such as the expectation or the covariance of the denoising posteriors. In turn, this can be used to isolate algorithmic errors from the errors due to learned components. We instantiate the benchmark with the minimum-mean-squared-error optimality gap and posterior-coverage tests and evaluate popular algorithms on the inverse problems of denoising, deconvolution, imputation, and reconstruction from partial Fourier measurements. We release the benchmark code at https://github.com/zacmar/dps-benchmark and invite the community to contribute and report results.

## 1 Introduction

Diffusion models are among the leading generative models in imaging (Rombach et al., 2022), visual computing (Po et al., 2024), finance and time-series analysis (Huang et al., 2024; Rasul et al., 2021), de novo protein and drug design (Watson et al., 2023; Alakhdar et al., 2024), natural language processing (Li et al., 2022), and other domains. Their ability to model complex distributions has motivated their use as priors in the Bayesian resolution of inverse problems. In fact, reconstruction methods that leverage diffusion models are competitive or state-of-the-art for problems such as deconvolution (Ren et al., 2023), magnetic resonance imaging and computed tomography reconstruction (Chung & Ye, 2022; Liu et al., 2023), weather-artifact removal (Özdenizci & Legenstein, 2023), task-conditioned protein design (Bogensperger et al., 2025), audio bandwidth extension and dereverberation (Lemercier et al., 2024), and denoising of financial time-series (Wang & Ventre, 2024).

This empirical success has come in spite of a lack of a natural mechanism for the conditioning on measurements and active research explores how to incorporate the likelihood (Yismaw et al., 2025; Erbach et al., 2025). Currently, conditioning strategies are evaluated in one of two ways. (i) With respect to downstream applications: As an example, evaluations with respect to perceptual metrics such as the structural similarity (Wang et al., 2004), the Fréchet inception distance (Heusel et al., 2017), or the learned perceptual image-patch similarity (Zhang et al., 2018) are common in the imaging sciences. As pointed out by Pierret & Galerne (2025b) and Cardoso et al. (2024), however, these metrics are ill-suited for the statistical evaluation of posterior-sampling algorithms. (ii) In overly simplistic settings: A common fallback is to evaluate conditioning strategies in synthetic settings with (finite-component) Gaussian-mixture priors. Such mixtures remain light-tailed with the tail decreasing exponentially like the widest component. Consequently, they cannot reproduce power-law-like extremes that are common in asset returns (Blattberg & Gonedes, 1974; Cont, 2001) or statistics of images (Wainwright & Simoncelli, 1999). We illustrate signals with such power-law-like extremes later in Figure 3. Benchmarks built on such priors can therefore overstate posterior quality. A proper statistical evaluation in realistic settings is critical in high-stakes applications such as medical imaging, remote sensing, and finance, where decisions based on reconstructions and their associated uncertainties may have significant consequences.

## 1.1 CONTRIBUTIONS

We propose such a statistical benchmark for diffusion-posterior-sampling (DPS) algorithms[1] for linear inverse problems. Our test signals are discretized Lévy processes that admit efficient posterior-sampling algorithms. Indeed, they admit efficient Gibbs methods with exact conditionals that provide gold-standard posterior samples. Our framework supports general posterior-level comparisons (*e.g.*, (sliced) Wasserstein or energy distances or calibration via coverage and posterior predictive checks) by furnishing matched samples obtained from the DPS algorithms and the Gibbs methods.

The Gibbs methods are also suited to sample from the denoising posteriors in the reverse diffusion. This motivates our contribution of a new template for DPS algorithms, in which update steps utilize *samples* from the corresponding denoising posterior. These samples can be used for arbitrary-precision Monte Carlo estimation of various objects that are needed in the update steps of the algorithms, such as the minimum-mean-squared-error (MMSE) denoiser or its Jacobian, which enables the isolation of algorithmic errors from approximation errors due to learned components. We show how several popular DPS algorithms can be re-expressed within our template.

Finally, we instantiate the framework with the MMSE optimality gap and highest-posterior-density coverage checks across the inverse problems of denoising, deconvolution, imputation, and reconstruction from partial Fourier measurements. We target the realistic scenario where a learned denoiser is used and check hyperparameter sensitivity by substituting the arbitrary-precision Monte Carlo counterparts for the learned components. The benchmark code—which is another substantial contribution—is available online. It contains efficient implementations of sampling routines and a containerized runtime that allows novel algorithms to be easily benchmarked.

## 1.2 RELATED WORK

For unconditional sampling, many works derive theoretical bounds on distances between a target distribution and the distribution obtained by (approximations of) the reverse stochastic differential equation (SDE) (see Section 2). For example, Gao et al. (2025) bound the Wasserstein-2 distance with respect to the discretization error of the SDE under the assumption that the target distribution is smooth and log-concave. This directly bounds the number of reverse-diffusion steps needed to obtain a desired accuracy. Under absolute continuity of the target with respect to a Gaussian, Strasman et al. (2025) bound the Kullback–Leibler divergence with respect to properties of the noise schedule.

A common assumption that simplifies the analysis and facilitates the computation of various errors and bounds is that of a Gaussian target. For example, Hurault et al. (2025) analyze the error incurred when using a finite number of prior samples for the estimation of the prior score and track its propagation through the iterations of the reverse-SDE solver. Pierret & Galerne (2025b) derive explicit solutions to the SDE and use them to derive bounds on the Wasserstein-2 distance to the distributions that are obtained via Euler–Maruyama discretizations.

For conditional sampling, Pierret & Galerne (2025a) derive expressions for the Wasserstein-2 distances between the conditional forward marginals and the distributions induced by specific likelihood approximations in the reverse SDE under the assumption of a Gaussian prior. Crafts & Villa (2025) systematically evaluate DPS algorithms numerically under the assumption of a (finite-component) Gaussian-mixture prior and provide reference objects to the DPS algorithms to ensure a fair evaluation. Cardoso et al. (2024) and Boys et al. (2024) also evaluate their algorithms on Gaussian-mixture priors. These Gaussian-mixture priors, however, cannot reproduce power-law-like extremes and can overstate posterior quality.

Beyond diffusion-specific theory, Thong et al. (2024) check the coverage of credible regions produced by different Bayesian recovery strategies and find that those that utilize diffusion models often under-report uncertainty. A shortcoming of their approach is that they use an empirical distribution of images as a surrogate for the prior distribution. Finally, Bohra et al. (2023) also used efficient Gibbs methods to obtain gold-standard posterior samples. Their main focus was to quantify the quality of neural MMSE estimators with different number of parameters. Our work extends this to posterior-level comparisons.

---

[1]We use "DPS algorithms" as an umbrella term for posterior-sampling methods with diffusion priors. The method due to Chung et al. (2023), often called DPS in the literature, will be referred to later as C-DPS.

## 2 BACKGROUND

**Bayesian Linear Inverse Problems**    We seek to estimate a signal $\mathbf{x} \in \mathbb{R}^d$ from the measurements

$$\mathbf{y} = \mathbf{A}\mathbf{x} + \mathbf{n}, \tag{1}$$

where the *forward operator* $\mathbf{A} \in \mathbb{R}^{m \times d}$ models the noiseless linear-measurement acquisition and $\mathbf{n} \in \mathbb{R}^m$ is additive noise. In the Bayesian resolution of this problem (Stuart, 2010), the signals are modeled as a random variable, denoted $\mathbf{X}$, with values in $\mathbb{R}^d$ and distribution $p_{\mathbf{X}}$, referred to as the *prior*. Given any measurement $\mathbf{y}$, the ultimate goal is to analyze the *posterior* $p_{\mathbf{X}|\mathbf{Y}=\mathbf{y}}$. It is related to the *likelihood* $p_{\mathbf{Y}|\mathbf{X}=\mathbf{x}}$ and the prior $p_{\mathbf{X}}$ via Bayes' rule, which states that

$$p_{\mathbf{X}|\mathbf{Y}=\mathbf{y}}(\mathbf{x}) \propto p_{\mathbf{Y}|\mathbf{X}=\mathbf{x}}(\mathbf{y})p_{\mathbf{X}}(\mathbf{x}). \tag{2}$$

In contrast to classical variational methods (Scherzer et al., 2008), the posterior distribution provides natural means to quantify uncertainty and can be summarized by various point estimators. We provide a precise description of point estimators that are relevant in this work in Appendix A.

For a given signal $\mathbf{x}$, the likelihood $p_{\mathbf{Y}|\mathbf{X}=\mathbf{x}}$ is fully specified by the distribution of the noise. A common assumption on the noise is that it is a vector of independent and identically distributed (i.i.d.) Gaussian random variables with mean zero and variance $\sigma_{\mathrm{n}}^2$.[2] In this case, the likelihood is given by

$$p_{\mathbf{Y}|\mathbf{X}=\mathbf{x}}(\mathbf{y}) \propto \exp\left(-\tfrac{1}{2\sigma_{\mathrm{n}}^2}\|\mathbf{A}\mathbf{x} - \mathbf{y}\|^2\right). \tag{3}$$

Thus, once the forward model and the noise distribution are specified, the remaining modeling choice is the prior. Diffusion models are good candidates due to their ability to encode complex distributions.

**Diffusion Models**    Diffusion models were introduced by Song et al. (2021) by unifying the discrete approaches proposed by Song & Ermon (2019) and Ho et al. (2020) in a continuous theory based on SDEs (Klenke, 2020, Chapters 25 and 26). We denote the (diffusion) SDE with *drift coefficient* $\mathbf{f} : \mathbb{R}^d \times \mathbb{R}_{\geq 0} \to \mathbb{R}^d$ and *diffusion coefficient* $g : \mathbb{R}_{\geq 0} \to \mathbb{R}$ as

$$\mathrm{d}\mathbf{X}_t = \mathbf{f}(\mathbf{X}_t, t)\,\mathrm{d}t + g(t)\,\mathrm{d}\mathbf{W}_t, \tag{4}$$

where $\mathbf{W}_t$ is the standard Wiener process. In our setup, the initial condition $\mathbf{X}_0$ is the random variable that describes the signal, thus, $\mathbf{X}_0 = \mathbf{X}$. Under suitable choices for $\mathbf{f}$ and $g$, the forward process admits a limiting marginal $\mathbf{X}_\infty$ as $t \to \infty$. Sampling from $p_{\mathbf{X}_0}$ can then proceed by simulating the SDE (4) in reverse with initial condition $\mathbf{X}_\infty$. By Anderson's theorem (Anderson, 1982), the reverse SDE that reproduces the forward marginals satisfies

$$\mathrm{d}\mathbf{X}_t = \left(\mathbf{f}(\mathbf{X}_t, t) - g^2(t)\nabla \log p_{\mathbf{X}_t}(\mathbf{X}_t)\right)\mathrm{d}t + g(t)\,\mathrm{d}\mathbf{W}_t, \tag{5}$$

where $p_{\mathbf{X}_t}$ denotes the density of $\mathbf{X}_t$ defined by the forward process, and $\mathrm{d}t$ is negative.

The primary challenge in this approach lies in the computation of the *scores* $\nabla \log p_{\mathbf{X}_t}$ for all $t > 0$. A fundamental relation known as Tweedie's formula connects the score with the MMSE denoiser: As we derive rigorously in Appendix B, for $\mathbf{f}(\mathbf{x}, t) = \left(-\tfrac{\beta(t)}{2}\right)\mathbf{x}$ and $g(t) = \sqrt{\beta(t)}$, we have that[3]

$$\nabla \log p_{\mathbf{X}_t}(\mathbf{x}) = -\sigma(t)^{-2}\left(\mathbf{x} - \alpha(t)\mathbb{E}[\mathbf{X}_0 \mid \mathbf{X}_t = \mathbf{x}]\right), \tag{6}$$

where $\alpha(t) = \exp(-\tfrac{1}{2}\int_0^t \beta(s)\mathrm{d}s)$ and $\sigma^2(t) = (1 - \alpha^2(t))$. This yields a practical way to compute $\nabla \log p_{\mathbf{X}_t}(\mathbf{x})$ through the resolution of the MMSE denoising problem of finding $\mathbb{E}[\mathbf{X}_0 \mid \mathbf{X}_t = \mathbf{x}]$. In standard applications where the goal is the generation of new signals, one typically tackles this by approximating the map $(\mathbf{x}, t) \mapsto \mathbb{E}[\mathbf{X}_0 \mid \mathbf{X}_t = \mathbf{x}]$ with a neural network that is learned in an offline step. In our framework, we can instead obtain arbitrary-precision MMSE denoisers via Gibbs methods and thereby eliminate approximation errors from a learned surrogate and isolate errors in DPS algorithms themselves.

The implementation of the reverse SDE for generation requires its own time discretization, for instance with Euler–Maruyama techniques (Higham, 2001). In this work, we will base our backward processes on the alternative denoising diffusion probabilistic model (DDPM) backward process (starting from $\mathrm{Gauss}(\mathbf{0}, \mathbf{I})$)

$$\mathbf{X}_{t-1} = \tfrac{1}{\sqrt{1-\beta_t}}\left(\mathbf{X}_t + \beta_t \nabla \log p_{\mathbf{X}_t}(\mathbf{X}_t)\right) + \sqrt{\beta_t}\mathbf{Z}_t, \tag{7}$$

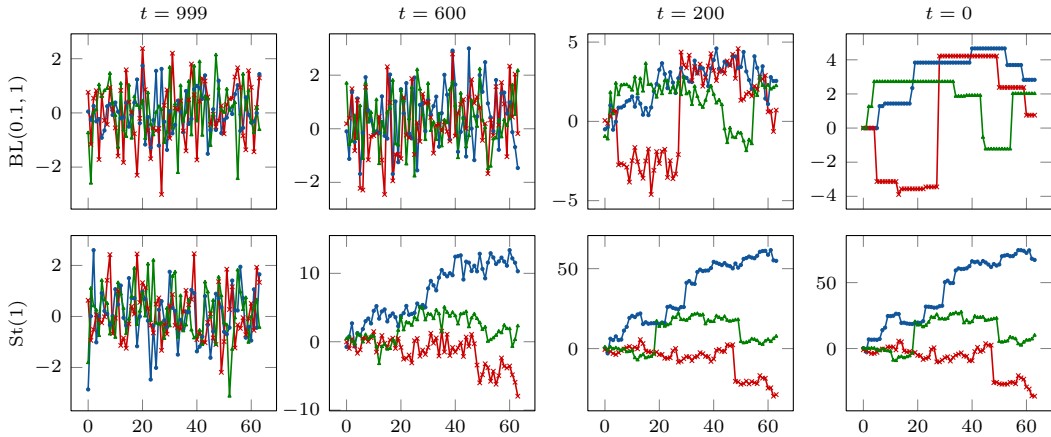

Figure 1: Unconditional reverse-diffusion trajectories obtained by DDPM using the arbitrary-precision Monte Carlo denoiser. Rows: Increment distributions. Columns: Diffusion times. Line styles: Different random states.

that originates from the discrete-time Markov chain that was initially proposed by Sohl-Dickstein et al. (2015) and revisited and popularized by Ho et al. (2020). We relate it to the Euler–Maruyama discretization of the reverse SDE through Taylor expansions in Appendix B.1.

Though we defer details on our signals and the Gibbs methods that we use to obtain the arbitrary-precision MMSE denoiser to Section 3, we demonstrate in Figure 1 that our signals can be generated by coupling the unconditional backward process in (7) with this denoiser. We further motivate this arbitrary-precision denoiser in Figure 2 by comparing histograms of signal increments produced by the learned denoiser and the arbitrary-precision denoiser for a $\mathrm{St}(1)$ increment target (notations summarized in Appendix C.2). The increments of signals gener-

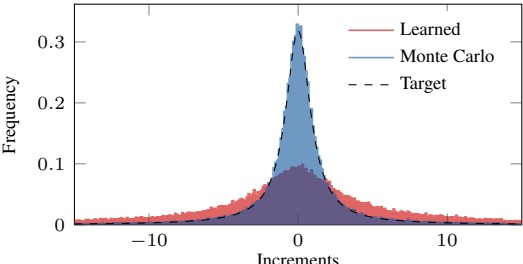

Figure 2: Histogram of increments of signals obtained by DDPM with different denoisers.

ated with the arbitrary-precision denoiser follow the target almost perfectly. Residual errors are due to the discretization of the reverse diffusion and Monte Carlo error of the arbitrary-precision denoiser.

**Diffusion Posterior Sampling**   Our reverse-diffusion sampler can be adapted to sample from a posterior by replacing the prior score $\nabla \log p_{\mathbf{X}_t}$ with the posterior score

$$\nabla \log p_{\mathbf{X}_t|\mathbf{Y}=\mathbf{y}} = \nabla \log p_{\mathbf{X}_t} + \nabla\big(\mathbf{x} \mapsto \log p_{\mathbf{Y}|\mathbf{X}_t=\mathbf{x}}(\mathbf{y})\big) \tag{8}$$

for some given measurement $\mathbf{y}$, obtained by Bayes' theorem. Although the dependence between $\mathbf{Y}$ and $\mathbf{X}_0$ is known through (1) and the likelihood is explicitly modeled via (3), it is generally challenging to relate $\mathbf{Y}$ and $\mathbf{X}_t$ for any $t > 0$. To overcome this, the conditioning on the measurements is usually done in one of two ways. (i) A learned component models the conditional posterior score and also gets the measurements as input. This strategy (pursued by Liu et al. (2023); Özdenizci & Legenstein (2023); Bogensperger et al. (2025); Saharia et al. (2023)) is advantageous when the measurement process is unknown, difficult to model, or prohibitively expensive to evaluate. However, its reconstructions typically degrade under shifts in measurement conditions since the learned components cannot adapt to the new measurement conditions. (ii) The Bayesian separation that is described in (8) is pursued and the likelihood score is approximated. This strategy (pursued by Chung & Ye (2022); Xue et al. (2025) and reviewed by Lemercier et al. (2024)) is advantageous when the measurement process is known, relatively inexpensive to evaluate, and subject to change, but prior knowledge should be reused, which is frequently the case in, *e.g.*, imaging or remote-sensing applications. However, this requires approximations to the likelihood score $\nabla(\mathbf{x} \mapsto \log p_{\mathbf{Y}|\mathbf{X}_t=\mathbf{x}}(\mathbf{y}))$ for all $t > 0$.

---

[2] Our framework supports more general (possibly non-Gaussian) likelihoods, see Section 3.

[3] This is the *variance-preserving* formulation (Song et al., 2021, Section 3.4) with standard normal limiting marginal, where $\beta : \mathbb{R}_{\geq 0} \to \mathbb{R}_{\geq 0}$ controls the speed of the contraction to zero and how much noise is injected.

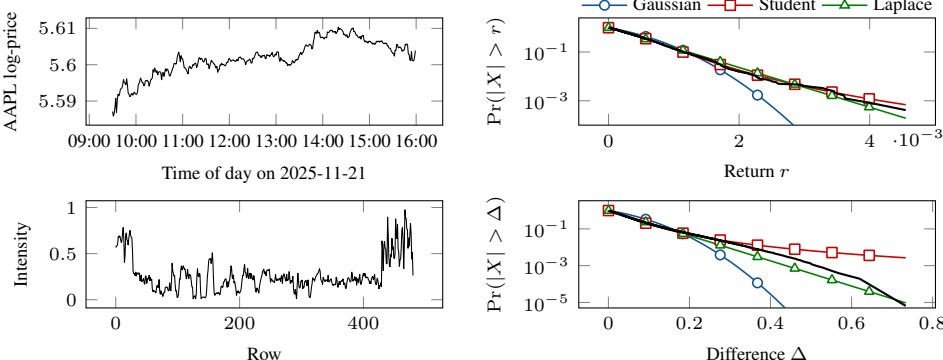

Figure 3: Examples of signals with heavy-tailed increment distributions. Top: Asset returns. Bottom: Columns in natural images. Left: Signals. Right: Survival function of absolute increments (no marker: empirical; markers: best fit to empirical within distribution).

Our benchmark can evaluate either strategy, as well as any other method that would claim to sample from a posterior distribution like in (2). Approach (i), however, relies on black-box learning of the conditional posterior score and its performance heavily depends on various implementation details. Thus, we primarily focus on approach (ii), which necessitates approximations of the likelihood score (and more general DPS algorithms with explicit conditioning, see our proposed generalization in Section 3). For those, our framework can supply arbitrary-precision Monte Carlo estimates of various objects to isolate and quantify the impact of these approximations.

## 3 PROPOSED FRAMEWORK

The prior distributions in our framework will be that of signals of length $d$ obtained by regularly spaced samples of processes with independent, stationary increments (Lévy processes, described in Appendix C). Let $s$ be such a process and let the unit-step increments be $[\mathbf{u}]_k = (s(k) - s(k-1))$ for $k = 1, 2, \ldots, d$. Independence and stationarity imply that the distribution of $[\mathbf{u}]_k$, the *increment distribution* $p_U$, does not depend on $k$. The increment vector is related to the signal $\mathbf{x}$ via $\mathbf{u} = \mathbf{D}\mathbf{x}$, where $\mathbf{D}$ is a finite-difference matrix with an initial condition that allows us to write $\mathbf{x} = \mathbf{D}^{-1}\mathbf{u}$ where $\mathbf{D}^{-1}$ is a lower-triangular matrix of ones. This gives a convenient way to synthesize signals once $\mathbf{u}$ is drawn. The independence of the increments implies that the density of the discrete signal is

$$p_{\mathbf{X}}(\mathbf{x}) = \prod_{k=1}^{d} p_U\big([\mathbf{D}\mathbf{x}]_k\big). \tag{9}$$

We consider four increment distributions that are commonly used in sparse-process models: Gaussian; Laplace; Student-t; and Bernoulli–Laplace (spike-and-slab). Such increment distributions are *sparse* or *heavy-tailed* according to the taxonomy due to Unser & Tafti (2014) and are relevant in signal and image processing, finance, and other fields (Schoutens, 2003). We show instances signals with such heavy-tailed increment distributions in Figure 3. A precise definition of Lévy processes, the matrix $\mathbf{D}$, the increment distributions and their notation along with a discussion about extensions to higher-dimensional signals or signals with more complicated graph structure are given in Appendix C.

**Efficient Posterior Sampling** With the prior distribution specified in (9) and the assumption of Gaussian noise, the posterior associated to the inverse problem intrinsic in (1) is

$$p_{\mathbf{X}|\mathbf{Y}=\mathbf{y}}(\mathbf{x}) \propto \exp\big(-\tfrac{1}{2\sigma_n^2}\|\mathbf{A}\mathbf{x} - \mathbf{y}\|^2\big)p_{\mathbf{X}}(\mathbf{x}) = \exp\big(-\tfrac{1}{2\sigma_n^2}\|\mathbf{A}\mathbf{x} - \mathbf{y}\|^2\big)\prod_{k=1}^{d} p_U\big([\mathbf{D}\mathbf{x}]_k\big). \tag{10}$$

Unless $p_U$ is a Gaussian (the simplified setting in Pierret & Galerne (2025b)), this posterior is not conjugate, so neither closed-form sampling nor direct evaluation of moments is available. Nevertheless, for the increment distributions used in this paper, the posterior distributions admit efficient Gibbs methods via standard latent-variable augmentations. Motivation and more details about the Gibbs methods, such as the burn-in period $B$ and the number of samples $S$, are provided in Appendix D.

The Gaussian, Laplace, and Student-t distributions admit latent representations as infinite-component Gaussian mixtures, which makes them suitable for the Gaussian latent machine (GLM) that was recently introduced by Kuric et al. (2025). It is generally applicable to distributions

$$p(\mathbf{x}) \propto \prod_{k=1}^{n} \phi_k\big([\mathbf{Kx}]_k\big), \qquad (11)$$

---

**Algorithm 1** GLM Gibbs method.

**Require:** $\mathbf{x}_0 \in \mathbb{R}^d$, $\mathbf{K} \in \mathbb{R}^{n \times d}$, conditional latent distributions $\{p_{[\mathbf{Z}]_k | X}\}_{k=1}^n$ and maps $\{\mu_k, \sigma_k^2\}_{k=1}^n$
1: **for** $s = 1, \ldots, B + S$ **do**
2:     Draw $[\mathbf{z}]_k \sim p_{[\mathbf{Z}]_k | X = [\mathbf{Kx}_{s-1}]_k}$    ▷ *par. over $k$*
3:     Draw $\mathbf{x}_s \sim \mathrm{Gauss}(\boldsymbol{\mu}(\mathbf{z}), \boldsymbol{\Sigma}(\mathbf{z}))$
4: **return** $\{\mathbf{x}_{B+s}\}_{s=1}^S$

---

where $\mathbf{K} \in \mathbb{R}^{n \times d}$ and all distributions $\phi_1, \phi_2, \ldots, \phi_n : \mathbb{R} \to \mathbb{R}$ have a latent representation

$$\phi_k(t) = \int_{\mathbb{R}} g_{\mu_k(z), \sigma_k^2(z)}(t) f_k(z) \, \mathrm{d}z, \qquad (12)$$

where the *latent distribution* $f_k$ and the *latent maps* $\mu_k, \sigma_k^2 : \mathbb{R} \to \mathbb{R}$ depend on the distribution $\phi_k$, and $g_{\mu, \sigma^2}$ is the density of a one-dimensional Gaussian distribution with mean $\mu$ and variance $\sigma^2$. We can cast the posterior distribution in (10) into this framework by rewriting it as

$$p_{\mathbf{X}|\mathbf{Y}=\mathbf{y}}(\mathbf{x}) \propto \prod_{k=1}^{m} g_{[\mathbf{y}]_k, \sigma_\mathrm{n}^2}\big([\mathbf{Ax}]_k\big) \prod_{k=1}^{d} p_U\big([\mathbf{Dx}]_k\big) = \prod_{k=1}^{m+d} \phi_k\big([\mathbf{Kx}]_k\big). \qquad (13)$$

There, $\mathbf{K} = [\mathbf{A}; \mathbf{D}]$, $\phi_k = g_{[\mathbf{y}]_k, \sigma_\mathrm{n}^2}$ for $k = 1, 2, \ldots, m$, and $\phi_k = p_U$ for $k = m+1, m+2, \ldots, m+d$. Importantly, non-Gaussian likelihoods can be handled by some appropriate definition of the first $m$ distributions.

The introduction of an appropriate $n$-dimensional random variable $\mathbf{Z}$ with nontrivial distribution (see the details in Kuric et al. (2025)) enables the efficient sampling from the conditionals: Sampling $\mathbf{X} \mid \mathbf{Z} = \mathbf{z}$ amounts to sampling a Gaussian with covariance and mean

$$\boldsymbol{\Sigma}(\mathbf{z}) = (\mathbf{K}^\top \boldsymbol{\Sigma}_0(\mathbf{z})^{-1} \mathbf{K})^{-1} \text{ and } \boldsymbol{\mu}(\mathbf{z}) = \boldsymbol{\Sigma}(\mathbf{z}) \mathbf{K}^\top \boldsymbol{\Sigma}_0(\mathbf{z})^{-1} \boldsymbol{\mu}_0(\mathbf{z}), \qquad (14)$$

respectively, where $\boldsymbol{\Sigma}_0(\mathbf{z}) = \mathbf{diag}\big(\sigma_1^2([\mathbf{z}]_1), \ldots, \sigma_n^2([\mathbf{z}]_n)\big)$ and $\boldsymbol{\mu}_0(\mathbf{z}) = \big(\mu_1([\mathbf{z}]_1), \ldots, \mu_n([\mathbf{z}]_n)\big)$. Sampling $\mathbf{Z} \mid \mathbf{X} = \mathbf{x}$ amounts to sampling $n$ independent one-dimensional *conditional latent distributions* $p_{[\mathbf{Z}]_1 | X = [\mathbf{Kx}]_1}, \ldots, p_{[\mathbf{Z}]_n | X = [\mathbf{Kx}]_n}$ that depend on the distributions $\phi_1, \ldots, \phi_n$ and are given in Table 3 in the appendix along with the corresponding latent distributions and latent maps. We summarize the GLM sampling in Algorithm 1.

For the Bernoulli–Laplace increment distribution, we adapt the algorithm proposed by Bohra et al. (2023) that introduces two $d$-dimensional latent variables: a Bernoulli indicator ("on"/"off"); and a Laplace-distributed increment height. For a self-contained exposition, we rigorously derive the resulting Gibbs method in Appendix D.1.

The Gibbs methods that we just described are suitable for the generation of the gold-standard samples from the posterior that corresponds to the initial inverse problem intrinsic in (1) as well as the generation of samples from the denoising posteriors in the DPS algorithms. In the latter case, the forward operator $\mathbf{A}$ is the identity, the measurements are the noisy intermediate reconstructions $\mathbf{x}_t$, and the noise variance $\sigma_\mathrm{n}^2 = \sigma_t^2$ follows the schedule at timestep $t$.

When these Gibbs methods are embedded within the reverse-diffusion loop, an efficient implementation is paramount to achieve acceptable runtimes. This is most crucial for the

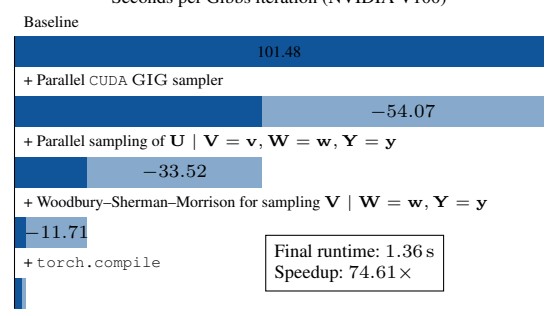

Figure 4: Runtime improvements of the Bernoulli–Laplace sampler.

Bernoulli–Laplace increment distribution, where the sequential drawing of the binary support vector is nested within the Gibbs loop, which in turn may be nested within the reverse-diffusion loop. Accordingly, we tailored our implementation to modern, highly parallel compute units and optimized several components, including custom CUDA- and Triton-compiled sampling routines and incremental updates based on the Woodbury–Sherman–Morrison identities . We achieved a cumulative speedup of $74.61\times$ over the baseline implementation (illustrated in Figure 4 with details in Appendix D.2).

**A Generalized DPS Template**  Widely used methods, such as diffusion plug-and-play (DPnP) (Xu & Chi, 2024), fall outside the pattern described in Section 2, where one approximates likelihood score inside the reverse diffusion. We therefore introduce a simple template that is natural in our setting and accommodates a broader set of DPS algorithms. More precisely, we characterize the iteration rule of

> **Algorithm 2** Template for DPS algorithms.
>
> **Require:** Initial point $\mathbf{x}_T, \mathbf{y}, \mathbf{A}, \boldsymbol{\lambda}$
> 1: **for** $t = T, \ldots, 1$ **do**          ▷ *Diffusion process*
> 2:   $\quad$ Sample $\{\bar{\mathbf{x}}_k\}_{k=1}^S \sim p_{\mathbf{X}_0 \mid \mathbf{X}_t = \mathbf{x}_t}$
> 3:   $\quad$ Update $\quad \mathbf{x}_{t-1} \quad =$
>      $\quad \mathcal{S}(\mathbf{x}_t, \{\bar{\mathbf{x}}_k\}_{k=1}^S, \mathbf{y}, \mathbf{A}, \boldsymbol{\lambda}, t)$
> 4: **return** $\hat{\mathbf{x}}^{\mathrm{alg}} = \mathbf{x}_0$          ▷ *Posterior sample*

DPS algorithms as a two-stage process: Given an iterate $\mathbf{x}_t$ with associated noise variance $\sigma_t^2$, the next iterate $\mathbf{x}_{t-1}$ is computed by (i) drawing $S$ samples denoted $\{\bar{\mathbf{x}}_k\}_{k=1}^S$ from the denoising posterior $p_{\mathbf{X}_0 \mid \mathbf{X}_t = \mathbf{x}_t} \propto \exp\left(-\frac{1}{2\sigma_t^2}\|\cdot - \mathbf{x}_t\|^2\right) p_{\mathbf{X}_0}(\cdot)$; and (ii) the subsequent computation of an update step $\mathcal{S}$ that may utilize the current iterate $\mathbf{x}_t$, the samples $\{\bar{\mathbf{x}}_k\}_{k=1}^S$, the measurements $\mathbf{y}$, the forward operator $\mathbf{A}$, and, possibly, other algorithm-internal parameters such as a scalar that weights likelihood and prior terms or parameters that define the noise schedule. This template is summarized in Algorithm 2 and specialized instances for the update step $\mathcal{S}$ for a variety of popular algorithms are given in Appendix E.2. We have absorbed the (variance-preserving) scaling into the step $\mathcal{S}$ since this template is not fundamentally limited to diffusion processes but supports any (also not monotonically decreasing) noise schedules. In addition, noise variances $\{\sigma_t\}_{t=1}^T$ are usually derived from the internal parameters $\boldsymbol{\lambda}$ that may include a noise schedule.

Through this construction, DPS algorithms can use any statistic $R$ of the samples $\{\bar{\mathbf{x}}_k\}_{k=1}^S$ in their update steps. Most methods use the mean $R(\bar{\mathbf{x}}_1, \ldots, \bar{\mathbf{x}}_S) = \frac{1}{S} \sum_{k=1}^S \bar{\mathbf{x}}_k := \bar{\boldsymbol{\mu}}$, which is the Monte Carlo estimate of $\mathbb{E}[\mathbf{X}_0 \mid \mathbf{X}_t = \mathbf{x}_t]$. An example of a DPS algorithm that utilizes additional statistics is Chung diffusion posterior sampling (C-DPS), which requires the Jacobian of $\mathbf{x}_t \mapsto \mathbb{E}[\mathbf{X}_0 \mid \mathbf{X}_t = \mathbf{x}_t]$. As we show in Appendix E.1, this Jacobian equals (up to the known variance-preserving scaling) the conditional covariance of $\mathbf{X}_0 \mid \mathbf{X}_t = \mathbf{x}_t$, an unbiased estimator of which can be obtained through the statistic $R(\bar{\mathbf{x}}_1, \ldots, \bar{\mathbf{x}}_S) = \frac{1}{S-1} \sum_{k=1}^S (\bar{\mathbf{x}}_k - \bar{\boldsymbol{\mu}})(\bar{\mathbf{x}}_k - \bar{\boldsymbol{\mu}})^\top$. An example of a DPS algorithm that utilizes an alternative statistic is the DPnP algorithm that alternately samples from $p_{\mathbf{X}_0 \mid \mathbf{X}_t = \mathbf{x}_t}$ and a data-proximal problem. There, $R(\bar{\mathbf{x}}_1, \ldots, \bar{\mathbf{x}}_S) = \bar{\mathbf{x}}_1$ is used to obtain one sample from $p_{\mathbf{X}_0 \mid \mathbf{X}_t = \mathbf{x}_t}$. This statistic is frequently used in the asymptotically exact and the CSGM-type algorithms (using the taxonomy due to Daras et al. (2024)). When only a learned MMSE denoiser is available, obtaining this one sample requires a full reverse diffusion. In contrast, it requires only one iteration (and the burn-in period) with the Gibbs methods. Thus, these algorithms are typically faster when they are endowed with the Gibbs methods (see the runtimes in Tables 5 and 6), which enables easy benchmarking. However, CSGM-type algorithms typically do not aim at posterior sampling and we do not benchmark them here.

Since the denoising posteriors are always sub-Gaussian, the Monte Carlo estimation of any object enjoys favorable convergence. For instance, the computational complexity of estimating the covariance up to a desired precision in the operator norm scales linearly with the dimensionality of the signal (Vershynin, 2018, Theorem 4.7.1).

**Extensions**  A prerequisite for a quantitative evaluation of posterior-sampling algorithms is the availability of reasonably efficient samplers that can provide gold-standard samples. The development of such samplers for posteriors arising from nonlinear measurement models and non-Gaussian noise is challenging, and existing methods currently address only specific cases (e.g., Wang et al. (2017) study a nonlinear-Gaussian measurement model with a Laplace prior). Importantly, our framework is modular: as more general-purpose samplers for these posteriors become available, they can be plugged into our benchmark directly. The denoising posteriors in the reverse diffusion do not change with the likelihood and can, therefore, always be efficiently sampled.

When going to higher dimensions, the primary challenge lies in the sampling of the high-dimensional Gaussian distributions required in the Gibbs methods. Luckily, the structure of the involved operators in our case is such that the Gaussians can be efficiently sampled with perturb-and-MAP approaches with matrix-free conjugate gradient implementations; we discuss this in more detail and show how the runtime of different samplers change with the dimensions in Appendix D.2. Sampling high-dimensional Gaussians is a well-studied problem and advances in that field can directly be used in our framework.

Our gold-standard posterior samples can be compared to samples obtained by *any* posterior-sampling algorithm. This includes classical Markov-chain Monte Carlo algorithms, algorithms that utilize flow-matching priors, and others. In this work, we primarily focus on DPS algorithms because our framework can supply arbitrary-precision Monte Carlo objects to them. We believe that this fundamental principle can be extended to other algorithms, in particular those that utilize flow-matching priors. Such algorithms are frequently evaluated on toy examples based on Gaussian mixtures (*e.g.* by Pourya et al. (2025)), that are overly simplistic.

## 4 NUMERICAL EXPERIMENTS

We consider signals of dimension $d = 64$ and four inverse problems that are frequently encountered in various estimation tasks throughout the natural sciences: denoising; deconvolution; imputation; and reconstruction from partial Fourier measurements. The dimension of the signal is large enough such that the corresponding operators can be sensibly defined, yet small enough such that the benchmark has acceptable runtimes. We provide experiments about the runtime with larger signals in Appendix D.2, details about the operators in Appendix F.1, and precise descriptions of the benchmarking pipeline (*e.g.*, the number of training, validation, and test signals, and the number of iterations in the Gibbs methods) in Appendix F.2.

### 4.1 RECONSTRUCTION ALGORITHMS

**Model-Based Methods** We consider the model-based methods

$$\hat{\mathbf{x}}^{\ell_2}(\mathbf{y}, \lambda) = \underset{\mathbf{x} \in \mathbb{R}^d}{\arg\min} \left( \tfrac{1}{2}\|\mathbf{A}\mathbf{x} - \mathbf{y}\|^2 + \lambda\|\mathbf{D}\mathbf{x}\|^2 \right), \tag{15}$$

and

$$\hat{\mathbf{x}}^{\ell_1}(\mathbf{y}, \lambda) = \underset{\mathbf{x} \in \mathbb{R}^d}{\arg\min} \left( \tfrac{1}{2}\|\mathbf{A}\mathbf{x} - \mathbf{y}\|^2 + \lambda\|\mathbf{D}\mathbf{x}\|_1 \right) \tag{16}$$

as baseline reconstruction algorithms. They coincide with the maximum-a-posteriori (MAP) estimators of Lévy processes associated with Gaussian and Laplace increment distributions, respectively.

**Diffusion Posterior Sampling Algorithms** We consider C-DPS (Chung et al., 2023), diffusion models for plug-and-play image restoration (DiffPIR), (Zhu et al., 2023) and DPnP (Xu & Chi, 2024). This selection demonstrates the applicability of the framework to algorithms that require denoising-posterior samples (DPnP), the MMSE denoiser (DiffPIR), and its Jacobian (C-DPS), which covers most of the existing DPS algorithms. For each DPS algorithm, we benchmark a variant that uses learned components (learning details are provided in Appendix F.3) and a variant that uses Gibbs samples of the denoising posterior. For DPnP, this fully removes approximation errors. For the others, the learned components and the Monte Carlo estimates of those components have varying quality for different distributions and noise variances that we systematically investigate in Appendix F.4. We provide our main results, the MMSE optimality gap, for the learned variant and then investigate changes when we substitute the Gibbs samples for the learned components.

The model-based methods and the DPS algorithms require the tuning of some hyperparameters. These were found by grid search on validation data independently for each algorithm, increment distribution, and forward operator. The precise setup for this grid search is given in Appendix F.5. The hyperparameters for the DPS algorithms were tuned to the learned denoiser. Parameters obtained with this procedure are later denoted with a star in the superscript.

**Gold-Standard Gibbs Methods** The Gibbs methods are used to obtain gold-standard samples from the posterior. As described in Section 3, the Gibbs methods are parameter- and bias-free and efficient. Consequently, they are well-suited for our purpose. Chain lengths, diagnostics, and implementation details are given in Appendix F.2; we reuse the same settings across operators and increment families.

### 4.2 RESULTS

For any measurement $\mathbf{y}$, some DPS algorithm $\mathrm{alg}$ that depends on the parameters $\boldsymbol{\lambda}$ will produce samples that we denote $\{\hat{\mathbf{x}}_k^{\mathrm{alg}}(\mathbf{y}, \boldsymbol{\lambda})\}_{k=1}^{N_{\mathrm{samples}}}$. We moreover denote $\hat{\mathbf{x}}_{\mathrm{MMSE}}^{\mathrm{alg}}(\mathbf{y}, \boldsymbol{\lambda}) :=$

Table 1: MMSE optimality gap in decibel (mean $\pm$ standard deviation; lower is better; 0 is a perfect reconstruction) of various estimation methods over the test set. Bold: best among DPS algorithms.

| | | Gauss(0, 0.25) | Laplace(1) | BL(0.1, 1) | St(1) | St(2) | St(3) |
|---|---|---|---|---|---|---|---|
| Denoising | C-DPS | **0.12 ± 0.18** | 0.12 ± 0.20 | 2.22 ± 2.26 | 3.26 ± 1.01 | 0.28 ± 0.30 | 0.10 ± 0.18 |
| | DiffPIR | 0.16 ± 0.21 | **0.09 ± 0.16** | **0.72 ± 1.10** | **0.93 ± 1.06** | **0.07 ± 0.14** | 0.15 ± 0.21 |
| | DPnP | 0.24 ± 0.25 | 0.11 ± 0.17 | 1.33 ± 2.12 | 1.19 ± 1.38 | 0.10 ± 0.17 | **0.10 ± 0.17** |
| | $\ell_1$ | 0.15 ± 0.21 | 0.06 ± 0.12 | 3.44 ± 2.38 | 0.38 ± 0.43 | 0.14 ± 0.19 | 0.11 ± 0.18 |
| | $\ell_2$ | 0.00 ± 0.01 | 0.16 ± 0.21 | 8.61 ± 3.10 | 3.25 ± 0.99 | 0.74 ± 0.83 | 0.25 ± 0.33 |
| Deconvolution | C-DPS | 0.12 ± 0.20 | 0.12 ± 0.23 | 4.30 ± 3.87 | 18.30 ± 5.28 | 0.46 ± 1.40 | 0.17 ± 0.53 |
| | DiffPIR | **0.07 ± 0.17** | **0.07 ± 0.19** | **1.09 ± 2.22** | 10.45 ± 6.10 | **0.09 ± 0.57** | **0.08 ± 0.26** |
| | DPnP | 0.10 ± 0.18 | 0.13 ± 0.22 | 1.71 ± 2.49 | **7.84 ± 5.66** | 0.35 ± 1.39 | 0.14 ± 0.41 |
| | $\ell_1$ | 1.65 ± 0.84 | 1.38 ± 0.86 | 1.86 ± 3.14 | 1.87 ± 4.01 | 1.10 ± 1.19 | 1.28 ± 0.94 |
| | $\ell_2$ | 0.00 ± 0.01 | 0.07 ± 0.23 | 6.11 ± 4.49 | 21.50 ± 4.46 | 1.44 ± 2.85 | 0.36 ± 1.09 |
| Imputation | C-DPS | 0.15 ± 0.29 | 0.18 ± 0.39 | 2.99 ± 2.82 | 23.33 ± 8.69 | 0.50 ± 1.09 | 0.14 ± 0.57 |
| | DiffPIR | **0.09 ± 0.23** | **0.08 ± 0.24** | **0.24 ± 1.14** | **0.88 ± 3.50** | **0.11 ± 0.62** | **0.08 ± 0.42** |
| | DPnP | 0.14 ± 0.32 | 0.17 ± 0.36 | 0.50 ± 1.28 | 10.89 ± 5.92 | 0.25 ± 0.82 | 0.27 ± 0.58 |
| | $\ell_1$ | 1.74 ± 1.12 | 1.77 ± 1.35 | 1.25 ± 2.78 | 13.32 ± 5.32 | 1.37 ± 2.56 | 1.55 ± 1.58 |
| | $\ell_2$ | 0.00 ± 0.01 | 0.01 ± 0.05 | 1.10 ± 1.88 | 0.42 ± 0.95 | 0.06 ± 0.34 | 0.02 ± 0.28 |
| Fourier | C-DPS | 0.15 ± 0.36 | 0.26 ± 0.65 | 5.90 ± 4.41 | 4.29 ± 5.78 | 0.53 ± 0.83 | 0.35 ± 0.77 |
| | DiffPIR | **0.11 ± 0.29** | **0.08 ± 0.31** | **0.83 ± 1.44** | 3.19 ± 4.37 | **0.11 ± 0.39** | **0.12 ± 0.37** |
| | DPnP | 0.11 ± 0.35 | 0.20 ± 0.51 | 1.88 ± 2.47 | 2.45 ± 4.83 | 0.39 ± 0.89 | 0.24 ± 0.64 |
| | $\ell_1$ | 1.50 ± 1.59 | 0.73 ± 0.94 | 3.57 ± 2.82 | 1.07 ± 2.98 | 0.71 ± 0.99 | 0.78 ± 0.97 |
| | $\ell_2$ | 0.00 ± 0.02 | 0.36 ± 0.73 | 12.22 ± 4.53 | 9.47 ± 8.34 | 2.66 ± 3.57 | 1.03 ± 1.79 |

$\frac{1}{N_{\text{samples}}} \sum_{k=1}^{N_{\text{samples}}} \hat{\mathbf{x}}_k^{\text{alg}}(\mathbf{y}, \boldsymbol{\lambda})$. For an estimation method $\hat{\mathbf{x}}^{\text{est}}(\,\cdot\,)$ and data $\mathbf{y}$ with corresponding data-generating signal $\mathbf{x}$ we measure the MMSE optimality gap (in decibel) defined by

$$10 \log_{10} \left( \frac{\|\hat{\mathbf{x}}^{\text{est}}(\mathbf{y}) - \mathbf{x}\|^2}{\|\hat{\mathbf{x}}_{\text{MMSE}}^{\text{Gibbs}}(\mathbf{y}) - \mathbf{x}\|^2} \right), \tag{17}$$

where $\hat{\mathbf{x}}^{\text{est}}(\mathbf{y}) = \hat{\mathbf{x}}^{\ell_{1 \vee 2}}(\mathbf{y}, \lambda^\star)$ for model-based methods and $\hat{\mathbf{x}}^{\text{est}}(\mathbf{y}) = \hat{\mathbf{x}}_{\text{MMSE}}^{\text{alg}}(\mathbf{y}, \lambda^\star)$ for DPS algorithms. A gap of $0$ indicates a perfect recovery of the gold-standard MMSE estimate and any positive values show the orders of magnitude of the error relative to the reference error. We found that $N_{\text{samples}} = 50$ provided a good tradeoff between runtime and accuracy by benchmarking the gold-standard Gibbs method with that number of samples.

We report in Table 1 the mean and standard deviation of the MMSE optimality gap over all signal-measurement pairs $(\mathbf{x}, \mathbf{y})$ in the test set obtained by the model-based methods and the DPS algorithms endowed with the learned denoiser . The Gaussian increment distribution validates the implementation: Since the MMSE and the MAP point estimates coincide, the model-based $\ell_2$ estimator matches the Gibbs reference up to the error due to the finite parameter-grid resolution. When the posterior mean is smooth (*e.g.*, imputation and some deconvolution cases), $\ell_2$ is the best model-based choice and frequently outperforms the DPS algorithms. When the posterior mean is close to piecewise-constant (typical in denoising of signals with sparse increments), the $\ell_1$ estimator is preferred. Among DPS algorithms, DiffPIR is typically the top performer and often exceeds $\ell_2$ and $\ell_1$ baselines in deconvolution, imputation, and reconstruction from partial Fourier measurements. For spike-and-slab settings (Bernoulli–Laplace), DPS algorithms substantially outperform the model-based baselines across operators. In deconvolution and reconstruction from partial Fourier measurements, DPS algorithms frequently match or surpass the best model-based estimator.

We now inspect the change in performance after we substitute higher-quality Monte Carlo components for the learned components. We do this *without retuning of the hyperparameters*, which allows us to see if the performance of the algorithms increases automatically with the quality of the denoiser. Here, we discuss general trends; an exhaustive quantitative evaluation and a precise quantification of the quality of the learned and Monte Carlo objects is given in Appendix G. For the same hyperparameters, the performance of DPnP increases significantly with the quality of the denoising-posterior samples. For example, the optimality gap decreases by $10.46\,\text{dB}$ for imputation of signals with $\text{St}(1)$ increments, and significantly for other measurement operators for signals with $\text{St}(1)$ increments and $\text{BL}(0.1, 1)$ increments. By contrast, C-DPS and DiffPIR can require a retuning when the denoiser changes: Scores can deteriorate after one has substituted a higher-quality Monte Carlo denoiser for the learned one, but a brief hand-tuning of the hyperparameters on the validation set improves them way beyond the learned denoiser. For instance, for DiffPIR and imputation of signals with $\text{St}(1)$ increments, reusing the hyperparameters deteriorates the gap by $13.56\,\text{dB}$, whereas a brief hand-tuning decreased the optimality gap by almost $10\,\text{dB}$ over what is reported in Table 1. Qualitative examples of the MMSE estimates and the marginal variances obtained by the DPS algorithms and the gold-standard Gibbs methods are shown in Figures 11 to 18 in the appendix.

Prototypical samples and the corresponding MMSE estimate obtained from a DPS algorithm (here DiffPIR for deconvolution of a signal with $\mathrm{BL}(0.1, 1)$ increments) are shown in Figure 5. (The full conditional reverse-diffusion trajectory, the data-generating signal, the measurements, and the MMSE estimated obtained with the gold-standard Gibbs methods are shown in Figure 19 in the appendix.) The figure highlights a key distinction: Posterior *samples* often preserve high-frequency structure and reflect prior variability, whereas the *MMSE point estimate*—obtained by

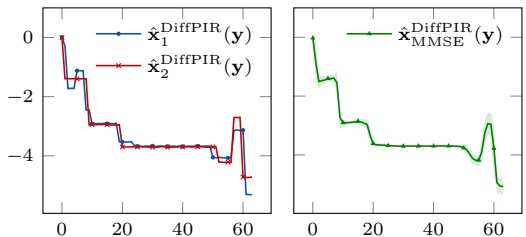

Figure 5: Conditional generation for deconvolution of a signal with $\mathrm{BL}(0.1, 1)$ increments with Diff-PIR. The shaded area indicates the variance.

averaging all samples—is much smoother. This explains why DPS methods tend to score higher on perception-oriented metrics, while regressors that target the MMSE point estimate (through training with the mean squared error) excel on distortion metrics like the peak signal-to-noise-ratio (PSNR). Consistent with this distinction, Saharia et al. (2023) fairly compare a sampling-based method to an MMSE regressor and find the expected tradeoff: higher PSNR and structural similarity for the regressor; and better perceptual scores for the sampler. We therefore recommend to make the Bayesian target explicit—point estimate versus sample quality—and to use evaluation protocols that are aligned to that target. Our framework supports this by offering gold-standard posterior samples and arbitrary-precision Monte Carlo estimates.

In addition to the evaluation of the MMSE optimality gap we analyze the highest-posterior-density coverage of the algorithms. Specifically, for any measurement $\mathbf{y}$ and any $k = 1, 2, \ldots, N_{\mathrm{samples}}$, we define[4] $l_k(\mathbf{y}) := \log p_{\mathbf{X}|\mathbf{Y}=\mathbf{y}}(\hat{\mathbf{x}}_{P(k)}^{\mathrm{alg}}(\mathbf{y}, \boldsymbol{\lambda}^{\mathrm{alg},\star}))$ where $P$ is the permutation that ensures that $l_1(\mathbf{y}) \geq l_2(\mathbf{y}) \geq \cdots \geq l_{N_{\mathrm{samples}}}(\mathbf{y})$ and define the empirical highest-posterior-density threshold at $\alpha \in [0, 1]$ as $l_{\lceil \alpha N_{\mathrm{samples}} \rceil}(\mathbf{y})$. We declare the data-generating signal $\mathbf{x}$ covered if $\log p_{\mathbf{X}|\mathbf{Y}=\mathbf{y}}(\mathbf{x}) \geq l_{\lceil \alpha N_{\mathrm{samples}} \rceil}(\mathbf{y})$ and define the coverage of a method as the fraction of signal-measurement pairs $(\mathbf{x}, \mathbf{y})$ in the test set for which $\mathbf{x}$ is covered by the threshold $l_{\lceil \alpha N_{\mathrm{samples}} \rceil}(\mathbf{y})$. The coverage of a calibrated posterior-sampling method will be $\alpha$, up to Monte Carlo error. A coverage result that is less than $\alpha$ indicates that the samples concentrate too heavily around the mode; a coverage result that is greater than $\alpha$ indicates that the samples are too spread out. We again discuss general trends here and present an exhaustive quantitative evaluation in Appendix G. The coverages obtained by the DPS algorithms are generally much smaller than $\alpha$, which indicates that they are uncalibrated and is in line with what is reported by Thong et al. (2024). For C-DPS and DiffPIR, the reported coverage values are almost always $0$ except for $\mathrm{BL}(0.1, 1)$ and $\mathrm{St}(1)$ increments, where the coverages are usually (close to) $1$ for C-DPS and inconsistent for DiffPIR. For almost all increment distributions and forward operators, DPnP reports coverage values that are closest to but typically smaller than $\alpha$.

## 5    CONCLUSION

We have introduced a statistical benchmark for diffusion posterior sampling algorithms for linear inverse problems. The framework constructs signals with a known distribution, simulates the measurement process, and subsequently generates samples from the posterior distribution that arises through the combination of the known prior and the known likelihood. Gold-standard samples from this distribution are obtained via efficient Gibbs methods. These samples are then compared to those obtained by the diffusion posterior sampling algorithms. In addition, the Gibbs methods can be used to obtain arbitrary-precision Monte Carlo estimates of objects that are needed in the reverse stochastic differential equation, such as the minimum-mean-squared-error denoiser or its Jacobian. Consequently, the framework also enables the isolation and quantification of the error attributable to the likelihood approximations in the conditional reverse diffusion. We have provided numerical results for three common diffusion posterior sampling algorithms applied to four common inverse problems. A consistent theme across all tested algorithms is that they are not calibrated, which demonstrates that research into algorithms that perform better in this respect remains crucial. We invite other researchers to benchmark their algorithms on our open implementation.

---

[4]With some slight abuse of notation, $\log p_{\mathbf{X}|\mathbf{Y}=\mathbf{y}}$ is the unnormalized ground-truth log-posterior (10). Since the additive constant is the same across all methods, this ranking is valid.

**Reproducibility Statement**    We release an online repository with complete algorithm implementations and step-by-step instructions to reproduce all results. A containerized runtime enables one-command setup and fully automated execution via the provided scripts. Each algorithm is specified at a level that supports independent re-implementation: The main text precisely details Gaussian latent-machine sampling; and the appendix presents the Bernoulli–Laplace Gibbs method in implementation-aligned notation, together with practical optimizations required for acceptable runtimes. The appendix also enumerates all experimental settings, including the numbers of training/validation/test signals, the samples-per-datum for each sampler, and the exact grid-search procedure used to select hyperparameters.

**Usage of Large Language Models**    We used large language models to adapt passages of already-written text for readability and conciseness.

**Acknowledgments**    The research leading to these results has received funding from the European Research Council under Grant ERC- 2020-AdG FunLearn-101020573 and by the Swiss National Science Foundation under Sinergia Grant CRSII5 198569. The authors thank Ludovic Reymond for his valuable contributions to preliminary experiments.

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

## A    BAYES ESTIMATORS

A benefit of the Bayesian approach over classical variational methods (see, e.g., (Scherzer et al., 2008)) is that different point estimates arise from a fixed prior. For a given measurement $\mathbf{y}$, these point estimates summarize the posterior distribution $p_{\mathbf{X}|\mathbf{Y}=\mathbf{y}}$ with respect to a given loss $\ell : \mathbb{R}^d \times \mathbb{R}^d \to \mathbb{R}$ via the optimization problem of finding the point $\hat{\mathbf{x}}_\ell(\mathbf{y})$ that minimizes the posterior risk:

$$\hat{\mathbf{x}}_\ell(\mathbf{y}) = \arg\min_{\hat{\mathbf{x}} \in \mathbb{R}^d} \left( \int_{\mathbb{R}^d} \ell(\hat{\mathbf{x}}, \mathbf{x})\, p_{\mathbf{X}|\mathbf{Y}=\mathbf{y}}(\mathbf{x})\, \mathrm{d}\mathbf{x} \right). \tag{18}$$

In this paper, the Bayes estimator with respect to the mean-squared error (MSE) $\ell = \frac{1}{d}\|\cdot - \cdot\|^2$ plays a key role due to its close relation to the prior *score* in the reverse diffusion (see Section 2) and because we quantify the performance of DPS algorithms via the MMSE optimality gap in Section 4. With this choice of $\ell$, (18) can be written as

$$\hat{\mathbf{x}}_{\mathrm{MMSE}}(\mathbf{y}) = \arg\min_{\hat{\mathbf{x}} \in \mathbb{R}^d} \left( \int_{\mathbb{R}^d} \tfrac{1}{d}\|\hat{\mathbf{x}} - \mathbf{x}\|^2\, p_{\mathbf{X}|\mathbf{Y}=\mathbf{y}}(\mathbf{x})\, \mathrm{d}\mathbf{x} \right) = \int_{\mathbb{R}^d} \mathbf{x} p_{\mathbf{X}|\mathbf{Y}=\mathbf{y}}(\mathbf{x})\, \mathrm{d}\mathbf{x} = \mathbb{E}[\mathbf{X} \mid \mathbf{Y} = \mathbf{y}], \tag{19}$$

which is the expectation of the posterior $p_{\mathbf{X}|\mathbf{Y}=\mathbf{y}}$.

Another widely-used estimator arises through the choice

$$\ell(\hat{\mathbf{x}}, \mathbf{x}) = -\chi_{\{\hat{\mathbf{x}}\}}(\mathbf{x}) \tag{20}$$

where

$$\chi_A(\mathbf{x}) := \begin{cases} 1 & \text{if } \mathbf{x} \in A, \\ 0 & \text{else}, \end{cases} \tag{21}$$

which leads to the MAP estimator that seeks the mode of the posterior:[5]

$$\hat{\mathbf{x}}_{\mathrm{MAP}}(\mathbf{y}) = \arg\min_{\hat{\mathbf{x}} \in \mathbb{R}^d} \left( \int_{\mathbb{R}^d} -\chi_{\{\hat{\mathbf{x}}\}}(\mathbf{x})\, p_{\mathbf{X}|\mathbf{Y}=\mathbf{y}}(\mathbf{x})\, \mathrm{d}\mathbf{x} \right) = \arg\max_{\hat{\mathbf{x}} \in \mathbb{R}^d} p_{\mathbf{X}|\mathbf{Y}=\mathbf{y}}(\hat{\mathbf{x}}). \tag{22}$$

Rewriting (22) as

$$\hat{\mathbf{x}}_{\mathrm{MAP}}(\mathbf{y}) = \arg\min_{\hat{\mathbf{x}} \in \mathbb{R}^d} \left( -\tfrac{1}{2\sigma_{\mathrm{n}}^2}\|\mathbf{A}\hat{\mathbf{x}} - \mathbf{y}\|^2 - \log p_{\mathbf{X}}(\hat{\mathbf{x}}) \right), \tag{23}$$

reveals a close relation to classical variational approaches after identifying the regularizer with $-\log p_{\mathbf{X}}$.

---

[5]This definition is informal but sufficient for the purposes of this paper. For continuous posteriors, the strict 0–1 loss Bayes' rule is ill-posed. A common formalization defines MAP as the limit of Bayes estimators under shrinking small-ball 0–1 losses; under additional regularity, this limit agrees with the posterior mode (Bassett & Deride, 2018; Clason et al., 2019). The MAP estimator may also not be unique.

## B  TWEEDIE'S FORMULA

In the setting of Section 2, we now derive an equality that relates $\nabla \log p_{\mathbf{X}_t}$ to $\mathbb{E}[\mathbf{X}_0 \mid \mathbf{X}_t = \cdot]$, *i.e.*, the MMSE estimate of $\mathbf{X}_0$ given that $\mathbf{X}_t$ takes on a certain value. Similar derivations can be found in, *e.g.*, (Song et al., 2021; Chung et al., 2023; Daras et al., 2024), but we include it to underscore the relevance of the MMSE estimate in this paper and to facilitate the understanding of its relation to various objects. Under the variance-preserving choice for $\mathbf{f}(\mathbf{x}, t) = -\frac{\beta(t)}{2}\mathbf{x}$ and $g(t) = \sqrt{\beta(t)}$ of the drift and diffusion coefficient, the diffusion SDE (4) simplifies to a time-inhomogeneous Ornstein–Uhlenbeck SDE (see Klenke (2020, Example 26.5))

$$\mathrm{d}\mathbf{X}_t = -\tfrac{\beta(t)}{2}\mathbf{X}_t \,\mathrm{d}t + \sqrt{\beta(t)}\,\mathrm{d}\mathbf{W}_t, \tag{24}$$

whose pathwise solution

$$\mathbf{X}_t = \alpha(t)\mathbf{X}_0 + \int_0^t \frac{\alpha(t)}{\alpha(s)}\sqrt{\beta(t)}\,\mathrm{d}\mathbf{W}_s, \tag{25}$$

where $\mathbf{X}_0$ is an appropriate initial condition and $\alpha(t) = \exp\left(-\frac{1}{2}\int_0^t \beta(s)\,\mathrm{d}s\right)$, can be computed with standard techniques, see, *e.g.*, (Gardiner, 1990, Section 4.4.4). In addition, since

$$\int_0^t \left(\frac{\alpha(t)}{\alpha(s)}\right)^2 \beta(t)\,\mathrm{d}s = \int_0^t \beta(s)\exp\left(-\int_s^t \beta(u)\,\mathrm{d}u\right)\mathrm{d}s = 1 - \alpha^2(t), \tag{26}$$

we can write that

$$\mathbf{X}_t = \alpha(t)\mathbf{X}_0 + \sigma(t)\mathbf{N} \tag{27}$$

in distribution, where $\sigma^2(t) = (1 - \alpha^2(t))$. Consequently, the density of $\mathbf{X}_t$ is given by the convolution of $p_{\mathbf{X}_0}$ with a Gaussian with variance $\sigma^2(t)$ and appropriate scaling by $\alpha(t)$, which we write as

$$p_{\mathbf{X}_t}(\mathbf{x}) = \int_{\mathbb{R}^d} g_{\mathbf{0},\sigma(t)^2\mathbf{I}}(\mathbf{x} - \alpha(t)\hat{\mathbf{x}}) p_{\mathbf{X}_0}(\hat{\mathbf{x}})\,\mathrm{d}\hat{\mathbf{x}}, \tag{28}$$

where $g_{\boldsymbol{\mu},\boldsymbol{\Sigma}}(\mathbf{x}) = (2\pi)^{-\frac{d}{2}}|\boldsymbol{\Sigma}|^{-\frac{1}{2}}\exp\left(-\frac{1}{2}\|\mathbf{x} - \boldsymbol{\mu}\|_{\boldsymbol{\Sigma}^{-1}}^2\right)$. Finally, after taking the gradient, we see that

$$\begin{aligned}
\nabla p_{\mathbf{X}_t}(\mathbf{x}) &= \int_{\mathbb{R}^d} \nabla g_{\mathbf{0},\sigma(t)^2\mathbf{I}}(\mathbf{x} - \alpha(t)\hat{\mathbf{x}}) p_{\mathbf{X}_0}(\hat{\mathbf{x}})\,\mathrm{d}\hat{\mathbf{x}} \\
&= \int_{\mathbb{R}^d} -\tfrac{1}{\sigma^2(t)}(\mathbf{x} - \alpha(t)\hat{\mathbf{x}}) g_{\mathbf{0},\sigma^2(t)\mathbf{I}}(\mathbf{x} - \alpha(t)\hat{\mathbf{x}}) p_{\mathbf{X}_0}(\hat{\mathbf{x}})\,\mathrm{d}\hat{\mathbf{x}} \\
&= -\tfrac{1}{\sigma^2(t)}\left(\mathbf{x}p_{\mathbf{X}_t}(\mathbf{x}) - \alpha(t)\int_{\mathbb{R}^d} \hat{\mathbf{x}} g_{\mathbf{0},\sigma^2(t)\mathbf{I}}(\mathbf{x} - \alpha(t)\hat{\mathbf{x}}) p_{\mathbf{X}_0}(\hat{\mathbf{x}})\,\mathrm{d}\hat{\mathbf{x}}\right) \\
&= -\tfrac{1}{\sigma(t)^2}\left(\mathbf{x}p_{\mathbf{X}_t}(\mathbf{x}) - \alpha(t)p_{\mathbf{X}_t}(\mathbf{x})\mathbb{E}[\mathbf{X}_0 \mid \mathbf{X}_t = \mathbf{x}]\right).
\end{aligned} \tag{29}$$

Finally, after dividing by $p_{\mathbf{X}_t}(\mathbf{x})$ and since $\frac{\nabla p_{\mathbf{X}_t}(\mathbf{x})}{p_{\mathbf{X}_t}(\mathbf{x})} = \nabla \log p_{\mathbf{X}_t}(\mathbf{x})$, we find the celebrated Tweedie identity

$$\nabla \log p_{\mathbf{X}_t}(\mathbf{x}) = -\sigma(t)^{-2}\left(\mathbf{x} - \alpha(t)\mathbb{E}[\mathbf{X}_0 \mid \mathbf{X}_t = \mathbf{x}]\right). \tag{30}$$

### B.1  A CONNECTION BETWEEN THE DISCRETIZED REVERSE SDE AND DDPM

To show the connection between the Euler–Maruyama discretization of the reverse-diffusion SDE and the DDPM backward process, we start by deriving the latter from the respective forward process. DDPM has been introduced by Sohl-Dickstein et al. (2015) as a discrete-time Markov chain of length $T$ with Gaussian transitions

$$p_{\mathbf{X}_t|\mathbf{X}_{t-1}=\mathbf{x}_{t-1}} = \mathrm{Gauss}(\sqrt{1-\beta_t}\mathbf{x}_{t-1}, \beta_t\mathbf{I}), \tag{31}$$

such that the transitions from $\mathbf{X}_0$ to $\mathbf{X}_t$ are also tractable as

$$\mathbf{X}_t = \sqrt{\bar{\alpha}_t}\mathbf{X}_0 + \sqrt{1-\bar{\alpha}_t}\mathbf{Z}_t, \tag{32}$$

where $\alpha_t = (1 - \beta_t)$, $\bar{\alpha}_t = \prod_{s=0}^t \alpha_s$, and $\mathbf{Z}_t \sim \mathrm{Gauss}(\mathbf{0}, \mathbf{I})$. By definition,

$$\mathbf{X}_t = \sqrt{1-\beta_t}\mathbf{X}_{t-1} + \sqrt{\beta_t}\mathbf{Z}_{t-1} \tag{33}$$

and a straightforward application of Tweedie's formula (6) gives that

$$\mathbb{E}[\mathbf{X}_{t-1}|\mathbf{X}_t] = \tfrac{1}{\sqrt{\alpha_t}}\big(\mathbf{X}_t + (1-\alpha_t)\nabla \log p_{\mathbf{X}_t}(\mathbf{X}_t)\big), \tag{34}$$

which leads to the DDPM backward transitions

$$\mathbf{X}_{t-1} = \tfrac{1}{\sqrt{1-\beta_t}}\big(\mathbf{X}_t + \beta_t \nabla \log p_{\mathbf{X}_t}(\mathbf{X}_t)\big) + \sqrt{\beta_t}\mathbf{Z}_t \tag{35}$$

like they appear in (7).

Now, we recall the reverse-diffusion SDE which, under our choice of the drift and diffusion coefficient, is given by

$$d\mathbf{X}_t = \big(-\tfrac{\beta(t)}{2}\mathbf{X}_t - \beta(t)\nabla \log p_{\mathbf{X}_t}(\mathbf{X}_t)\big)\,dt + \sqrt{\beta(t)}\,d\mathbf{W}_t. \tag{36}$$

A first-order step from $t$ to $(t-1)$ ($dt = -1$) of gives the Euler–Maruyama update

$$\mathbf{X}_{t-1} = \big(1 + \tfrac{\beta_t}{2}\big)\mathbf{X}_t + \beta_t \nabla \log p_{\mathbf{X}_t}(\mathbf{X}_t) + \sqrt{\beta_t}\mathbf{Z}_t, \tag{37}$$

where $\beta_t := \beta(t)$ and $\mathbf{Z}_t \sim \mathrm{Gauss}(\mathbf{0}, \mathbf{I})$.

The DDPM reverse process (35) can be related to the the Euler–Maruyama discretization of the reverse SDE (37) via Taylor expansions, since

$$\frac{1}{\sqrt{1-\beta_t}} = 1 + \frac{\beta_t}{2} + \mathcal{O}(\beta_t^2) \tag{38}$$

and

$$\frac{\beta_t}{\sqrt{1-\beta_t}} = \beta_t + \mathcal{O}(\beta_t^2) \tag{39}$$

as $\beta_t \to 0$.

## C  LÉVY PROCESSES AND INCREMENT DISTRIBUTIONS

The prior distributions in our framework are those of signals obtained by regularly spaced samples of processes with independent, stationary increments (Lévy processes and their discrete-time counterparts). We briefly recall the definition; see Unser & Tafti (2014); Sato (1999) for background and the link to infinitely divisible laws.

**Definition C.1** (Lévy process). A stochastic process $s = \{s(t) : t \geq 0\}$ is a Lévy process if

1. (anchor at the origin) It holds that $s(0) = 0$ almost surely;

2. (independent increments) for any $N \in \mathbb{N} \setminus \{0, 1\}$ and $0 \leq t_1 < t_2 < \cdots < t_N < \infty$, the increments $(s(t_2)-s(t_1)), (s(t_3)-s(t_2)), \ldots, (s(t_N)-s(t_{N-1}))$ are mutually independent;

3. (stationary increments) for any given step $h$, the increment process $u_h = \{s(t) - s(t-h) : t > h\}$ is stationary;

4. (stochastic continuity) for any $\varepsilon > 0$ and $t \geq 0$,
$$\lim_{h \to 0} \mathrm{Pr}\big(|s(t+h) - s(t)| > \varepsilon\big) = 0.$$

We form discrete and finite-length signals by sampling $s$ at integer times and stacking the values into $\mathbf{x} = (s(1), s(2), \ldots, s(d))$. Let the unit-step increments be $[\mathbf{u}]_k = (s(k) - s(k-1))$ for $k = 1, 2, \ldots, d$. By independence and stationarity, the law[6] of $[\mathbf{u}]_k$ does not depend on $k$ and we denote it $p_U$. We define the finite-difference matrix

$$\mathbf{D} = \begin{bmatrix} 1 & 0 & 0 & \cdots & 0 \\ -1 & 1 & 0 & \cdots & 0 \\ 0 & -1 & 1 & \cdots & 0 \\ \vdots & \vdots & \ddots & \ddots & 0 \\ 0 & 0 & \cdots & -1 & 1 \end{bmatrix} \tag{40}$$

---

[6]For our choices, it always has a density w.r.t. a suitable reference measure.

Table 2: Univariate distributions used throughout this work. Parameters appear in the order they are specified in this table, *e.g.* $\mathrm{Gauss}(\mu, \sigma^2)$.

| Name | Distribution | Parameter(s) | Supp. | Notation |
|---|---|---|---|---|
| Gaussian | $\frac{1}{\sqrt{2\pi\sigma^2}} \exp\left(-\frac{(x-\mu)^2}{\sigma^2}\right)$ | $\mu \in \mathbb{R}, \sigma^2 \in \mathbb{R}_{>0}$ | $\mathbb{R}$ | Gauss |
| Exponential | $\lambda \exp(-\lambda x)$ | $\lambda \in \mathbb{R}_{>0}$ | $\mathbb{R}_{\geq 0}$ | Exp |
| Laplace | $\frac{1}{2b} \exp\left(-\frac{|x|}{b}\right)$ | $b \in \mathbb{R}_{>0}$ | $\mathbb{R}$ | Laplace |
| Student-t | $\frac{\Gamma\left(\frac{\nu+1}{2}\right)}{\sqrt{\pi\nu}\Gamma\left(\frac{\nu}{2}\right)}\left(1+\frac{x^2}{\nu}\right)^{-\frac{\nu+1}{2}}$ | $\nu \in \mathbb{R}_{>0}$ | $\mathbb{R}$ | St |
| Gamma | $\frac{\beta^\alpha}{\Gamma(\alpha)} x^{\alpha-1} \exp(-\beta x)$ | $\alpha, \beta \in \mathbb{R}_{>0}$ | $\mathbb{R}_{>0}$ | Gamma |
| Gen. inv. Gaussian | $\frac{(\frac{a}{b})^{\frac{p}{2}}}{2K_p(\sqrt{ab})} x^{p-1} \exp\left(-\frac{ax+b/x}{2}\right)$ | $a, b \in \mathbb{R}_{>0}, p \in \mathbb{R}$ | $\mathbb{R}_{>0}$ | GIG |
| Bernoulli–Laplace | $\lambda\delta(x) + (1-\lambda)\frac{1}{2b}\exp(-\frac{|x|}{b})$ | $\lambda \in [0, 1], b \in \mathbb{R}_{>0}$ | $\mathbb{R}$ | BL |

Moreover, the gamma function is defined as $\Gamma(x) = \int_0^\infty t^{x-1} \exp(-t)\, \mathrm{d}t$ for any $x \in \mathbb{R}_{>0}$. The modified Bessel function of the second kind with parameter $\nu$ is denoted by $K_\nu$.

such that the increment vector satisfies

$$\mathbf{u} = \mathbf{D}\mathbf{x}. \tag{41}$$

Because $s(0) = 0$, the finite-difference matrix $\mathbf{D}$ has an initial condition that makes it invertible and $\mathbf{D}^{-1}$ is a lower-triangular matrix of ones. This also implies that for all $k = 1, 2, \ldots, d$,

$$[\mathbf{x}]_k = \sum_{n=1}^{k} [\mathbf{u}]_n, \tag{42}$$

which is a convenient way to synthesize signals once $\mathbf{u}$ is drawn. The combination of (41) with the independence of the increments implies that the density of the discrete signal is

$$p_{\mathbf{X}}(\mathbf{x}) = \prod_{k=1}^{d} p_U\big([\mathbf{D}\mathbf{x}]_k\big). \tag{43}$$

## C.1 EXTENSIONS

The approach in this paper can be extended to two- or higher-dimensional signals on grids, such as images or videos, and even to more specialized structures like signals defined over trees or graphs. The structure of the signal is effectively encoded through the choice of the matrix $\mathbf{D}$. For instance, a two-dimensional finite-difference matrix would result in a signal vector that can be interpreted as a two-dimensional image. The main additional (computational) challenge is sampling during signal generation: Whenever $\mathbf{D}$ is not trivially reducible to a one-dimensional operator, the model (43) will be overcomplete and, in general, no whitening transformation exists to decouple increments for independent sampling. The extension to higher-dimensional signals and the complications that arise in that context are rigorously treated in Kuric et al. (2025).

## C.2 LATENT DISTRIBUTIONS AND NOTATION

Some of the distributions that we rely on in this work have multiple competing parametrizations. To avoid ambiguities, we provide precise definitions of the four increment distributions that we consider in this work: Gaussian; Laplace; Student-t; and Bernoulli–Laplace (spike-and-slab). We give in Table 2 our notations of these and other distributions that we use in this work. We list in Table 3 the latent maps and conditional latent distributions that are needed for the GLM for the distributions in this work.

## D GIBBS METHODS AND SAMPLING EFFICIENCY

Gibbs methods are Markov chain Monte Carlo (MCMC) methods to sample from a joint distribution $p_{\mathbf{X}, \mathbf{z}_1, \mathbf{z}_2, \ldots, \mathbf{z}_n}$ of $(n+1)$ blocks of variables that are advantageous when the direct sampling is compu-

Table 3: Latent variable representations and conditional distributions for common distributions.

| Dist. $\phi_k$ | Latent dist. $f_k$ | Latent maps | Cond. latent dist. $p_{[\mathbf{Z}]_k \mid X=[\mathbf{Kx}]_k}$ |
|---|---|---|---|
| Gauss$(\mu,\sigma^2)$ | $\delta(0)$ | $\mu_k(z)=\mu,\ \sigma_k^2(z)=\sigma^2$ | $\delta(0)$ |
| Laplace$(b)$ | Exp$\left(\frac{1}{2b^2}\right)$ | $\mu_k(z)=0,\ \sigma_k^2(z)=z$ | GIG$\left(\frac{1}{b^2},[\mathbf{Kx}]_k^2,\frac{1}{2}\right)$ |
| St$(\nu)$ | Gamma$\left(\frac{\nu}{2},\frac{\nu}{2}\right)$ | $\mu_k(z)=0,\ \sigma_k^2(z)=\frac{1}{z}$ | Gamma$\left(\frac{\nu+1}{2},\frac{\nu+[\mathbf{Kx}]_k^2}{2}\right)$ |

---

**Algorithm 3** Latent-variable Gibbs sampling of $p_{\mathbf{X},\mathbf{Z}_1,\dots,\mathbf{Z}_n}$.

---

**Require:** Burn-in period $B \in \mathbb{N}$, number of samples $S \in \mathbb{N}$, initial point $(\mathbf{x}_0, \mathbf{z}_1, \dots \mathbf{z}_n)$.
1: **for** $k = 1, 2, \dots, B + S$ **do**
2:     $\mathbf{x}_k \sim p_{\mathbf{X}\mid\mathbf{Z}_1=\mathbf{z}_1,\dots,\mathbf{Z}_n=\mathbf{z}_n}$
3:     $\mathbf{z}_1 \sim p_{\mathbf{Z}_1\mid X=\mathbf{x}_k,\dots,\mathbf{Z}_n=\mathbf{z}_n}$           ▷ *Latent blocks do not need to be stored*
4:     $\vdots$
5: **return** $\{\mathbf{x}_{B+k}\}_{k=1}^S$

---

tationally difficult but sampling from the conditional distributions $p_{\mathbf{X}\mid\mathbf{Z}_1,\mathbf{Z}_2,\dots,\mathbf{Z}_n}, p_{\mathbf{Z}_1\mid\mathbf{X},\mathbf{Z}_2,\dots,\mathbf{Z}_n}, \cdots$ is easy. Gibbs methods cycle through the conditional distributions with repeated draws, which maintains the joint distribution invariant (Casella & George, 1992). The naming of the variables $\mathbf{X}, \mathbf{Z}_1, \mathbf{Z}_2, \dots, \mathbf{Z}_n$ is deliberately chosen to emphasize that we use *latent-variable* Gibbs methods that rely on auxiliary variables that are introduced solely to make the conditionals simple. The steps of a general latent variable Gibbs sampler are shown in Algorithm 3, where the iteration counter in the sampling of the latent variables is omitted since they need not be stored and previous iterations can immediately be overwritten.

Kuric et al. (2025) recently showed that such methods are significantly faster than other standard sampling routines that are commonly used in settings similar to the one in this paper. They report sampling efficiencies of close to 1, while alternatives, such as the Metropolis-adjusted Langevin algorithm, achieve sampling efficiencies[7] of around $1 \times 10^{-3}$. In addition, Gibbs methods require no stepsize or acceptance-rate tuning and introduce no discretization bias. These properties motivate our use of Gibbs methods for the fast and robust posterior sampling throughout this work.

Like all MCMC methods, in practice Gibbs methods benefit from the discarding of some number of initial samples (the *burn-in period*) when the initial point is located in low-density regions. After the burn-in period, the quality of the Monte Carlo estimate of any object depends on the number of samples one uses in their estimation. We discuss our choice of the burn-in period and the number of samples for the various problems in Appendix F.2.

### D.1 A GIBBS METHOD FOR BERNOULLI–LAPLACE INCREMENTS

Let $\delta$ be the Dirac distribution. Then, letting $\lambda$ be the Bernoulli parameter and $b$ the scale parameter, we note that the Bernoulli–Laplace density

$$p_U(u) = \lambda\delta(u) + (1-\lambda)\frac{b}{2}\exp(-b|u|) \tag{44}$$

admits the representation

$$p_U(u) = \int_{\mathbb{R}}\left(\sum_{v=0}^{1} p_{U\mid V=v,W=w}(u)p_V(v)\right)p_W(w)\,\mathrm{d}w, \tag{45}$$

where

$$p_V(v) = \lambda^{1-v}(1-\lambda)^v \tag{46}$$

for $v \in \{0,1\}$ is a Bernoulli distribution,

$$p_W(w) = \frac{b^2}{2}\exp\left(-\frac{b^2 w}{2}\right)\chi_{\mathbb{R}_{\geq 0}}(w) \tag{47}$$

---

[7]Sampling efficiency refers to effective samples per iteration; an efficiency of $\rho$ means roughly $1/\rho$ iterations per "effective sample" (Gelman et al., 2013, Section 11.5).

---

**Algorithm 4** Bernoulli–Laplace Gibbs sampler.

---

**Require:** Initial increments $\mathbf{u}_0 \in \mathbb{R}^d$, initial support vector $\mathbf{v} \in \mathbb{R}^d$
1: **for** $s = 1, 2, \ldots, B + S$ **do**
2:      Draw $[\mathbf{w}]_k \sim p_{W|U=[\mathbf{u}_{s-1}]_k, V=[\mathbf{v}]_k}$                ▷ *parallel over* $k$
3:      **for** $k = 1, 2, \ldots, d$ **do**
4:          Draw $[\mathbf{v}]_k \sim \text{Bernoulli}(p_k(\mathbf{v}))$
5:      Draw $\mathbf{u}_s \sim p_{\mathbf{U}|V=\mathbf{v}, W=\mathbf{w}, Y=\mathbf{y}}$
6: **return** $\{\mathbf{D}^{-1}\mathbf{u}_{B+k}\}_{k=1}^S$

---

is an exponential distribution, and

$$p_{U|V=v, W=w}(u) = \begin{cases} \delta(u) & \text{if } v = 0, \\ \text{Gauss}(0, w) & \text{if } v = 1. \end{cases} \tag{48}$$

The algorithm relies on the introduction of two latent vectors $\mathbf{v}, \mathbf{w} \in \mathbb{R}^d$ that satisfy

$$p_{\mathbf{U}|\mathbf{V}=\mathbf{v}, \mathbf{W}=\mathbf{w}}(\mathbf{u}) = \prod_{k=1}^d p_{U|V=[\mathbf{v}]_k, W=[\mathbf{w}]_k}([\mathbf{u}]_k) \tag{49}$$

such that, as a result, the distribution conditioned on the measurements can be written as

$$p_{\mathbf{U}, \mathbf{V}, \mathbf{W}|\mathbf{Y}=\mathbf{y}}(\mathbf{u}, \mathbf{v}, \mathbf{w}) \propto \exp\left(-\tfrac{1}{2\sigma_n^2}\|\mathbf{H}\mathbf{u} - \mathbf{y}\|^2\right) \prod_{k=1}^d p_{U|V=[\mathbf{v}]_k, W=[\mathbf{w}]_k}([\mathbf{u}]_k)$$

$$\times \prod_{k=1}^d \lambda^{1-[\mathbf{v}]_k}(1-\lambda)^{[\mathbf{v}]_k} \prod_{k=1}^d \frac{b^2}{2} \exp\left(-\frac{b^2[\mathbf{w}]_k}{2}\right), \tag{50}$$

where $\mathbf{H} = \mathbf{A}\mathbf{D}^{-1}$. Equations (48) and (50) imply that any sample from $p_{\mathbf{U}|\mathbf{V}=\mathbf{v}, \mathbf{W}=\mathbf{w}, \mathbf{Y}=\mathbf{y}}$ takes the value zero at those indices where $\mathbf{v}$ is zero, and values from a multivariate Gaussian distribution with covariance $\mathbf{C} = \left(\sigma_n^2\mathbf{H}\mathbf{H}^\top + \mathbf{diag}(\mathbf{w})\right)^{-1}$ and mean $\sigma_n^{-2}\mathbf{C}\mathbf{H}^T\mathbf{y}$ otherwise. Sampling $\mathbf{W} \mid \mathbf{U} = \mathbf{u}, \mathbf{V} = \mathbf{v}, \mathbf{Y} = \mathbf{y}$ amounts to the independent sampling of $d$ one-dimensional distributions, which are $\text{Exp}(2/b^2)$ at indices where $\mathbf{v}$ is zero and $\text{GIG}(b^2, [\mathbf{u}]_k^2, 0.5)$ those indices $k$ where $\mathbf{v}$ is one. The conditional distribution of the binary support vector is

$$p_{\mathbf{V}|\mathbf{W}=\mathbf{w}, \mathbf{Y}=\mathbf{y}}(\mathbf{v}) \propto |\mathbf{B}(\mathbf{v}, \mathbf{w})|^{-\frac{1}{2}} \exp\left(-\tfrac{1}{2}\mathbf{y}^\top\mathbf{B}(\mathbf{v}, \mathbf{w})^{-1}\mathbf{y}\right) \prod_{k=1}^d \lambda^{1-[\mathbf{v}]_k}(1-\lambda)^{[\mathbf{v}]_k}, \tag{51}$$

where[8] $\mathbf{B}(\mathbf{v}, \mathbf{w}) = \sigma_n^2\mathbf{I} + \mathbf{H}\mathbf{diag}(\mathbf{v} \odot \mathbf{w})\mathbf{H}^\top$. The standard way to sample from this distribution is to use a coordinate-wise Gibbs sampler that updates $[\mathbf{v}]_k \sim \text{Bernoulli}(p_k(\mathbf{v}))$ with

$$p_k(\mathbf{v}) = (1 + \exp(-\Delta_k(\mathbf{v})))^{-1} \tag{52}$$

where the log-odds increment

$$\Delta_k(\mathbf{v}) = \log\tfrac{1-\lambda}{\lambda} - \tfrac{1}{2}\left(\log|\mathbf{B}(\mathbf{v}_{k=1}, \mathbf{w})| - \log|\mathbf{B}(\mathbf{v}_{k=0}, \mathbf{w})|\right)$$

$$- \tfrac{1}{2}\left(\mathbf{y}^\top\mathbf{B}(\mathbf{v}_{k=1}, \mathbf{w})^{-1}\mathbf{y} - \mathbf{y}^\top\mathbf{B}(\mathbf{v}_{k=0}, \mathbf{w})^{-1})\mathbf{y}\right), \tag{53}$$

where $\mathbf{v}_{k=\cdot} := (\mathbf{v}_1, \ldots, \mathbf{v}_{k-1}, \cdot, \mathbf{v}_{k+1}, \ldots, \mathbf{v}_d)$ is the difference between the log-posterior when the bit is on and when it is off. The resulting algorithm is summarized in Algorithm 4 and can be interpreted[9] as $(d + 2)$-block (*i.e.*, dimension-dependent) Gibbs method.

---

[8]This formulation is equivalent to the one presented by Bohra et al. (2023), who explicitly "slice" the matrices $\mathbf{H}$ and $\mathbf{diag}(\mathbf{w})$ with the indices where $\mathbf{v}$ is one. We stick to this formulation since it requires less notation and emphasizes that implementations need not build variable-sized matrices, which is crucial for an efficient implementation on modern compute units that utilize highly parallelized computations.

[9]This is only an interpretation because the density violates the classical positivity conditions that are needed for Gibbs methods. It is a *partially collapsed* Gibbs method, see (Bohra et al., 2023; van Dyk & Park, 2008).

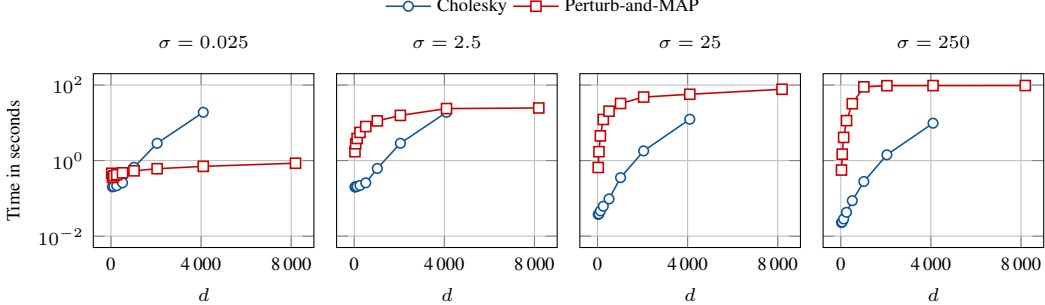

Figure 6: Runtimes needed to perform 20 Gibbs iterations on a denoising posterior ($\mathrm{Laplace}(1)$ increment distribution, 10 parallel chains) depending on the dimensionality of the signal. Missing entries are due to excessive memory requirements.

### D.2 PRACTICAL GIBBS IMPLEMENTATIONS

**Sampling the Gaussians** The sampling of $\mathbf{X} \mid \mathbf{Z}$ in the GLM and of $\mathbf{U} \mid \mathbf{V}, \mathbf{W}, \mathbf{Y}$ for the Bernoulli–Laplace case reduces to drawing from a high-dimensional Gaussian, which is a well-studied problem. For settings that necessitate a matrix-free implementation such as those that are commonly encountered in imaging applications, Kuric et al. (2025) advocate a Perturb-and-MAP sampler with preconditioned conjugate-gradient solvers. We report the runtime of the Gibbs method as a function of signal dimension for a $\mathrm{Laplace}(1)$ increment distribution in Figure 6. A standard implementation based on a Cholesky factorization of the covariance matrix—which requires explicitly instantiating the matrices $\mathbf{A}$ and $\mathbf{D}$ in memory—is faster than the Perturb-and-MAP sampler with a conjugate-gradient solver across a broad range of noise variances and dimensions. For our moderate-dimensional setting with $d = 64$, the Cholesky-based implementation is approximately an order of magnitude faster.

However, explicitly storing these matrices becomes infeasible at larger dimensions (in our setup, we ran out of memory at $d = 8096$), and the expected cubic scaling is apparent in the figure. In contrast, the Perturb-and-MAP sampler (convergence criterion: squared residual norm below $1 \times 10^{-6}$), while slower than Cholesky at small dimensions, exhibits substantially better scaling with signal dimension. In particular, it does not require materializing the operators: both the measurement operator $\mathbf{A}$ and the finite-difference operator $\mathbf{D}$ can be implemented efficiently in a matrix-free manner. Moreover, the sublinear runtime observed in this experiment suggests that the corresponding linear systems are well conditioned.

The sampling accuracy of Perturb-and-MAP depends on the termination criterion used by the optimization solver, and any finite stopping rule yields approximate samples. A principled refinement is to incorporate a Metropolis–Hastings correction step to remove bias, and to tune the solver accuracy to optimize overall runtime; this strategy was proposed by Gilavert et al. (2015), to which we refer for details. Overall, these results indicate that the Gibbs method scales favorably to higher dimensions. Combined with the fact that the denoising posteriors are sub-Gaussian, and with the relatively mild sample-complexity requirements for estimator accuracy in this setting, this suggests that the proposed framework scales well as the dimension increases.

**Sampling the Latent Variables** The sampling of the latent variables necessitates the sampling of the one-dimensional conditional latent distributions. All the conditional latent distributions that are relevant in this paper admit efficient samplers that are readily available in standard scientific computing packages or can be implemented with little effort. We reuse the `CUDA` implementation of the generalized inverse Gaussian sampler from Kuric et al. (2025) that implements the method proposed by Devroye (2012) and rely on `PyTorch` (Paszke et al., 2017) for all others. Wherever possible, latent updates are parallelized.

In the Gibbs methods for the Bernoulli–Laplace increments, the sequential drawing of the binary support vector $\mathbf{V}$ is embedded in the outer Gibbs loop which, in turn, may be embedded in the reverse-diffusion loop. This makes it crucial to minimize the use of heavy linear-algebra operations to achieve acceptable runtimes. Writing $\mathbf{B}(\mathbf{v}, \mathbf{w}) = \sigma_{\mathrm{n}}^2 \mathbf{I} + \mathbf{H}\mathrm{diag}(\mathbf{v} \odot \mathbf{w})\mathbf{H}^{\top}$, we recognize that

the flipping of the $k$th bit of $\mathbf{v}$ adds or removes a rank-one term $[\mathbf{w}]_k\mathbf{H}_k\mathbf{H}_k^\top$, where $\mathbf{H}_k$ is the $k$th column of $\mathbf{H}$. Using the matrix-determinant lemma and Woodbury–Sherman–Morrison, we update

$$\log|\mathbf{B}(\mathbf{v}_{k=1},\mathbf{w})| = \log|\mathbf{B}(\mathbf{v}_{k=0},\mathbf{w})| + \log(1+[\mathbf{w}]_k\tau_k) \tag{54}$$

and

$$\mathbf{y}^\top\mathbf{B}(\mathbf{v}_{k=1},\mathbf{w})^{-1}\mathbf{y} = \mathbf{y}^\top\mathbf{B}(\mathbf{v}_{k=0},\mathbf{w})^{-1}\mathbf{y} - \frac{[\mathbf{w}]_k(\mathbf{H}_k^\top\mathbf{B}(\mathbf{v}_{k=0},\mathbf{w})^{-1}\mathbf{y})^2}{1+[\mathbf{w}]_k\tau_k}, \tag{55}$$

where $\tau_k = \mathbf{H}_k^\top\mathbf{B}(\mathbf{v}_{k=0},\mathbf{w})^{-1}\mathbf{H}_k$. Thus, an efficient implementation factors $\mathbf{B}(\mathbf{v},\mathbf{w})$ once per latent state, obtains the needed scalars via triangular solves, and performs rank-one updates as bits flip. We report our cumulative runtime improvement over a naive implementation in Figure 4.

## E  DPS UPDATE STEPS

### E.1  COVARIANCE IN C-DPS

C-DPS (Chung et al., 2023) uses the approximation of the likelihood

$$p_{\mathbf{Y}|\mathbf{X}_t=\mathbf{x}}(\mathbf{y}) \approx p_{\mathbf{Y}|\mathbf{X}_0=\mathbb{E}[\mathbf{X}_0|\mathbf{X}_t=\mathbf{x}]}(\mathbf{y}). \tag{56}$$

When the noise in the inverse problem is Gaussian, the likelihood score $\nabla(\mathbf{x}\mapsto\log p_{\mathbf{Y}|\mathbf{X}_0=\mathbb{E}[\mathbf{X}_0|\mathbf{X}_t=\mathbf{x}]}(\mathbf{y}))$ necessitates the computation of

$$\nabla\left(\mathbf{x}\mapsto\tfrac{1}{2}\|\mathbf{A}\mathbb{E}[\mathbf{X}_0\mid\mathbf{X}_t=\mathbf{x}]-\mathbf{y}\|^2\right), \tag{57}$$

which is

$$\mathbf{J}\left(\mathbf{x}\mapsto\mathbb{E}[\mathbf{X}_0\mid\mathbf{X}_t=\mathbf{x}]\right)(\cdot)\mathbf{A}^\top(\mathbf{A}\mathbb{E}[\mathbf{X}_0\mid\mathbf{X}_t=\cdot]-\mathbf{y}) \tag{58}$$

after an application of the chain rule. The Jacobian $\mathbf{J}\left(\mathbf{x}\mapsto\mathbb{E}[\mathbf{X}_0\mid\mathbf{X}_t=\mathbf{x}]\right)$ is typically computed with automatic differentiation when $(\mathbf{x},t)\mapsto\mathbb{E}[\mathbf{X}_0\mid\mathbf{X}_t=\mathbf{x}]$ is approximated with a neural network. In our framework, we use the connection with the covariance matrix $\mathrm{Cov}[\mathbf{X}_0\mid\mathbf{X}_t=\cdot]$. Indeed, as also shown by Rissanen et al. (2025), if $\mathbf{X}_0$ and $\mathbf{X}_t$ verify (32), then

$$\tfrac{1}{1-\bar{\alpha}_t}\mathrm{Cov}[\mathbf{X}_0\mid\mathbf{X}_t=\mathbf{x}] = \tfrac{1}{\bar{\alpha}_t}\left(\mathbf{I}+(1-\bar{\alpha}_t)^2\nabla^2\log p_{\mathbf{X}_t}(\mathbf{x})\right). \tag{59}$$

This identity, combined with the derivative of (6), yields

$$\mathbf{J}\left(\mathbf{x}\mapsto\mathbb{E}[\mathbf{X}_0\mid\mathbf{X}_t=\mathbf{x}]\right)(\mathbf{x}_t) = \frac{\sqrt{\bar{\alpha}_t}}{1-\bar{\alpha}_t}\mathrm{Cov}[\mathbf{X}_0\mid\mathbf{X}_t=\mathbf{x}_t]. \tag{60}$$

### E.2  EXPLICIT UPDATE STEPS

We give the instantiations of the update step $\mathcal{S}$ a variety of DPS algorithms below. Each $\mathbf{z}_t$ is a $d$-dimensional random vector with i.i.d. standard Gaussian entries.

**Score-ALD (Jalal et al., 2021)**  The input parameters of this algorithm are composed of the following: A noise schedule $\{\beta_t\}_{t=0}^{T-1}$, the noise level of the inverse problem $\sigma_\mathrm{n}$, and annealing parameters $\{\eta_t\}_{t=0}^{T-1}$ and $\{\gamma_t\}_{t=0}^{T-1}$. The update step goes

$$\begin{aligned}
\bar{\boldsymbol{\mu}} &= \frac{1}{S}\sum_{s=1}^{S}\bar{\mathbf{x}}_s,\\
\mathbf{s}_t &= (\bar{\boldsymbol{\mu}}-\mathbf{x}_t)/\beta_t^2,\\
\mathbf{x}_{t-1} &= \mathbf{x}_t+\eta_t\big(\mathbf{s}_t+\tfrac{1}{\gamma_t^2+\sigma_\mathrm{n}^2}\mathbf{A}^\top(\mathbf{y}-\mathbf{A}\mathbf{x}_t)\big)+\sqrt{2\eta_t}\mathbf{z}_t.
\end{aligned} \tag{61}$$

**C-DPS (Chung et al., 2023)**  The input parameters are the the variance-preserving scaling weight $\bar{\alpha}_t$ as in (32), the variance of the diffusion transitions $\beta_t$ as in (35), and a scalar $\zeta$ that governs the

likelihood-guidance strength. The diffusion noise level that corresponds to the denoising posterior is denoted $\sigma_t = (1 - \bar{\alpha}_t)/\sqrt{\bar{\alpha}_t}$, which is used to compute the samples $\{\bar{\mathbf{x}}_k\}_{k=1}^{S}$. The update step goes

$$
\begin{aligned}
\bar{\boldsymbol{\mu}} &= \frac{1}{S} \sum_{k=1}^{S} \bar{\mathbf{x}}_s, \\
\mathbf{C} &= \frac{1}{S} \sum_{k=1}^{S} (\bar{\mathbf{x}}_k - \bar{\boldsymbol{\mu}})(\bar{\mathbf{x}}_k - \bar{\boldsymbol{\mu}})^\top, \\
\mathbf{x}'_{t-1} &= \frac{\sqrt{\bar{\alpha}_t}(1 - \bar{\alpha}_{t-1})}{1 - \bar{\alpha}_t} \mathbf{x}_t + \frac{\sqrt{\bar{\alpha}_{t-1}} \beta_t}{1 - \bar{\alpha}_t} \bar{\boldsymbol{\mu}} + \sigma_t \mathbf{z}_t, \\
\tilde{\mathbf{x}}_{t-1} &= \mathbf{x}'_{t-1} - \frac{\zeta}{\|\mathbf{A}\bar{\boldsymbol{\mu}} - \mathbf{y}\|} \frac{\sqrt{\bar{\alpha}_t}}{1 - \bar{\alpha}_t} \mathbf{C}^\top \mathbf{A}^\top (\mathbf{A}\bar{\boldsymbol{\mu}} - \mathbf{y}), \\
\mathbf{x}_{t-1} &= \tilde{\mathbf{x}}_{t-1}/\sqrt{\bar{\alpha}_{t-1}}.
\end{aligned}
\tag{62}
$$

**DiffPIR (Zhu et al., 2023)** The input parameters are similar to those of C-DPS. It also uses the noise level of the inverse problem $\sigma_{\mathrm{n}}$ and an additional balance hyperparameter $\gamma$. The update step goes

$$
\begin{aligned}
\bar{\boldsymbol{\mu}} &= \frac{1}{S} \sum_{k=1}^{S} \bar{\mathbf{x}}_k, \\
\rho_t &= \zeta \frac{\sigma_{\mathrm{n}}^2}{\sigma_t^2}, \\
\bar{\mathbf{x}}_0 &= \arg\min_{\mathbf{x} \in \mathbb{R}^d} \left( \frac{1}{2} \|\mathbf{A}\mathbf{x} - \mathbf{y}\|^2 + \frac{\rho_t}{2} \|\mathbf{x} - \bar{\boldsymbol{\mu}}\|^2 \right), \\
\hat{\epsilon} &= \frac{1}{\sqrt{1 - \bar{\alpha}_t}} \left( \mathbf{x}_t - \sqrt{\bar{\alpha}_t} \bar{\mathbf{x}}_0 \right), \\
\tilde{\mathbf{x}}_{t-1} &= \sqrt{\bar{\alpha}_{t-1}} \bar{\mathbf{x}}_0 + \sqrt{1 - \bar{\alpha}_{t-1}} (\sqrt{1 - \gamma} \hat{\epsilon} + \sqrt{\gamma} \mathbf{z}_t), \\
\mathbf{x}_{t-1} &= \tilde{\mathbf{x}}_{t-1}/\sqrt{\bar{\alpha}_{t-1}}.
\end{aligned}
\tag{63}
$$

**ΠGDM (Song et al., 2023)** The input parameters are the noise schedule $\{\sigma_t\}_{t=0}^{T-1}$, the data-dependent noise schedule $\{r_t\}_{t=0}^{T-1}$, and the DDIM (Song et al., 2020) time-dependent coefficients $\{c_t^{(1)}\}_{t=0}^{T-1}$ and $\{c_t^{(2)}\}_{t=0}^{T-1}$. The update step goes

$$
\begin{aligned}
\bar{\boldsymbol{\mu}} &= \frac{1}{S} \sum_{k=1}^{S} \bar{\mathbf{x}}_k, \\
\mathbf{C} &= \frac{1}{S} \sum_{s=1}^{S} (\bar{\mathbf{x}}_k - \bar{\boldsymbol{\mu}})(\bar{\mathbf{x}}_k - \bar{\boldsymbol{\mu}})^\top, \\
\mathbf{J} &= \frac{\sqrt{\bar{\alpha}_t}}{1 - \bar{\alpha}_t} \mathbf{C}, \\
\mathbf{g} &= \left( (\mathbf{y} - \mathbf{A}\bar{\boldsymbol{\mu}})^\top \left( \mathbf{A}\mathbf{A}^\top + \frac{\sigma_{\mathrm{n}}^2}{r_t^2} \mathbf{I} \right)^{-1} \mathbf{A}\mathbf{J} \right)^\top, \\
\hat{\epsilon} &= \frac{1}{\sqrt{1 - \bar{\alpha}_t}} \left( \mathbf{x}_t - \sqrt{\bar{\alpha}_t} \bar{\boldsymbol{\mu}} \right), \\
\tilde{\mathbf{x}}_{t-1} &= \sqrt{\alpha_s} \bar{\boldsymbol{\mu}} + c_1 \mathbf{z}_t + c_2 \hat{\epsilon} + \sqrt{\alpha_t} \mathbf{g}, \\
\mathbf{x}_{t-1} &= \tilde{\mathbf{x}}_{t-1}/\sqrt{\bar{\alpha}_{t-1}}.
\end{aligned}
\tag{64}
$$

**DPnP (Xu & Chi, 2024)** The diffusion noise level that corresponds to the denoising posterior is denoted $\eta_t$, which is used to compute the sample $\bar{\mathbf{x}}_1$. This same $\eta_t$ defines the likelihood-guidance strength. The update step goes

$$
\begin{aligned}
\mathbf{x}_0 &= \bar{\mathbf{x}}_1, \\
\mathbf{x}_{t-1} &\sim \exp\left( -\frac{1}{2} \|\mathbf{A} \cdot - \mathbf{y}\|^2 - \frac{1}{2\eta_t^2} \| \cdot - \mathbf{x}_0 \|^2 \right).
\end{aligned}
\tag{65}
$$

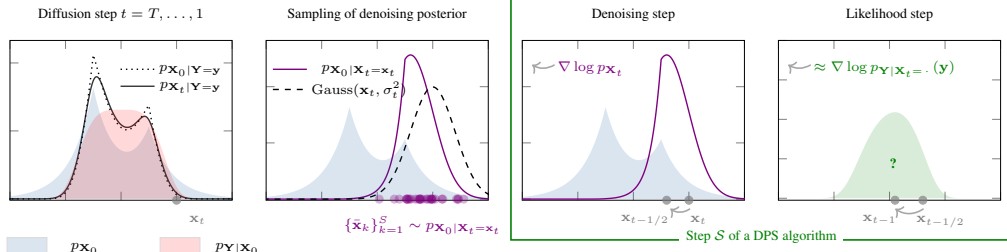

Figure 7: Illustration of the proposed template for DPS algorithms.

**Annealed Plug-and-Play Monte Carlo (PnP and RED variants)** (Sun et al., 2024)    The diffusion noise level that corresponds to the denoising posterior is denoted $\sigma_t$, which are used to compute the samples $\{\bar{\mathbf{x}}_k\}_{k=1}^S$. The parameter $\eta$ denotes the likelihood guidance strength, and $\kappa_t$ is an annealing parameter.

The update step for the PnP variant goes

$$
\begin{aligned}
\bar{\boldsymbol{\mu}} &= \frac{1}{S}\sum_{k=1}^S \bar{\mathbf{x}}_k, \\
\mathbf{s} &= (\bar{\boldsymbol{\mu}} - \mathbf{x}_t)/\sigma_t^2, \\
\mathbf{x}_{t-1} &= \mathbf{x}_t + \gamma\kappa_t\mathbf{s} + \sqrt{2\gamma}\mathbf{z}_t, \\
\mathbf{x}_t &\leftarrow \mathbf{x}_t - \gamma\mathbf{A}^\top(\mathbf{A}\mathbf{x} - \mathbf{y}).
\end{aligned}
\tag{66}
$$

The update step for the RED variant goes

$$
\begin{aligned}
\bar{\boldsymbol{\mu}} &= \frac{1}{S}\sum_{k=1}^S \bar{\mathbf{x}}_k, \\
\mathbf{s} &= (\bar{\boldsymbol{\mu}} - \mathbf{x}_t)/\sigma_t^2, \\
\mathbf{x}_{t-1} &= \mathbf{x}_t - \gamma\big(\mathbf{A}^\top(\mathbf{A}\mathbf{x} - \mathbf{y}) - \kappa_t\mathbf{s}\big) + \sqrt{2\gamma}\mathbf{z}_t.
\end{aligned}
\tag{67}
$$

The DPS template that is summarized in Algorithm 2 is illustrated with a one-dimensional toy-example in Figure 7.

## F    NUMERICAL EXPERIMENTS

### F.1    FORWARD OPERATORS

We consider four forward operators $\mathbf{A}$ in our experiments. The first operator is the identity $\mathbf{A} = \mathbf{I} \in \mathbb{R}^{d\times d}$. This choice is motivated by the fundamental role that denoising algorithms currently play in many restoration algorithms and even in labeling problems such as edge detection (Le et al., 2025). The second operator $\mathbf{A} \in \mathbb{R}^{d\times d}$ implements the convolution with a kernel that consists of the 13 central samples of a truncated Gaussian with variance 2 that are normalized to unit sum. We adopt circular boundary conditions to enable a fast computation of the proximal map that arises in the update step of DiffPIR (see Appendix E.2) via the fast Fourier transform. Deconvolution is a relevant problem with applications like microscopy or astronomy. The third operator is a sampling operator $\mathbf{A} \in \mathbb{R}^{m\times d}$ that returns $m < d$ entries of its argument unchanged. This operator is relevant in many fields such as image reconstruction and time-series forecasting. In particular, in a forecasting or prediction problem the operator would return the first $m$ known entries, and the resolution of the inverse problem estimates the remaining $(d - m)$ entries. In our experiments, each entry has an independent chance of 40 % of being kept. The fourth and last operator is $\mathbf{A} = \mathbf{MF} \in \mathbb{R}^{m\times d}$, where $\mathbf{F} \in \mathbb{R}^{2(\lfloor d/2\rfloor+1)\times d}$ is the matrix representation of the "real" one-dimensional discrete Fourier transform with separated real and imaginary components, and $\mathbf{M} \in \mathbb{R}^{m\times 2(\lfloor d/2\rfloor+1)}$ is a sampling operator. Such operators are relevant in medical imaging or astronomy. The sampling operator is constructed such that the 5 lowest

frequencies (the DC term included) are acquired, while the remaining frequencies independently have a $40\,\%$ chance of being kept.

For all operators, the noise variance $\sigma_\mathrm{n}^2$ is chosen such that the median measurement signal-to-noise ratio (SNR) is around $25\,\mathrm{dB}$. We set $N_\mathrm{train} = 1 \times 10^6$, $N_\mathrm{val} = 1 \times 10^3$, and $N_\mathrm{test} = 1 \times 10^3$.

## F.2 Benchmark Implementation

The benchmarking pipeline starts with the generation of $N_\mathrm{test}$ test signals denoted $\{\mathbf{x}_k^\mathrm{test}\}_{k=1}^{N_\mathrm{test}}$ per increment distribution, each of which is independently synthesized by first drawing i.i.d. increments from the respective increment distribution and forming the signals via (42). It then proceeds to synthesize the $N_\mathrm{test}$ measurements (*i.e.* we use one noise instance per signal) denoted $\{\mathbf{y}_k^\mathrm{test}\}_{k=1}^{N_\mathrm{test}}$ according to (1) and, for each of the measurements, computes the gold-standard posterior samples of the various inverse problems via the Gibbs methods described in Section 3. This stage is off-line (no reverse-diffusion loop) and trivially parallel across the measurements, which allows us to run long chains with burn-in periods of $1 \times 10^5$ iterations and obtain $2 \times 10^5$ draws from the posterior distribution. This far exceeds any values reported by Kuric et al. (2025) or Bohra et al. (2023) and results in precise MMSE estimates.

The dataset-generation stage also involves the generation of $N_\mathrm{train}$ training signals $\{\mathbf{x}_k^\mathrm{train}\}_{k=1}^{N_\mathrm{train}}$ and $N_\mathrm{val}$ validation signals (mutually disjoint from the test signals) $\{\mathbf{x}_k^\mathrm{val}\}_{k=1}^{N_\mathrm{val}}$, along with the corresponding validation measurements $\{\mathbf{y}_k^\mathrm{val}\}_{k=1}^{N_\mathrm{val}}$. The training signals are used for the learning of a neural score function like those that are used for the resolution of inverse problems when the prior is unknown or too expensive to evaluate. Training details are provided in Appendix F.3 The validation signals are used to monitor the performance of the neural score function on unseen signals during the training stage and to tune the regularization parameters for the model-based approaches as well as the parameters of the DPS algorithms, see Section 4.1 .

Unlike for the computation of the gold-standard MMSE estimate of the initial inverse problem, the denoising posteriors are sampled $T$ times per trajectory (we use $T = 1000$). To ensure acceptable runtimes in this setting, we therefore pick the smallest burn-in period and sample count that still yield accurate estimates of the required statistics. We determine these settings with a rigorous protocol that is detailed in Appendix F.4. Ultimately, this protocol resulted in the choice of a burn-in period of 100 iterations and a sample count of 300.

## F.3 Learning Details

For learned-based denoisers, a noise-conditional neural network with UNet architecture (305 761 learnable parameters) is trained in an off-line step on the $N_\mathrm{train}$ training signals in a standard setup (Adam optimizer with learning rate $1 \times 10^{-4}$ with exponential decay with factor 0.9999, 100 000 parameter updates, batch size 10 000). The noise schedule in C-DPS and DiffPIR is defined by the two endpoints $\beta_0 = 1 \times 10^{-4}$ and $\beta_T = 2 \times 10^{-2}$ with linear equidistant samples in-between. The learned variant of DPnP is the "DDS-DDPM" variant (Xu & Chi, 2024, Algorithms 1 and 3) that contains an inner denoising-sampling loop. The arbitrary-precision variant does not require an inner loop at all (except for the burn-in period), which makes the arbitrary-precision variant the faster one for this case.

## F.4 Burn-In Period and the Number of Samples

As discussed in Appendix F.2, the burn-in period and the number of samples of the Gibbs samplers needs to be chosen appropriately to ensure an acceptable runtimes and a sufficiently small Monte Carlo error. We determine the burn-in period and the number of samples through the following protocol that is run in an off-line stage prior to running the benchmark. We synthesize $\mathbf{x}_t = \mathbf{x}_0 + \sigma_t \mathbf{n}$ where $\sigma_t$ is in the range defined by the noise schedule $\beta$, $\mathbf{x}_0$ is constructed via (42) for all four considered increment distributions, and $\mathbf{n}$ is some unknown but fixed vector of standard Gaussian noise. For each of the synthesized signals, we then launch $C = 1000$ parallel Gibbs chains on the corresponding denoising posterior and run those chains for $N_\mathrm{sufficient}$ iterations, where $N_\mathrm{sufficient}$ is a sufficiently large natural number that guarantees that the chains are stationary for at least $N_\mathrm{avg}$ (which is also relatively large) iterations and that, consequently, we can compute precise estimates of

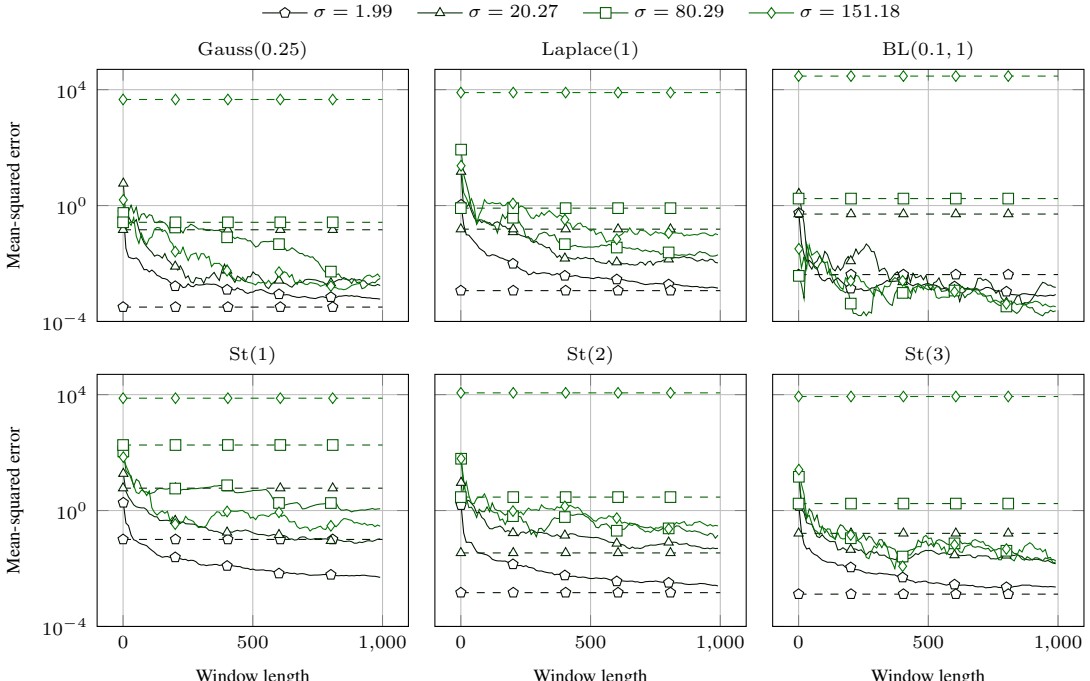

Figure 9: Mean squared error between MMSE estimates and the reference MMSE. Dashed lines: Learned neural MMSE estimate. Solid lines: Monte Carlo MMSE estimate in terms of the window length.

various statistics of the posterior distribution from the iterates from the last $N_{\mathrm{avg}}$ iterations across all $C$ chains.

To determine the burn-in period, we then proceed to calculate a statistic that we can monitor throughout the iterations and that we can compare against the reference statistic. Specifically, denoting with $\mathbf{X}$ the random variable of the Gibbs sampler, we compute the empirical distribution of the increments at index 32 like $(\mathbf{X}_{33} - \mathbf{X}_{32})$. The distribution of differences that is obtained by taking the last $N_{\mathrm{avg}}$ iterations across all $C$ chains is considered the reference distribution. Then, we compute the Wasserstein-1 distance of that distribution to the one obtained by taking the average across $N_{\mathrm{avg}}$ iterations and all $C$ in a sliding-window starting from the first Gibbs iterations. This allows us to gauge the burn-in period through a visual inspection of the

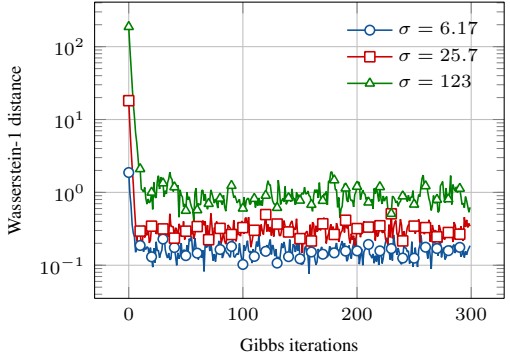

Figure 8: Wasserstein-1 distance of intermediate marginal distributions to that of the final sample.

Wasserstein-1 distance through the Gibbs iterations. In particular, we expect the Wasserstein-1 distance to be large for a number of initial samples where the Gibbs sampler is not stationary and then to oscillate around a small but nonzero value. The value will be nonzero due to the finite sample size. The Wasserstein-1 distance between the reference distribution and the one obtained through the Gibbs iterations is shown in Figure 8 (for the exemplary case of a $\mathrm{St}(1)$ distribution and a selection of noise variances). We observe that the empirical distribution of increments converges rapidly to the reference one. The Wasserstein-1 distance reaches the noise level after a single-digit number of iterations, which is in line with the analysis provided by Kuric et al. (2025). Based on these findings, we chose the burn-in period as $B = 100$ iterations for all our experiments, which is more than sufficient to reach stationarity and has acceptable runtime.

To determine the number of samples that are needed for a sufficiently accurate computation of various statistics that any DPS algorithm may utilize in their update steps, we compute a precise estimation of the MMSE estimate of the denoising posterior by averaging the last $N_{\text{avg}}$ iterations across all $C$ chains. Then, we pick one arbitrary chain and grow a window from iteration $(N_{\text{avg}} - 1)$ to the left, average the samples in that window, and compute the MSE from the MMSE estimates obtained in the one-chain window to the precise estimate obtained by averaging the $C$ chains and the last $N_{\text{avg}}$ iterations. We show this error in terms of the window length and the noise variance for all increment distributions in Figure 9. The quality of the learned denoiser and the Monte Carlo denoiser differ over the noise variances and the learned denoiser improves relative to the Monte Carlo denoiser as the noise variance vanishes. Our final choice of $S = 300$ samples is motivated by the fact that the quality of the Monte Carlo denoiser, *when averaged across all noise variances that appear in the reverse diffusion*, is always strictly better than the learned denoiser. Since it is relevant for the discussion in Section 4.2, we highlight that for this choice the quality of the Monte Carlo denoiser is superior to the learned one *across all noise variances* for the $\text{St}(1)$ and $\text{BL}(0.1, 1)$ increment distributions.

### F.5 ALGORITHM PARAMETERS

The adjustable regularization parameter for $\text{est} \in \{\ell_2, \ell_1\}$ satisfies

$$\lambda^{\text{est},\star} = \arg\min_{\lambda \in \Lambda} \frac{1}{N_{\text{val}}} \sum_{k=1}^{N_{\text{val}}} \frac{1}{d} \|\hat{\mathbf{x}}^{\text{est}}(\mathbf{y}_k^{\text{val}}, \lambda) - \mathbf{x}_k^{\text{val}}\|^2. \tag{68}$$

There, $\Lambda$ is the loglinear grid $\Lambda = \{\lambda_1, \lambda_2, \ldots, \lambda_{N_{\text{mb}}}\}$ with

$$\lambda_n = 10^{a + (n-1) \frac{(b-a)}{N_{\text{mb}} - 1}} \tag{69}$$

with $a = (-5)$ and $b = 5$. Since the model-based methods are very fast, we can use the relatively high $N_{\text{mb}} = 1000$.

The adjustable hyperparameters of the DPS methods were found by

$$\boldsymbol{\lambda}^{\text{alg},\star} = \arg\min_{\boldsymbol{\lambda} \in \Theta^{\text{alg}}} \frac{1}{N_{\text{val}}} \sum_{k=1}^{N_{\text{val}}} \frac{1}{d} \|\hat{\mathbf{x}}_{\text{MMSE}}^{\text{alg}}(\mathbf{y}_k^{\text{val}}, \boldsymbol{\lambda}) - \mathbf{x}_k^{\text{val}}\|^2 \tag{70}$$

where the grid $\Theta^{\text{alg}}$ is method-dependent. This tuning is tailored to the evaluation with respect to the MMSE optimality gap. Due to resource constraints, the parameters are tuned for the learned denoiser. We use $N_{\text{samples}} = 10$ for the grid search on the validation set. We define a modest number of $N_{\text{dps}} = 40$ grid-points and found the extreme points of the grid (*i.e.*, the values of the parameters that clearly lead to worse results) by hand. For C-DPS and DiffPIR, we fix the diffusion schedule to standard choices ($\beta_0 = 1 \times 10^{-4}, \beta_T = 0.02$). In addition to the diffusion schedule, C-DPS has one tunable parameter $\gamma$ that we tune on 40 loglinear grid points ($n = 1, \ldots, N_{\text{dps}}$)

$$10^{a + (n-1) \frac{(b-a)}{N_{\text{dps}} - 1}}. \tag{71}$$

There, $a = (-3)$ and $b = 1$. DiffPIR has two tunable parameters $\gamma$ and $\zeta$, with $\gamma$ being typically considered uncritical. Thus, we split the 40 grid points into a two-dimensional grid $\Theta^{\text{DiffPIR}} = \{0.3, 0.7\} \times \Theta^{\zeta}$, with 2 points for $\gamma$ and 20 points for $\zeta$ given by $\Theta^{\zeta} = \{\Theta_1^{\zeta}, \ldots, \Theta_{N_{\text{dps}}/2}^{\zeta}\}$, where

$$\Theta_n^{\zeta} = 10^{a + (n-1) \frac{(b-a)}{(N_{\text{dps}}/2) - 1}} \tag{72}$$

with $a = (-4)$ and $b = 1$. The DPnP algorithm only has the schedule $\{\eta_t\}_{t=1}^T$ to tune. In this case, since DPnP is asymptotically correct, the schedule is a practical vehicle that enables to trade off between speed and accuracy. Therefore, the schedule of this paper is similar to the one that was proposed by (Xu & Chi, 2024): We fix a small $\eta_{\text{final}} = 0.15$ and linearly decrease $\eta$ from some $\eta_{\text{initial}}$ to $\eta_{\text{final}}$ after $K/5$ initial iterations with $\eta_{\text{initial}}$, like

$$\eta_n = \begin{cases} \eta_{\text{initial}} & \text{if } n = 1, \ldots, K/5 \\ \frac{\eta_{\text{final}}}{\eta_{\text{initial}}}^{\frac{i - K/5}{K - K/5}} \eta_{\text{initial}} & \text{if } n = K/5 + 1, \ldots, K \end{cases} \tag{73}$$

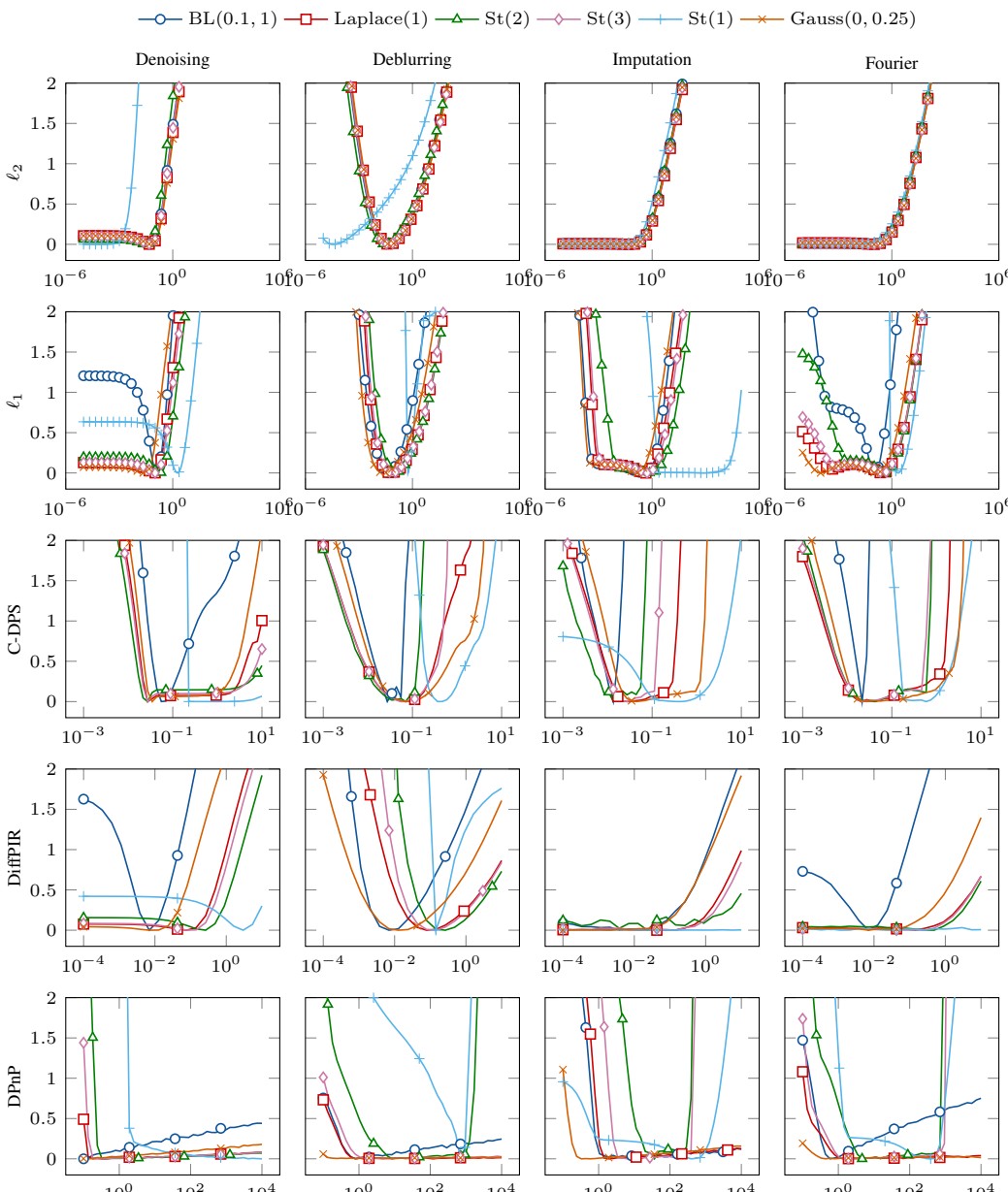

Figure 10: Grid-search diagnostics (logarithm of the MSE over the validation data set) for the model-based methods and the DPS algorithms. Rows: $\ell_2$; $\ell_1$; C-DPS; DiffPIR; DPnP. Columns: Denoising; deconvolution; imputation; reconstruction from partial Fourier measurements. For better visualization, each curve has had its minimum subtracted. To limit clutter, marks are spaced ten apart.

We treat $\eta_{\text{initial}}$ as a tunable parameter and search over $\Theta^{\text{DPnP}} = \{\eta_1, \eta_2, \ldots, \eta_{40}\}$, where

$$\eta_n = 10^{a + (n-1)\frac{(b-a)}{40-1}}. \tag{74}$$

There, $a = (-1)$ and $b = 4$. Like in the original publication, we use the comparatively small $K = 40$.

The MSE over the validation data depending on the value of the adjustable regularization parameter of the $\ell_2$ and $\ell_1$ estimators and the adjustable hyperparameters of C-DPS, DiffPIR, and DPnP is shown in Figure 10. Since the $\gamma$ parameter of DiffPIR is assumed to be uncritical, we only show the values of the MSE for various choices of $\zeta$, where $\gamma$ is set to the value of the optimal $(\gamma, \zeta)$ pair.

Table 4: Change in MMSE optimality gap (mean $\pm$ standard deviation) after substituting the learned denoiser with the arbitrary-precision denoiser. An asterisk indicates a significant changes according to a Wilcoxon signed-rank test ($p = 0.05$). Negative number with asterisk: MMSE estimates obtained with the arbitrary-precision denoiser are significantly better. Positive number with asterisk: MMSE estimates obtained with the learned denoiser are significantly better.

| | | Gauss(0.25) | Laplace(1) | BL(0.1, 1) | St(1) | St(2) | St(3) |
|---|---|---|---|---|---|---|---|
| Denoising | C-DPS | $0.00 \pm 0.11$ | $0.00 \pm 0.16$ | $-0.46 \pm 1.16^*$ | $0.00 \pm 0.01$ | $0.02 \pm 0.79^*$ | $-0.01 \pm 0.14$ |
| | DiffPIR | $0.00 \pm 0.13$ | $0.00 \pm 0.17$ | $-0.05 \pm 0.78^*$ | $-0.41 \pm 0.80^*$ | $0.00 \pm 0.20$ | $0.00 \pm 0.15$ |
| | DPnP | $0.04 \pm 0.27^*$ | $-0.01 \pm 0.22$ | $-0.55 \pm 1.31^*$ | $-0.77 \pm 1.31^*$ | $0.00 \pm 0.24$ | $0.00 \pm 0.23$ |
| Deconvolution | C-DPS | $-0.01 \pm 0.24$ | $0.00 \pm 0.26$ | $0.09 \pm 0.97^*$ | $6.64 \pm 3.21^*$ | $-0.12 \pm 1.11^*$ | $-0.03 \pm 0.43$ |
| | DiffPIR | $-0.01 \pm 0.23$ | $0.00 \pm 0.23$ | $0.04 \pm 1.12$ | $13.56 \pm 9.90^*$ | $-0.01 \pm 0.47$ | $0.00 \pm 0.31$ |
| | DPnP | $0.00 \pm 0.25$ | $-0.01 \pm 0.27^*$ | $-0.02 \pm 1.20$ | $-4.98 \pm 3.86^*$ | $0.06 \pm 0.77$ | $-0.02 \pm 0.34$ |
| Imputation | C-DPS | $0.00 \pm 0.30$ | $0.01 \pm 0.35$ | $0.41 \pm 1.51^*$ | $3.41 \pm 4.99^*$ | $-0.12 \pm 1.01^*$ | $-0.01 \pm 0.57$ |
| | DiffPIR | $0.00 \pm 0.29$ | $0.00 \pm 0.33$ | $0.03 \pm 1.05$ | $-0.20 \pm 3.05^*$ | $0.03 \pm 0.71$ | $0.00 \pm 0.47$ |
| | DPnP | $0.00 \pm 0.35$ | $-0.02 \pm 0.38$ | $-0.02 \pm 1.02$ | $-10.46 \pm 5.70^*$ | $0.02 \pm 0.67$ | $-0.01 \pm 0.48$ |
| Fourier | C-DPS | $-0.02 \pm 0.43$ | $-0.01 \pm 0.49$ | $0.80 \pm 1.43^*$ | $0.09 \pm 5.63^*$ | $-0.03 \pm 0.79^*$ | $0.01 \pm 0.49$ |
| | DiffPIR | $-0.01 \pm 0.39$ | $0.00 \pm 0.40$ | $0.12 \pm 0.83^*$ | $-0.64 \pm 1.70^*$ | $-0.03 \pm 0.42^*$ | $-0.02 \pm 0.38$ |
| | DPnP | $-0.01 \pm 0.43$ | $0.00 \pm 0.45$ | $-0.33 \pm 1.13^*$ | $-1.32 \pm 3.18^*$ | $0.00 \pm 0.54$ | $0.01 \pm 0.46$ |

Table 5: Runtime of the benchmark with learned objects.

| | | Gauss(0.25) | Laplace(1) | BL(0.1, 1) | St(1) | St(2) | St(3) |
|---|---|---|---|---|---|---|---|
| Denoising | C-DPS | 00:04:52 | 00:04:52 | 00:02:56 | 00:04:52 | 00:04:52 | 00:04:52 |
| | DiffPIR | 00:01:59 | 00:01:58 | 00:01:12 | 00:01:58 | 00:01:59 | 00:01:59 |
| | DPnP | 00:02:33 | 00:04:58 | 00:01:15 | 00:59:33 | 00:06:13 | 00:04:58 |
| Deconvolution | C-DPS | 00:04:52 | 00:04:53 | 00:02:57 | 00:04:53 | 00:04:53 | 00:04:52 |
| | DiffPIR | 00:01:59 | 00:01:59 | 00:01:12 | 00:01:59 | 00:01:59 | 00:01:59 |
| | DPnP | 00:13:54 | 00:46:39 | 00:05:48 | 00:53:30 | 00:28:24 | 00:28:24 |
| Imputation | C-DPS | 00:04:53 | 00:04:53 | 00:02:59 | 00:04:53 | 00:04:53 | 00:04:53 |
| | DiffPIR | 00:01:59 | 00:01:59 | 00:01:13 | 00:01:59 | 00:01:59 | 00:01:59 |
| | DPnP | 00:04:58 | 00:16:18 | 00:18:56 | 00:51:41 | 00:39:04 | 00:32:50 |
| Fourier | C-DPS | 00:04:54 | 00:04:54 | 00:02:59 | 00:04:55 | 00:04:55 | 00:04:54 |
| | DiffPIR | 00:01:59 | 00:01:59 | 00:01:13 | 00:01:59 | 00:01:59 | 00:01:59 |
| | DPnP | 00:06:13 | 00:13:53 | 00:04:42 | 00:51:41 | 00:23:39 | 00:16:18 |

## G    ADDITIONAL RESULTS

We provide in Table 4 an exhaustive quantitative evaluation of the change in the optimality gap after we substitute the arbitrary-precision Monte Carlo denoiser for the learned denoiser. We also report for which cases the arbitrary-precision denoiser enjoys significantly better results than the learned denoiser according to a Wilcoxon signed-rank test ($p = 0.05$, $N_{\text{test}}$ pairs, two-sided test with the winner determined by the median of differences). We attribute a better performance of the learned denoiser to the fact that the algorithms are fine-tuned using the learned component or to the cases where the likelihood score approximation is compensated by the one of the learned component. Note that this table must be interpreted with the quality of the denoisers in mind. As we show in Figure 9, for our particular choice of $S = 300$ samples, the Monte Carlo denoiser is strictly better than the learned denoiser over all noise variances only for signals with $\mathrm{BL}(0.1, 1)$ and $\mathrm{St}(1)$ increment distributions.

We show uncurated qualitative results of the MMSE estimate obtained by the DPS algorithms and the gold-standard Gibbs methods in Figures 11 to 18. The figures alternate between the arbitrary-precision denoiser and the learned denoiser and show the results for deconvolution, denoising, imputation, and reconstruction from partial Fourier samples, in that order. Each figure contains results for $\mathrm{BL}(0.1, 1)$, $\mathrm{St}(1)$, $\mathrm{St}(2)$, and $\mathrm{Laplace}(1)$ increment distributions.

The coverage results for $\alpha = 0.9$ are presented in Table 7. The Gibbs row again validates the implementation; for all forward operators, they achieve coverages that are very close to $0.9$. In contrast, the coverage values obtained by the DPS algorithms are generally much smaller than $0.9$. For C-DPS and DiffPIR, the reported coverage values are almost always $0$ except for $\mathrm{BL}(0.1, 1)$ and $\mathrm{St}(1)$ increments, where the coverages are usually (close to) $1$ for C-DPS and inconsistent for DiffPIR. For almost all increment distributions and forward operators, DPnP reports coverage values that are closest to, but typically smaller than, $0.9$. Note that a coverage of $1$ can be considered the worst case even at a target of $0.9$. For instance, it would be achieved by setting all samples to a constant vector with extremely large (*i.e.*, "unlikely") entries.

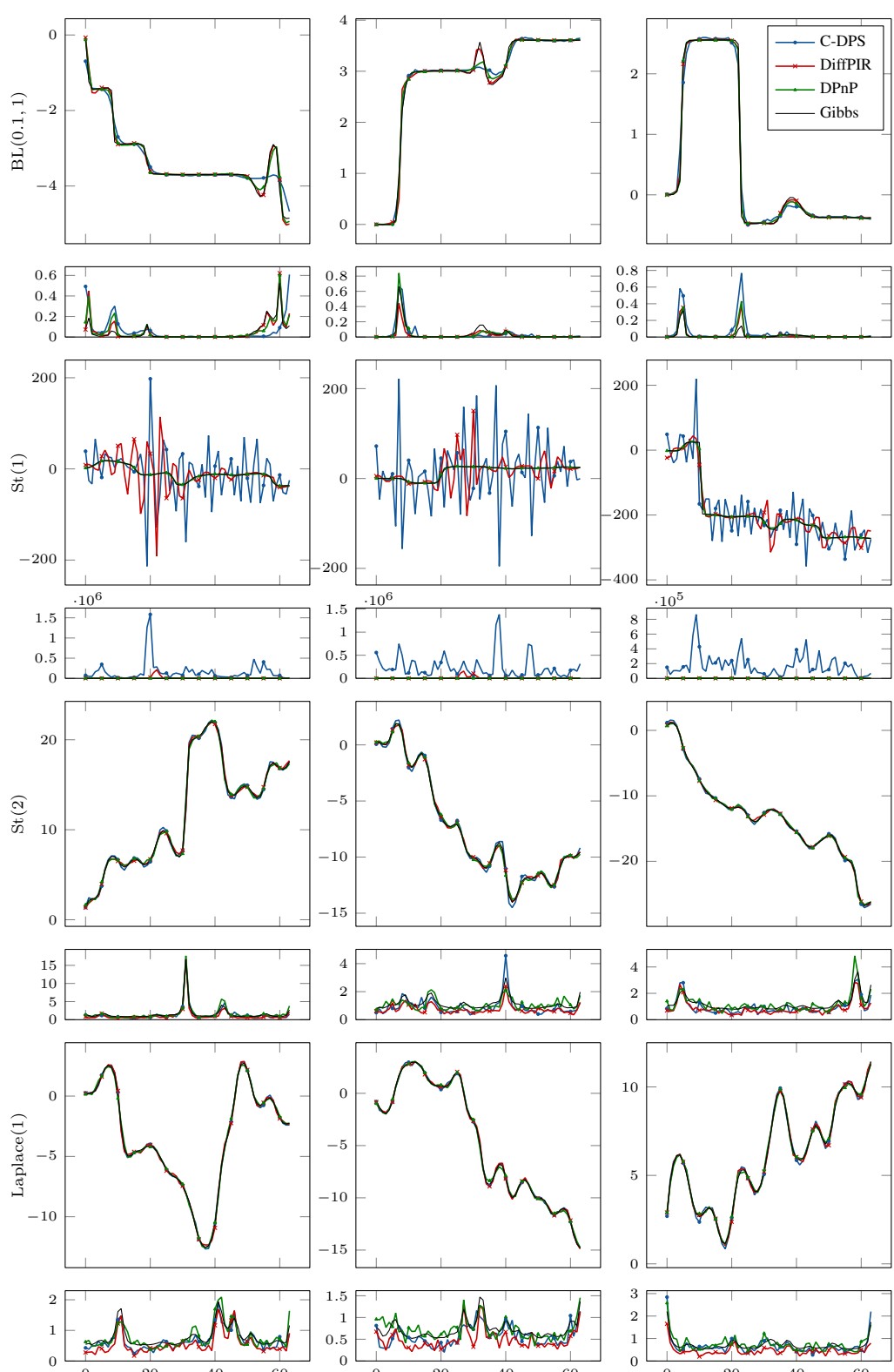

Figure 11: Qualitative results for deconvolution using the Monte Carlo. Rows: increment distributions. For each increment distribution, the MMSE estimates obtained by the different DPS algorithms and the gold-standard Gibbs methods are shown on top of the corresponding index-wise marginal variances. Columns: Different measurements.

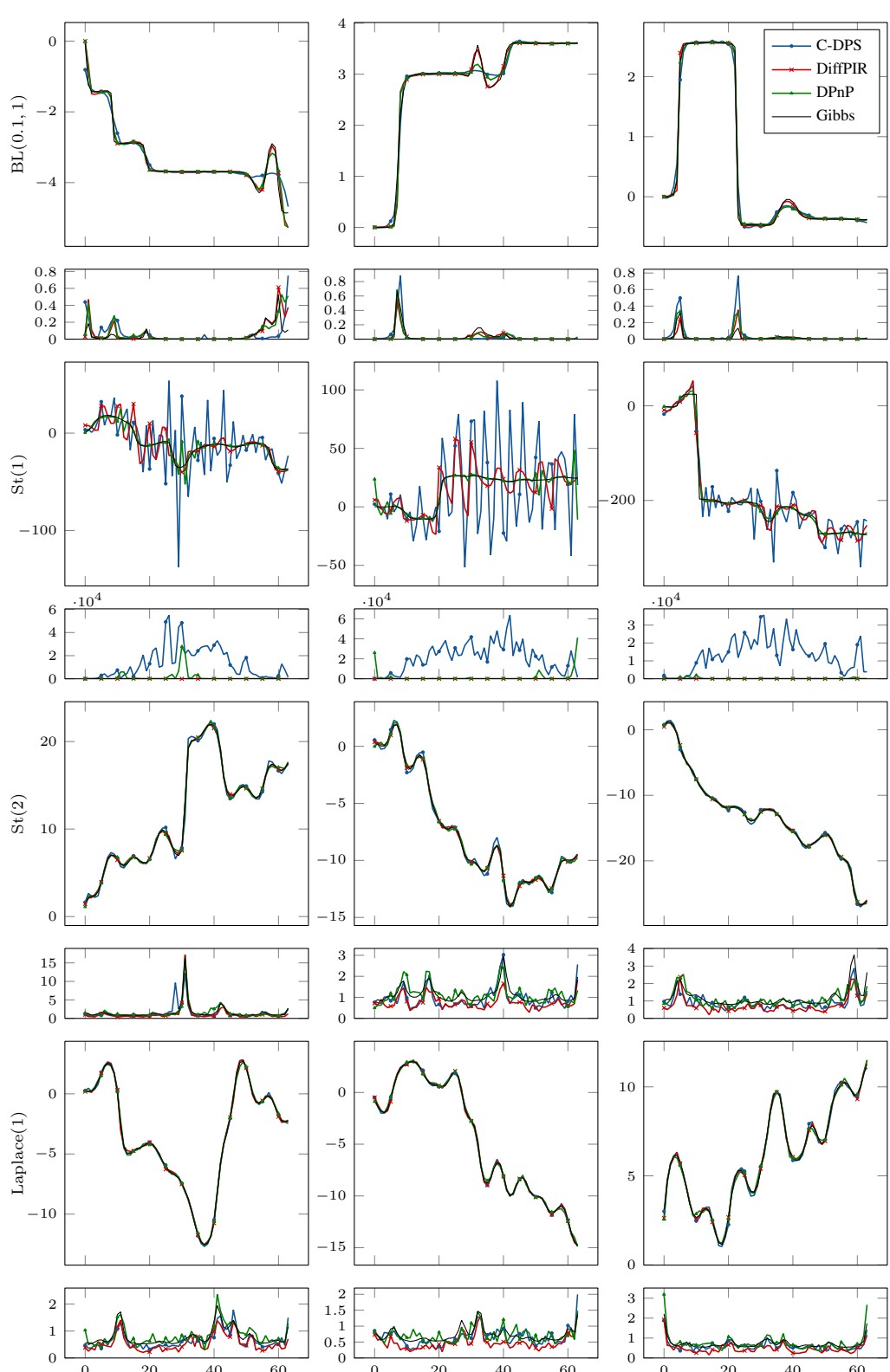

Figure 12: Qualitative results for deconvolution using the learned denoiser. Rows: increment distributions. For each increment distribution, the MMSE estimates obtained by the different DPS algorithms and the gold-standard Gibbs methods are shown on top of the corresponding index-wise marginal variances. Columns: Different measurements.

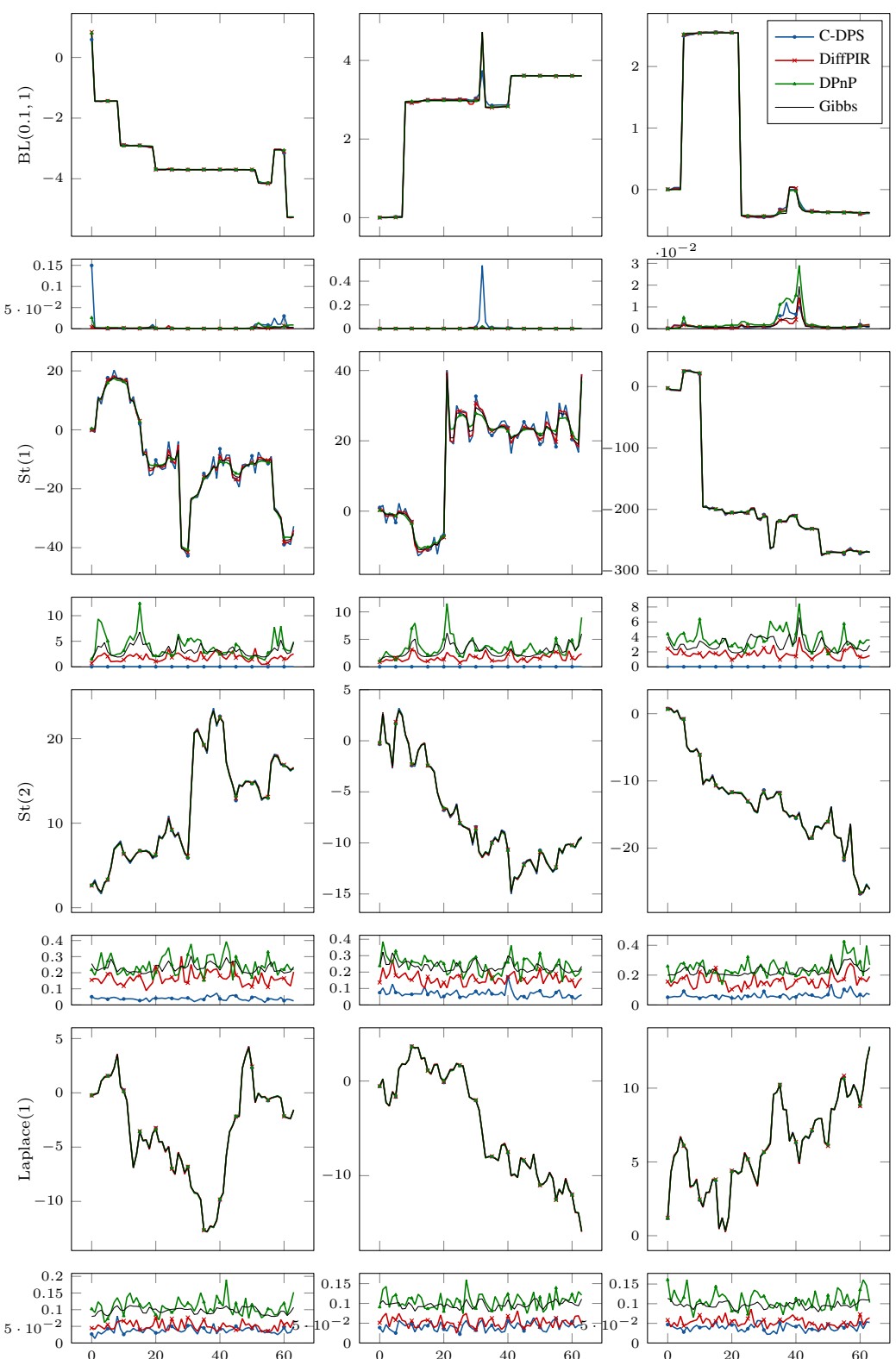

Figure 13: Qualitative results for denoising using the Monte Carlo denoiser. Rows: increment distributions. For each increment distribution, the MMSE estimates obtained by the different DPS algorithms and the gold-standard Gibbs methods are shown on top of the corresponding index-wise marginal variances. Columns: Different measurements.

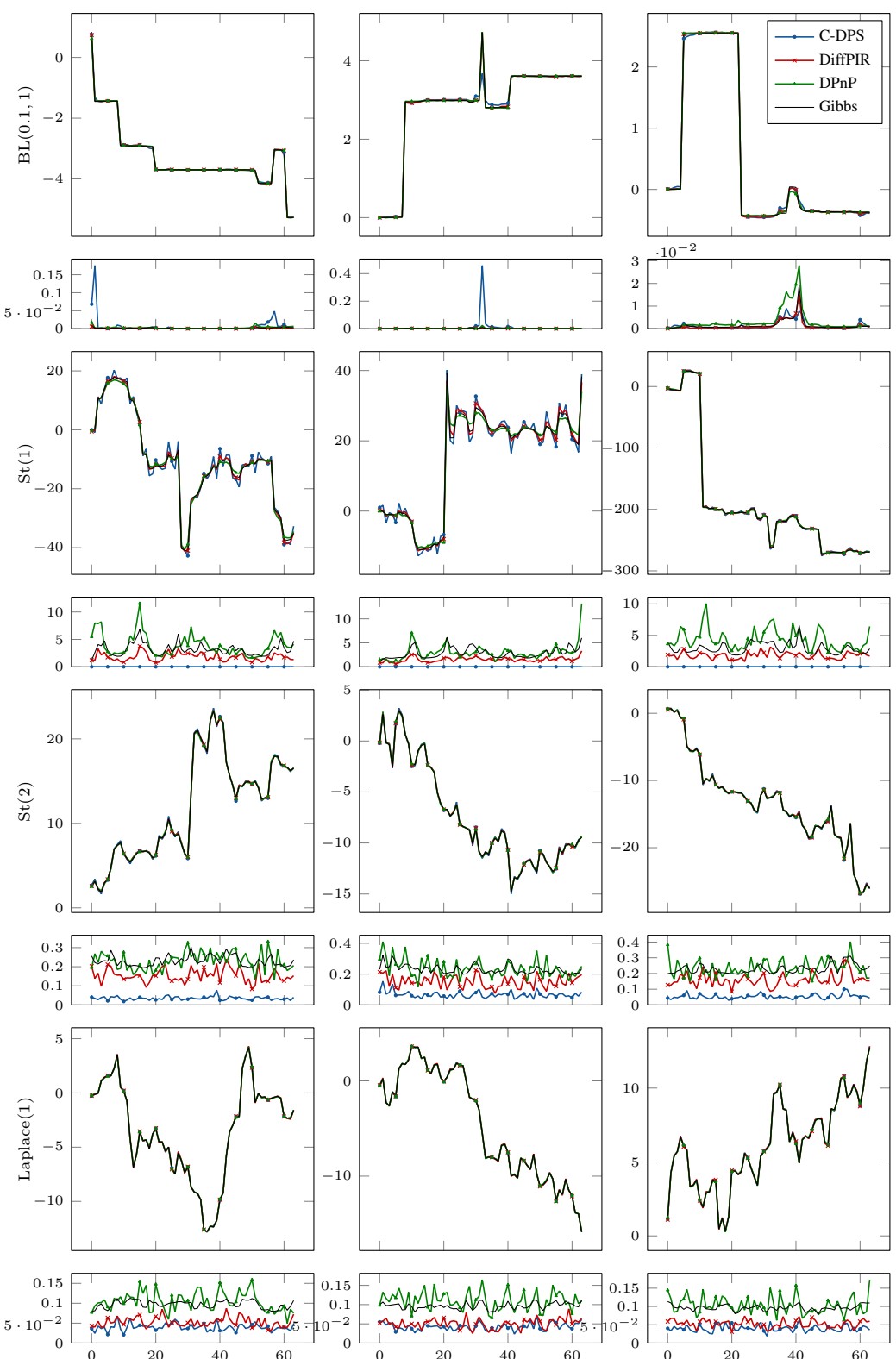

Figure 14: Qualitative results for denoising using the learned denoiser. Rows: increment distributions. For each increment distribution, the MMSE estimates obtained by the different DPS algorithms and the gold-standard Gibbs methods are shown on top of the corresponding index-wise marginal variances. Columns: Different measurements.

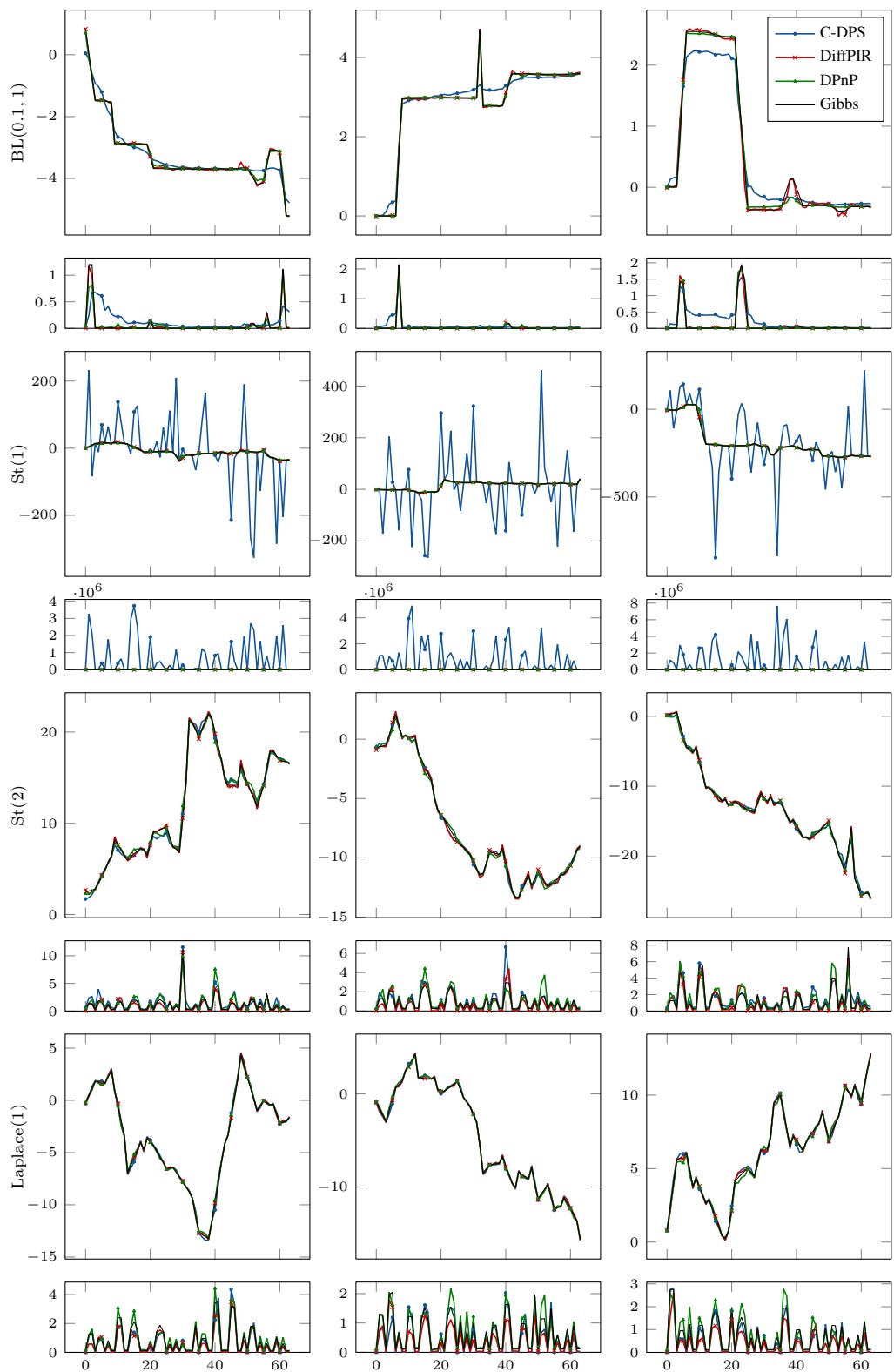

Figure 15: Qualitative results for imputation using the Monte Carlo denoiser. Rows: increment distributions. For each increment distribution, the MMSE estimates obtained by the different DPS algorithms and the gold-standard Gibbs methods are shown on top of the corresponding index-wise marginal variances. Columns: Different measurements.

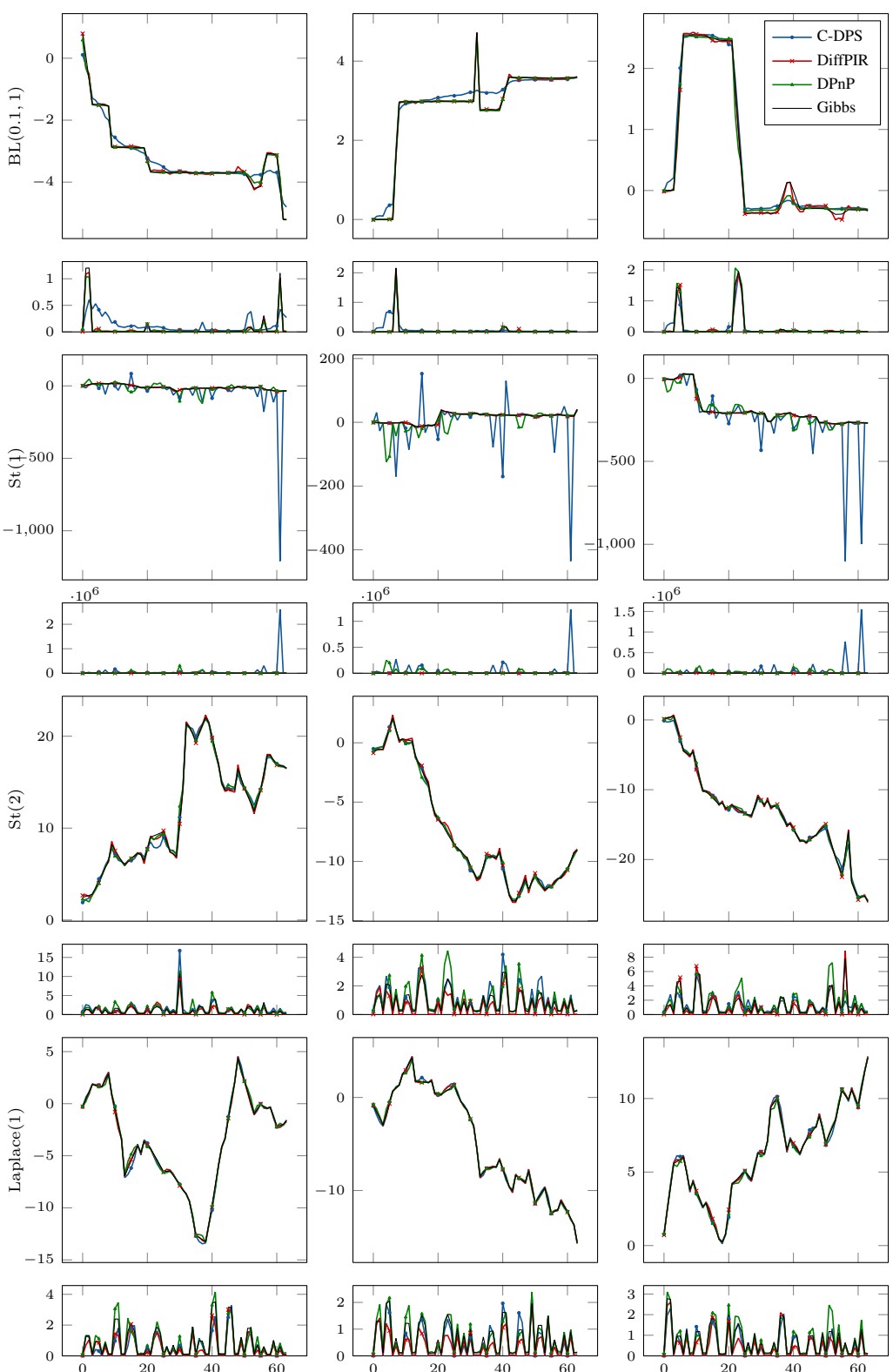

Figure 16: Qualitative results for imputation using the learned denoiser. Rows: increment distributions. For each increment distribution, the MMSE estimates obtained by the different DPS algorithms and the gold-standard Gibbs methods are shown on top of the corresponding index-wise marginal variances. Columns: Different measurements.

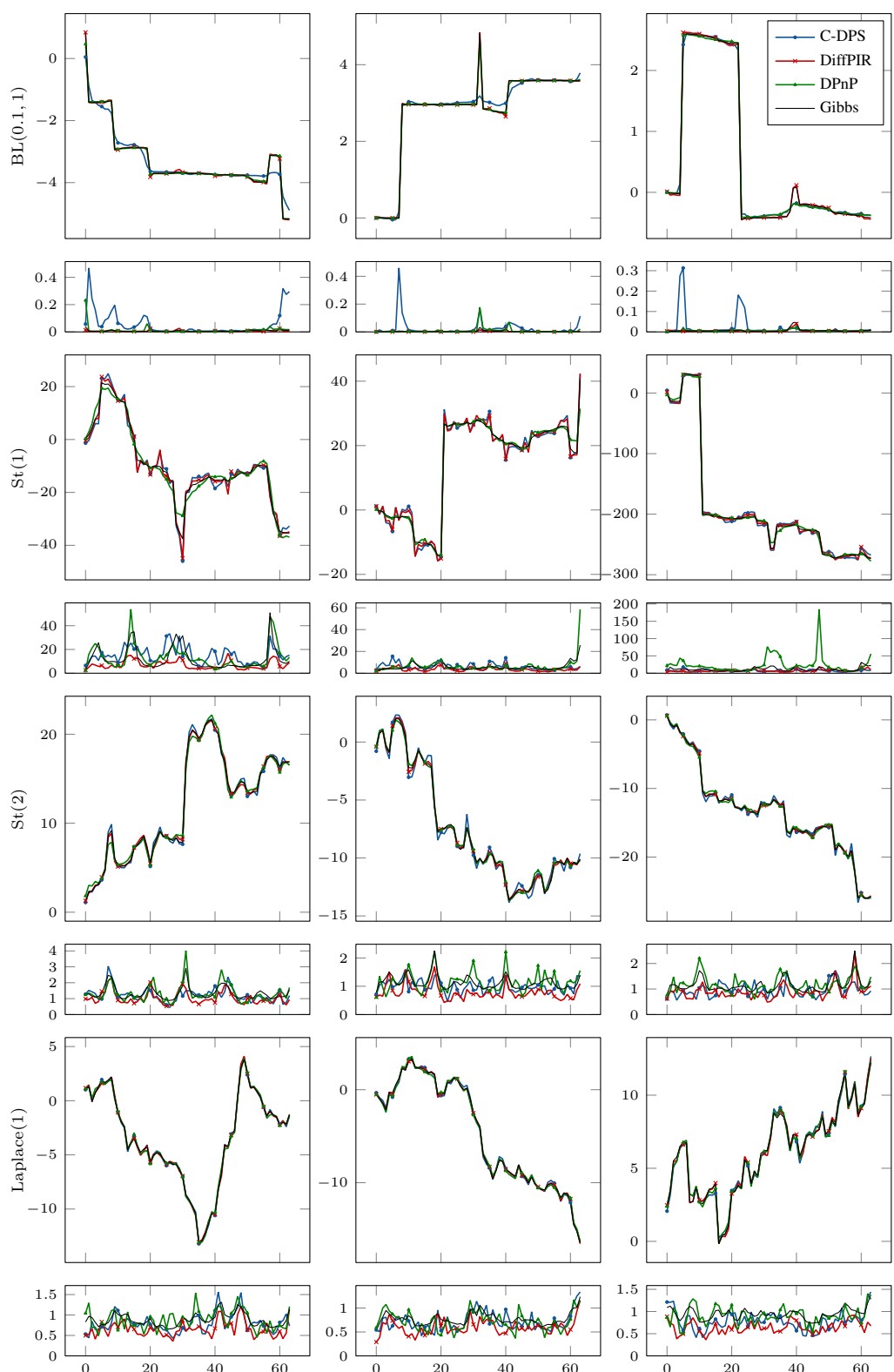

Figure 17: Qualitative results for reconstruction from partial Fourier measurements using the Monte Carlo denoiser. Rows: increment distributions. For each increment distribution, the MMSE estimates obtained by the different DPS algorithms and the gold-standard Gibbs methods are shown on top of the corresponding index-wise marginal variances. Columns: Different measurements.

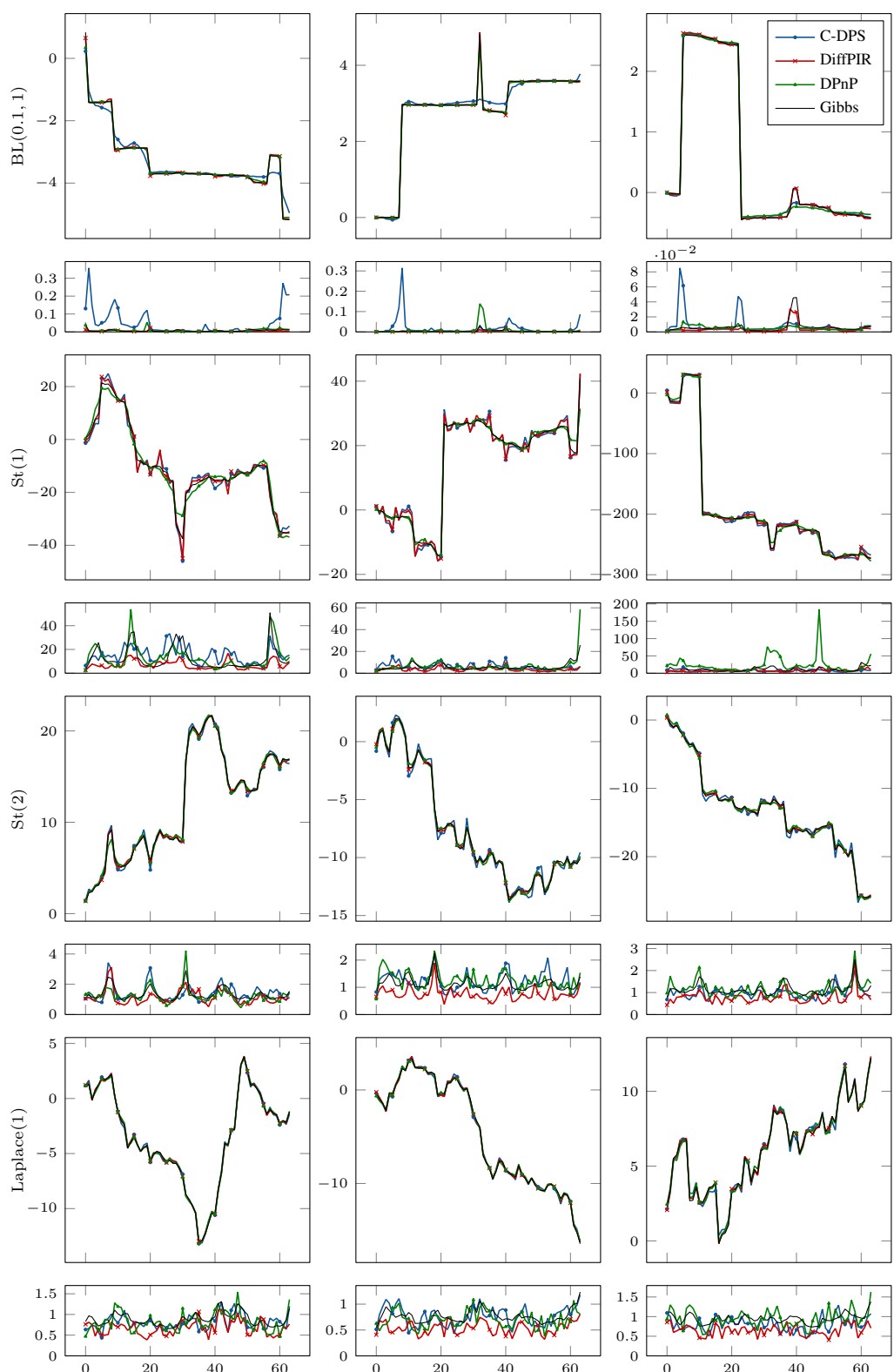

Figure 18: Qualitative results for reconstruction from partial Fourier measurements using the learned denoiser. Rows: increment distributions. For each increment distribution, the MMSE estimates obtained by the different DPS algorithms and the gold-standard Gibbs methods are shown on top of the corresponding index-wise marginal variances. Columns: Different measurements.

Table 6: Runtime of the benchmark with Monte Carlo objects.

|  |  | Gauss(0.25) | Laplace(1) | BL(0.1,1) | St(1) | St(2) | St(3) |
|---|---|---|---|---|---|---|---|
| Denoising | C-DPS | 05:52:28 | 07:23:23 | 34:07:44 | 05:52:40 | 05:34:51 | 05:31:10 |
|  | DiffPIR | 05:04:40 | 06:36:25 | 33:46:29 | 05:12:30 | 05:39:06 | 05:38:24 |
|  | DPnP | 00:03:04 | 00:03:57 | 00:20:36 | 00:03:23 | 00:03:08 | 00:03:10 |
| Deconvolution | C-DPS | 05:53:40 | 07:25:17 | 34:17:12 | 05:28:38 | 05:24:12 | 05:24:00 |
|  | DiffPIR | 05:28:09 | 06:55:34 | 34:16:17 | 05:31:29 | 05:32:32 | 05:22:39 |
|  | DPnP | 00:03:05 | 00:03:59 | 00:21:01 | 00:03:13 | 00:03:21 | 00:03:21 |
| Imputation | C-DPS | 05:49:07 | 07:15:41 | 34:29:37 | 05:53:44 | 05:27:44 | 05:26:05 |
|  | DiffPIR | 05:50:15 | 07:00:13 | 33:52:26 | 05:34:00 | 05:24:16 | 05:09:56 |
|  | DPnP | 00:03:23 | 00:04:18 | 00:20:58 | 00:03:09 | 00:03:05 | 00:03:22 |
| Fourier | C-DPS | 05:49:49 | 07:09:51 | 34:30:13 | 05:49:44 | 05:49:26 | 05:49:07 |
|  | DiffPIR | 05:13:06 | 06:38:32 | 34:31:38 | 05:17:58 | 06:14:52 | 05:15:14 |
|  | DPnP | 00:03:04 | 00:04:12 | 00:20:59 | 00:03:05 | 00:03:19 | 00:03:32 |

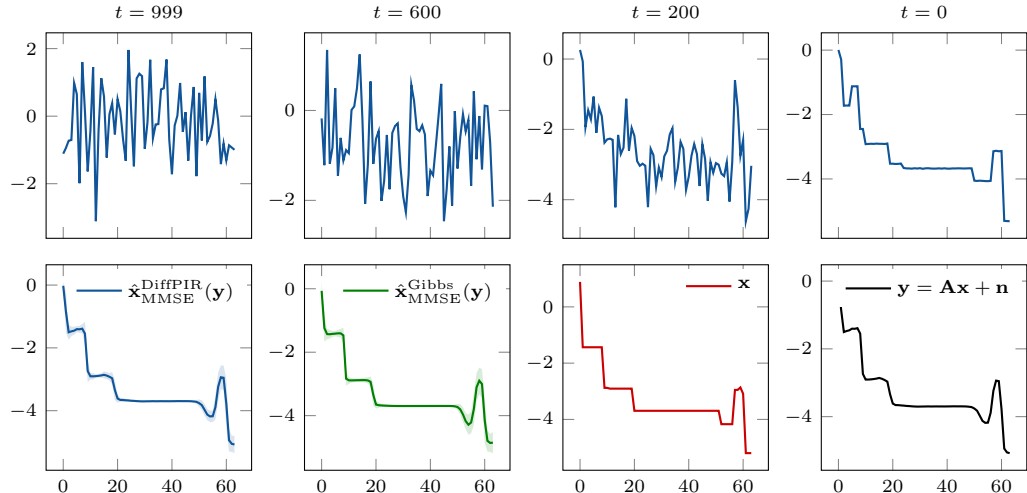

Figure 19: Conditional generation for deconvolution of a signal with $\mathrm{BL}(0.1,1)$ increments with DiffPIR. Top: Prototypical sampling trajectory at times $t = 999, 600, 200, 0$. Bottom: From left to right: MMSE estimate obtained by averaging all DiffPIR samples; gold-standard MMSE estimate obtained by the Gibbs method; the data-generating signal; the data.

Table 7: Posterior coverage of various estimation methods at $\alpha = 0.9$. MC: Monte Carlo.

|  |  | Gauss(0, 0.25) | | Laplace(1) | | BL(0.1,1) | | St(1) | | St(2) | | St(3) | |
|---|---|---|---|---|---|---|---|---|---|---|---|---|---|
|  |  | Learned | MC | Learned | MC | Learned | MC | Learned | MC | Learned | MC | Learned | MC |
| Denoising | Gibbs | — | 0.90 | — | 0.91 | — | 0.91 | — | 0.89 | — | 0.91 | — | 0.89 |
|  | C-DPS | 0.00 | 0.00 | 0.00 | 0.00 | 1.00 | 1.00 | 1.00 | 1.00 | 0.00 | 0.00 | 0.00 | 0.00 |
|  | DiffPIR | 0.00 | 0.00 | 0.00 | 0.00 | 1.00 | 1.00 | 0.28 | 0.02 | 0.00 | 0.00 | 0.00 | 0.00 |
|  | DPnP | 0.58 | 0.67 | 0.11 | 0.11 | 1.00 | 0.41 | 0.53 | 0.08 | 0.09 | 0.09 | 0.09 | 0.10 |
| Deconvolution | Gibbs | — | 0.89 | — | 0.90 | — | 0.90 | — | 0.91 | — | 0.91 | — | 0.91 |
|  | C-DPS | 0.00 | 0.00 | 0.01 | 0.00 | 1.00 | 1.00 | 1.00 | 0.83 | 0.01 | 0.00 | 0.00 | 0.00 |
|  | DiffPIR | 0.00 | 0.00 | 0.00 | 0.00 | 1.00 | 1.00 | 0.97 | 0.92 | 0.00 | 0.00 | 0.00 | 0.00 |
|  | DPnP | 0.12 | 0.12 | 0.06 | 0.07 | 1.00 | 0.31 | 0.50 | 0.06 | 0.06 | 0.06 | 0.07 | 0.06 |
| Imputation | Gibbs | — | 0.89 | — | 0.90 | — | 0.86 | — | 0.91 | — | 0.91 | — | 0.91 |
|  | C-DPS | 0.00 | 0.00 | 0.00 | 0.00 | 1.00 | 1.00 | 0.94 | 0.78 | 0.15 | 0.15 | 0.00 | 0.00 |
|  | DiffPIR | 0.00 | 0.00 | 0.00 | 0.00 | 1.00 | 1.00 | 0.72 | 0.32 | 0.00 | 0.00 | 0.00 | 0.00 |
|  | DPnP | 0.28 | 0.31 | 0.09 | 0.08 | 1.00 | 0.41 | 0.56 | 0.07 | 0.14 | 0.13 | 0.12 | 0.13 |
| Fourier | Gibbs | — | 0.91 | — | 0.90 | — | 0.90 | — | 0.91 | — | 0.92 | — | 0.91 |
|  | C-DPS | 0.00 | 0.00 | 0.00 | 0.00 | 1.00 | 1.00 | 0.96 | 0.74 | 0.01 | 0.01 | 0.00 | 0.00 |
|  | DiffPIR | 0.00 | 0.00 | 0.00 | 0.00 | 1.00 | 1.00 | 0.92 | 0.65 | 0.00 | 0.01 | 0.00 | 0.00 |
|  | DPnP | 0.19 | 0.19 | 0.08 | 0.06 | 1.00 | 0.32 | 0.50 | 0.06 | 0.07 | 0.07 | 0.07 | 0.06 |

