# OpenReview forum: "A Statistical Benchmark for Diffusion-Posterior-Sampling Algorithms"
_ICLR.cc/2026/Conference — ICLR 2026 Poster_

### Official Review · Reviewer_pM9c · 2025-10-15

**Soundness:** 3
**Presentation:** 2
**Contribution:** 3
**Rating:** 6
**Confidence:** 4

**Summary:**

The authors propose a new benchmark for evaluating different posterior sampling algorithms using diffusion models (dubbed DPS in this paper; Different from DPS [1]), where the posterior samples can be computed analytically, so that the ground truth is given.
Previous *benchmarks* that admit analytical posterior samples were constrained to settings where the prior is a mixture of Gaussians, which largely differs from the natural data statistics. The prior distributions considered in this paper is much larger, and the authors propose methods to efficiently compute ground truth posterior distributions. Several widely established baselines are compared.

**References**

[1] Chung et al., "Diffusion posterior sampling for general noisy inverse problems", ICLR 2023

**Strengths:**

1. To the best of my knowledge, this is the first approach to go beyond mixture of gaussian priors when attempting to build a ground truth posterior distribution.

2. The paper is well-written and easy to follow, with sufficient background given in the appendix.

3. The method of acquiring the posterior distribution by extending Kuric et al. [1] is sound.


**References**

[1] Kuric et al., "The Gaussian latent machine: Efficient prior and posterior sampling for inverse problems", arxiv 2025

**Weaknesses:**

1. Being able to use different prior/posterior distributions as ground truth is, in and of itself, important. Nevertheless, the argument would be strengthened if the paper shows that the proposed distributions in this paper are closer to real-world statistics in some cases. Currently, only some references are given.

2. The authors mention that the proposed framework can be extended to higher-dimensional settings, but there are complications. It would add much value if the authors were to include experiments with $d$ that match the typical image resolutions. Currently, it seems like the experiments are conducted with low dimensionality ($d$). What's the value of $d$ chosen here?

**Questions:**

Is there any reason to constrain the benchmark for *diffusion* posterior sampling algorithms?

---

> ### Author Response · Authors · 2025-11-18
>
> Thank you for affirming that going beyond Gaussian is important and that the paper is well written and sound.
>
> ---
>
> **Weaknesses**
> > Being able to use different prior/posterior distributions as ground truth is, in and of itself, important. Nevertheless, the argument would be strengthened if the paper shows that the proposed distributions in this paper are closer to real-world statistics in some cases. Currently, only some references are given.
>
> Thank you for this suggestion. *We will add concrete examples of instances where real-world statistics are well-approximated by the distributions that we consider in the paper.*
>
> > The authors mention that the proposed framework can be extended to higher-dimensional settings, but there are complications. It would add much value if the authors were to include experiments with $d$ that match the typical image resolutions. Currently, it seems like the experiments are conducted with low dimensionality ($d$). What's the value of $d$ chosen here?
>
> The results presented in the paper are obtained with signals of length $d = 64$. *We will move this crucial piece of information from the appendix of the main body of the updated manuscript.* We believe that the relative ordering of the performance of the algorithms does not change with the dimensionality of the problems, and that the relative ordering of the algorithms on these moderate-dimensional problems is valuable in and of itself.
>
> The primary challenge when going to higher dimensions lies in the sampling of the Gaussian distributions that arise in the conditional sampling of $\mathbf{X} \mid \mathbf{Z} = \mathbf{z}$. As we discuss in Appendix D.2, sampling high-dimensional Gaussians is a well-studied problem and advances in that field can directly be used in our framework. In particular, the Gaussians that arise in our sampling algorithms can be efficiently samples with perturb-and-MAP approaches with matrix-free conjugate gradient implementations. Indeed, the implementation of the benchmarking framework already contains such routines. *We will expand this discussion and add the runtimes of the methods depending on the dimensionality of the signal.*
>
> **Questions**
> >Is there any reason to constrain the benchmark for diffusion posterior sampling algorithms?
>
> Thank you for this important question. As we briefly mention in lines 168 and 169, the reference samples that we obtain with the Gibbs methods can be compared to any samples obtained by any other method with arbitrary distribution-level comparisons such as (sliced) Wasserstein distances. In the present paper, we focus on diffusion models specifically because the Gibbs methods can be used to sample the denoising posteriors that arise there, and we can thereby isolate algorithmic error from any suboptimal learned components. We believe that our approach of isolating systematic error can be extended to other approaches, such as plug-and-play solvers with flow-matching priors and *will mention this as future work in the updated manuscript.*

---

> > ### Comment · Reviewer_pM9c · 2025-11-26
> >
> > I thank the authors for the addressing my comments. I will maintain my positive score.

---

### Official Review · Reviewer_TmEt · 2025-10-31

**Soundness:** 3
**Presentation:** 3
**Contribution:** 3
**Rating:** 4
**Confidence:** 3

**Summary:**

This paper introduces a statistical benchmark for evaluating diffusion posterior sampling algorithms using discretized Lévy processes with tractable Gibbs posteriors as ground truth. While the framework enables rigorous distribution-level comparisons, the evaluation is severely limited to low-dimensional (d=64) linear inverse problems, raising serious concerns about scalability and practical relevance to realistic imaging applications.

**Strengths:**

Developing a benchmark for posterior sampling in high-dimensional problems is important.

**Weaknesses:**

• All experiments use d=64 signals with only linear operators. No evidence is provided that the framework scales to realistic dimensions (e.g., 256×256 images) or nonlinear problems, fundamentally limiting the practical applicability and making it unclear whether findings transfer to problems researchers actually solve.

• The authors cite power-law phenomena in finance and images to motivate heavy-tailed priors, but never demonstrate that their 1D discretized Lévy processes meaningfully capture structure in realistic signals. The connection to actual image statistics remains unsubstantiated.

• Table 4 shows learned denoisers often match or exceed oracle performance, undermining claims about isolating likelihood approximation errors. The paper doesn't establish whether likelihood errors dominate versus other sources (discretization, hyperparameter sensitivity), weakening the diagnostic utility argument.

• DPS algorithms are tuned with learned denoisers but evaluated with oracle denoisers using the same hyperparameters (lines 276-278). This mismatch means oracle results may be suboptimal, contradicting claims about properly isolating algorithmic errors.

• Claims of "efficient implementations" and "acceptable runtimes" (lines 231-234, 822-823) lack any quantitative evidence; no runtime comparisons, memory usage, or scalability analysis is provided to substantiate efficiency claims or assess practical feasibility at higher dimensions.

**Questions:**

Does this benchmark can be used for amortized diffusion sampling methods, i.e., learning the full posterior?

---

> ### Author Response · Authors · 2025-11-18
>
> Thank you for affirming that our work is important and that the soundness, the presentation, and the contribution is good.
>
> ---
>
> **Weaknesses**
>
> > All experiments use d=64 signals with only linear operators...
>
> We kindly refer to the comment that we added to the root of the discussion for a more general discussion about the relevance of the setting.
>
> **The relevance of the setting:** Linear measurement models are ubiquitous across the natural sciences, for instance imaging (MRI, CT, PET, SPECT, microscopy, conventional cameras, photoaccoustic tomography, seismic tomography, and many more), audio, spectroscopy, finance, and classical machine learning. 64 dimensional signals are considered reasonably large in classical statistics or blocked problems (for instance, JPEG blocks consist of exactly 64 pixels). Thus, there exist many fields in which our setting is of interest. In any case, many diffusion posterior sampling algorithms are specifically designed for linear inverse problems, which is the setting that we consider. As we discuss in the comment that we added to the root of the discussion, failures in this moderate-dimensional regime are highly indicative of failures in more complex, higher-dimensional problems.
>
> **The extension to nonlinear problems:** Since our framework fundamentally relies on the existence of (relatively) efficient sampling routines for the generation of the reference samples from the posteriors, we only consider linear measurements for which such routines exist. Sampling of the posteriors that arise from nonlinear measurement models and possibly non-Gaussian noise is highly challenging, and only few algorithms that are specialized for certain setups (for example, the one in [1]) have been proposed. We believe that a comprehensive overview of the field of sampling the posteriors that arise from various nonliear measurement models is out of the scope of the present paper. Nevertheless, *we will add a discussion about possible extensions to nonlinear problems to the updated manuscript.*
>
> **Scaling to higher-dimensional signals:** The primary challenge when going to higher dimensions lies in the sampling of the Gaussian distributions that arise in the conditional sampling of $\mathbf{X} \mid \mathbf{Z} = \mathbf{z}$. As we discuss in Appendix D.2, sampling high-dimensional Gaussians is a well-studied problem and advances in that field can directly be used in our framework. In particular, the Gaussians that arise in our sampling algorithms can be efficiently sampled with perturb-and-MAP approaches with matrix-free conjugate gradient implementations. Indeed, our publicly-available implementation of the benchmarking framework already contains such routines. *We will add a description of the computational complexity of our framework to the updated manuscript. In addition, we will show how our framework scales to higher-dimensional signals by adding experiments that showcase the runtimes of the methods in terms of the dimensionality of the signal.* In any case, we believe that the relative ordering of the performance of the algorithms does not change with the dimensionality of the problems, and that the relative ordering of the algorithms on these moderate-dimensional problems is valuable in and of itself.
>
> [1] Wang, Bardsley, et al. "Bayesian Inverse Problems with l1 Priors: A Randomize-Then-Optimize Approach", SIAM Journal on Scientific Computing, Vol. 39, Iss. 5 (2017).
>
> > The authors cite power-law phenomena ...
>
> Lévy processes are used extensively in finance. For example, references Cont, 2001 (Figures 1 and 2) and Schoutens, 2003 (Figure 4.1 and 4.3) contain real examples of financial signals that are well-modeled with Lévy processes. Chapter 6 in Schoutens, 2003 is devoted to modeling stock prizes with Lévy processes. The relevant reference for images, Wainwright & Simoncelli (1999) shows that the histograms of responses in a particular wavelet sub-band are heavy-tailed in Figure 1. The histogram of the difference of horizontally- or vertically-adjacent pixels (i.e., the "increment distribution") in natural images is shown in [2], and the distribution is well-modeled by those that we consider in the paper. In other words, rows or columns in natural images are well-modeled by Lévy processes with heavy-tailed increment distributions.
>
> In any case, we are not claiming to model any particular signal with the Lévy processes; we are proposing a benchmark based on those. This is a first step towards a rigorous evaluation, but not necessarily the endpoint: If a particular method does not perform well on Lévy processes, it will likely not perform well on more complicated distributions either.
>
> In accordance with pM9c's comment, we *will add concrete examples of instances where real-world signals are well-approximated by Lévy processes.* Does this addition address this remark? We would appreciate any clarification.
>
> [2] Huang and Mumford, "Statistics of Natural Images and Models", ICCV (1999)

---

> ### Author Response · Authors · 2025-11-18
>
> **Weaknesses (continued)**
>
> > Table 4 shows learned denoisers often match or exceed oracle performance ...
>
> We agree that the current evaluation does not isolate the error due to the likelihood approximations. The current evaluation benchmarks the realistic scenario where learned components are used and the hyperparameters of the methods are tuned to those components. It also currently benchmarks the robustness to swapping in a higher-quality denoiser. This is valuable in and of itself. We resorted to this due to computational limitations, but nothing fundamentally restricts the evaluation to that. Our framework also provides a means to quantify the influence of the errors introduced by some fixed discretization by substituting one for the other and utilizing the reference Gibbs objects. This could ease the development and evaluation of alternative discretizations in future works.
>
> *We will make it explicit in the updated manuscript that the current evaluation targets the realistic setting and quantifies the hyperparameter sensitivity. In addition, we will tune all algorithms on a selected subset of all problems (one forward operator and one increment distribution) to showcase that possibility of isolating algorithmic errors.*
>
> > DPS algorithms are tuned with learned denoisers but evaluated with oracle denoisers ...
>
> We fully agree. This experiment shows to which extent the hyperparameters of the methods are robust to the quality of the denoiser. We can see that DPnP benefits from better denoisers, which is valuable in and of itself. *We will make this clearer in the updated manuscript.*
>
> > Claims of "efficient implementations" and "acceptable runtimes" ...
>
> Thank you for this remark. We agree that runtimes and the quantitative evidence regarding the achieved speedup were entirely missing and believe that providing that information will foster the adoption of the algorithm in the community. *The updated manuscript will contain the crucial information of the runtimes of the experiments that are currently in the paper (per forward operator/increment distribution/denoiser). In addition, we will quantify the achieved speedup of each of the improvements (paralellization, Sherman-Woodbury-Morrison, CUDA sampling routines, Triton kernels) over the naive implementation.*
>
> **Questions**
> > Does this benchmark can be used for amortized diffusion sampling methods, i.e., learning the full posterior?
>
> Any method for sampling the posteriors that arise in linear inverse problem can be benchmarked. However, the framework is of limited utility for methods that rely on learned components that can not be replaced by applying the Gibbs method to some appropriate distribution. *We will update the manuscript to make this clearer.*

---

### Official Review · Reviewer_bQ8j · 2025-11-01

**Soundness:** 4
**Presentation:** 3
**Contribution:** 4
**Rating:** 8
**Confidence:** 5

**Summary:**

Diffusion posterior sampling algorithms have become prominent methods for sampling from posterior diffusion with a denoising diffusion model prior. While many methods have been proposed in the recent years, most of the interesting benchmarks do not come with ground-truth posterior samples to which one can compare against. The aim of this paper is to close this gap by proposing a statistical benchmark that mimicks the behaviour of realistic data (power-law-like extremes as stated in the paper). To this end the authors consider the posterior associated to Lévy processes and use an efficient Gibbs sampler to obtain gold-standard posterior samples that serve as reference.

**Strengths:**

- This paper tackles a fundamental problem in the evaluation of diffusion posterior samplers and proposes a very useful benchmark which in my opinion could be useful to the community and should be present in all the forthcoming papers.
- The model is general enough to contain different instantiations such as Laplace and spike and slab and thus goes beyond the existing gaussian mixture toy examples.
- The paper is rather well-written and quite pedagogical, I enjoyed reading it.

**Weaknesses:**

The only weakness I see is the structuring of the main paper. For example I think that some parts of the related works (such as the first two paragraphs) could be moved to the appendix as they are slightly relevant to the content of the paper. This space could be used to provide for example more background on the GLM framework, as one needs to go to the appendix to read more interesting details about it.
I also think that Figure 1 and 2 are misplaced as at this stage of the paper the Lévy process is not introduced and we don't know yet what St(1) means.

**Questions:**

I have a few suggestions and related works to be considered:
- I think it would have been interesting to include samples from a conditional diffusion model, by either training the conditional denoiser or estimating the denoiser using Monte Carlo samples as is done for DPS methods. I believe that it could be relevant since it provides a lower bound on the performance that one hopes to achieve with DPS methods.
- [1] considers an actual real world setting where gold standard samples can be obtained using MCMC.
- The toy Gaussian mixture benchmark is introduced in [2, 3]

[1] Cardoso, G.V. and Pereira, M., 2025. Predictive posterior sampling from non-stationnary Gaussian process priors via Diffusion models with application to climate data.
[2] Cardoso, G., Idrissi, Y.J.E., Corff, S.L. and Moulines, E., 2023. Monte Carlo guided diffusion for Bayesian linear inverse problems.
[3] Boys, B., Girolami, M., Pidstrigach, J., Reich, S., Mosca, A. and Akyildiz, O.D., 2023. Tweedie moment projected diffusions for inverse problems.

---

> ### Author Response · Authors · 2025-11-18
>
> Thank you for affirming that the paper tackles a fundamental problem in a useful way. We share your opinion that forthcoming papers could strengthen their positioning by testing with our framework. *We will also continuously update the github and the website with new results from the community.* Thank you for affirming that the soundness and the contribution are excellent, and that the presentation is good. We are glad that you enjoyed reading our paper.
>
> ---
>
> **Weaknesses**
> > The only weakness I see ...
>
> We fully agree that the structure of the paper could be improved. *We aim to use the additional page in the updated manuscript to provide more details about the GLM framework in the main text. In addition, we will improve the writing around Figures 1 and 2 by providing forward pointers to where the relevant concepts are described.*
>
> **Questions**
> > I think it would have been interesting to include samples from a conditional diffusion model, by either training the conditional denoiser or estimating the denoiser using Monte Carlo samples as is done for DPS methods. I believe that it could be relevant since it provides a lower bound on the performance that one hopes to achieve with DPS methods.
>
> We thank the reviewer for the comment. We agree that it is interesting to benchmark conditional denoisers to obtain some baseline, and, once obtained, *we will add such results to the github and the website.*
>
> > [1] considers an actual real world setting where gold standard samples can be obtained using MCMC.
>
> > The toy Gaussian mixture benchmark is introduced in [2, 3]
>
> Thank you for bringing those works to our attention. Indeed, they are relevant related work and *we will discuss them in the updated manuscript.*

---

### Official Review · Reviewer_tkeZ · 2025-11-02

**Soundness:** 2
**Presentation:** 3
**Contribution:** 3
**Rating:** 4
**Confidence:** 4

**Summary:**

- The authors introduce a benchmark suite for evaluating algorithms designed to solve linear inverse problems with diffusion model priors
- The benchmark is built on a synthetic setup derived from discretized Lévy processes
- It hence include setting of heavy-tailed/power-law–like distributions beyond the Gaussian case
- The key motivation lies in the fact that Lévy processes possess explicit marginal distributions and can be targeted using Gibbs sampling
- This property allows the benchmark to generate ground-truth posterior samples (from inverse problem and denoising posterior) for quantitative comparison across algorithms

**Strengths:**

- The paper is well-written and accompanied with concise explanations in the appendix
- The motivation of the paper is well articulated namely for principled benchmarking in diffusion-based inverse problem solvers
- The proposed benchmark is a valuable contribution, as it extends evaluation on Gaussian setup to a broader family of distributions

**Weaknesses:**

**Overstated or misleading claims**
The repeated use of the term "oracle", e.g., Lines 56, 129, 277, 355 is misleading.
The samples used in the benchmark are produced via Gibbs sampling—an iterative procedure—hence they are approximate, not exact. The quality of these samples depends on choices such as burn-in time, which are hyperparameters of the framework.
This issue becomes more apparent when the benchmark is applied to algorithms requiring gradients of the denoiser (Line 257-263 and equation (60)): the paper substitutes the latter with a covariance estimator of $X_0 | X_t,$ and hence further deviating from the notion of an "oracle".


**Template for posterior samplers**
The proposed benchmark template seems overly restrictive. By focusing on algorithms that use only the denoiser, it neglects methods that require the Jacobian of the denoiser.
Although the paper connects this to the covariance $Cov(X_0 \mid X_t)$, estimating this covariance is far more computationally demanding and less stable, and therefore it downgrade the claim that the benchmark offers "oracle" quantities with minimal approximation error.

**Evaluation design**
- The inclusion of learned denoisers in the evaluation is conceptually inconsistent with the paper’s stated goal of removing approximation errors (Section 1.1).
If the benchmark aims to isolate algorithmic performance, learned denoisers reintroduce training-dependent variability. While the authors justify this by citing robustness testing, the notion of robustness is loose and in practice requires hyperparameter tuning, which introduces additional confounding factors.
- The experimental comparison is limited. Only 3 algorithms are evaluated, and these do not represent the diversity of available approaches, e.g., optimization-based, variational, or midpoint-guided methods; see the literature in [1] and [2]

**Remarks and minor issues**

- In background, rephrase the statement in Line 132 about DDPM, sampling in fact depend on several parameters and it is bold to say " researchers typically use"; I would argue that frequently DDIM sampling is used with $\eta = 0$ (simulating the probability-flow ODE) for sharp samples with few diffusion steps
- The used abbreviation **DPS** is already/actually the name of a well-known algorithm in diffusion models and inverse problems [4], hence the abbreviation might be misleading using it here to refer to something else may cause confusion.
- The authors may also consider adding the following reference on inverse problems benchmarks [3]
- Line 288: The statement that DiffPIR is an extension of C-DPS is incorrect. DiffPIR follows a distinct formulation based on quadratic half-splitting with an auxiliary variable and does not rely on the VJP of the denoiser.


---

.. [1] Daras, Giannis, et al. "A survey on diffusion models for inverse problems." arXiv preprint arXiv:2410.00083 (2024).

.. [2] Oliviero-Durmus, Alain, et al. "Generative modelling meets Bayesian inference: a new paradigm for inverse problems." Philosophical Transactions A 383.2299 (2025): 20240334.

.. [3] Zheng, Hongkai, et al. "Inversebench: Benchmarking plug-and-play diffusion priors for inverse problems in physical sciences." arXiv preprint arXiv:2503.11043 (2025).

.. [4] Chung, Hyungjin, et al. "Diffusion posterior sampling for general noisy inverse problems." arXiv preprint arXiv:2209.14687 (2022).

**Questions:**

- I generally found the figures hard to understand and interpret, I'm referring namely to figure 1, it says that it shows reverse using the oracle denoiser, but it is not clear, similarly, for figure 3, it is hard to interpret namely to say wether the algorithms performs well or not
- can the authors provides hints/explanation on the derivation of equation (13)
- the authors claim that introduced framework can also assess the approach where the conditional components is learned (Line: 168-170), but it is not clear how it can be achieved given that in some tasks the likelihood is not known, e.g. tasks such deraining or dehazing, see for instance [1]; yet the benchmark is built on the ability to explicitly write the posteriors/marginals and target them using Gibbs sampling

- A more broad question: did the authors think about how the benchmark can be extend to nonlinear inverse problems ?

---

.. [1] Wang, Hanting, et al. "IRBridge: Solving Image Restoration Bridge with Pre-trained Generative Diffusion Models." arXiv preprint arXiv:2505.24406 (2025).

---

> ### Author Response · Authors · 2025-11-18
>
> Thank you for affirming that the paper is well-written, well-motivated, and that the benchmark is a valuable contribution.
>
> ---
>
> > Overstated or misleading claims
>
> Although Gibbs sampling is an iterative procedure, it is parameter- and bias-free (lines 300–303) and typically much faster than other general-purpose samplers (e.g., MALA) that could be used in this setup (lines 697–702): Sampling efficiencies close to 1 are reported in Table 4 in Kuric et al. (2025) in settings that are very close to ours. Bias-free sampling with an efficiency close to 1 is practically indistinguishable from direct sampling.
>
> Ultimately, regardless of the "speed" of the sampler, the relevant quantity is the accuracy of the Monte Carlo estimates of objects that are needed in the algorithms, such as the expectation or the covariance of the denoising posterior. The accuracy of these estimates is influenced by the burn-in period and the number of samples and we thoroughly describe the procedure that we use to obtain sensible values for them in Appendix F.4. In particular, we show that the Gibbs sampler converges rapidly (in a single-digit number of iterations) to the stationary distribution in Figure 5. The chosen burn-in period of 100 samples is, therefore, sufficiently large to ensure that the samples taken for the Monte Carlo estimation are from the stationary distribution (which is identical to the target distribution). The number of samples used in the Monte Carlo estimation is then chosen such that the estimate of the denoising-posterior expectation is significantly better (about one order of magnitude less in terms of MSE) than any learned denoiser (Figure 6). In any case, these numbers are tunable by the user and arbitrary precision can be achieved with a sufficiently large number of samples. In addition, some algorithms (e.g., DPnP, PnP-DM, and most of the CSGM-type algorithms) do not require Monte Carlo estimation of any object at all. They require *samples* from the denoising posterior that the Gibbs methods naturally provide.
>
> *We will replace all instances of "oracle" with "Gibbs" and clarify these points upon the introduction of that concept in the updated manuscript.*
>
> > Template for posterior samplers
>
> **On restrictiveness of the template:** Our template is not restricted to algorithms that only use the "denoiser" (the expectation of the denoising posterior). Fundamentally, it relies on the Monte Carlo estimation of various mathematical objects that are needed in the update steps of the different algorithms. These objects are accurately estimated from Gibbs samples of the denoising posterior. In particular, the Jacobian of the MMSE denoiser $\mathbf{x} \mapsto \mathbb{E}[\mathbf{X}_0 \mid \mathbf{X}_t = \mathbf{x}]$ at some point $\mathbf{x}_t$ is *identical* to the covariance of $\mathbf{X}_0 \mid \mathbf{X}_t = \mathbf{x}_t$. This fact (and the corresponding statistic that one can use to obtain it from the Gibbs samples) is stated in lines 259–263 and its derivation is given in Appendix E.1. Thus, the template does indeed accommodate methods that require the Jacobian of the denoiser.
>
> With the help of reference [1] that you provided, we (non-exhaustively) enumerate the algorithms that fit our template as follows (acronyms taken from Table 1 in [1]): In the family of explicit approximations for measurement matching, Score-ALD, Score-SDE, ILVR, DPS, $\Pi$GDM, Moment Matching, SNIPS, DDRM, DDNM, DDS, DiffPIR fit the template. In the family of asymptotically exact methods, DPnP, PnP-DM, FPS, and PMC fit the template. *In the updated manuscript, we will give the corresponding update steps $\mathcal{S}$ for a selection of them that we deem most important to illustrate the general concept.* We plan to run additional experiments with those algorithms, but we will not add them to the present paper in order to keep the revisions as concise and clear as possible. However, we will add those results to the *github and the website*.
>
> **On the computational complexity of the estimation of the covariance:** Since the likelihood of the denoising problem always ensures that the denoising posterior is sub-Gaussian—irrespective of the prior distribution—, the number of samples needed to achieve a given error in the operator norm of the sample covariance scales linearly with the dimension (Theorem 4.7.1 in Vershynin, “High-Dimensional Probability”). *We will discuss this in the updated manuscript.* As already mentioned, should it be required, arbitrary precision can be achieved with a sufficiently large number of samples.

---

> ### Author Response · Authors · 2025-11-18
>
> **Evaluation design**
>
> > The inclusion of learned denoisers ...
>
> Thank you for this remark. We fully agree that the current numerical experiments do not remove approximation errors. The current evaluation benchmarks the performance of the algorithms in the practical setting where learned components approximate various objects, and the robustness to swapping out the learned denoiser with the Gibbs denoiser. Since the Gibbs denosier provides higher-quality denoising estimates (or denoising-posterior samples in the case of DPnP), the current results allow one to see if there is an entanglement between the quality of the denoiser and the hyperparameters. As expected, the DPnP algorithm (an "asymptotically exact" algorithm whose hyperparamters are mostly not influenced by the quality of the denoiser) performs best in this respect. This evaluation is valuable in an of itself.
>
> *We will adapt various passages in the updated manuscript such that they emphasize that the current numerical setup benchmarks the performance in the practical setting and the robustness to hyperparameter choice rather than the quality of samples obtained by the algorithms under the assumption of an optimal denoiser. In addition, the updated manuscript will contain the results obtained by tuning the methods with the Gibbs denoiser. Due to computational limitations, we will restrict this to one increment distribution and forward operator.* We hope that we can fill the remaining results in the github and the website fully with community support.
>
> > The experimental comparison is limited.
>
> We agree that the current experimental comparison is limited and hope to change that with the help of the community. For this purpose, *we provide the github repository along with a dedicated website that will list all available benchmark results.*
>
> We are aware of [1] which is, indeed, a good reference upon which we fit our template so that almost all algorithms with explicit guidance (and others) can easily be cast into our framework (see the list above). The authors of [1] also "strongly believe that the field would benefit from a standardized benchmark" and speculate that "under certain distributional assumptions, it should be possible to characterize analytically the propagation of the approximation errors induced by the different methods." Our benchmark does precisely that, albeit by providing arbitrary-precision Monte Carlo reference objects instead of analytical expressions.
>
> The general idea of supplying reference objects can easily be extended to optimization-based or variational methods. In fact, many of these methods (most in the CSGM family and others) require the unconditional sampling of the prior given some initial condition (see, e.g., lines 3–7 in Algorithm 2 in [5]) in each iteration of the inference procedure. This can trivially be replaced with Gibbs sampling, which does not necessitate any reverse diffusion at all and, consequently, can even be *faster* than the reverse diffusion with the learned denoisers. This is also the case for DPnP, as we briefly mention in lines 833–835 in the appendix. We *will expand this discussion and move it to the main body of the updated manuscript.*
>
> However, the goal of these methods is often not formulated as posterior sampling. For instance, the optimization-based DMPlug aims to recover that initial condition whose corresponding resulting sample best fits the data term, see equation (7) in [5]. Similarly, variational approximations typically aim to find the best Gaussian approximation to the posterior, but typically only evaluate the quality of the obtained mean (with domain-dependent quality metrics such as SSIM or LPIPS) and do not sample this Gaussian. The inference stage of these algorithms is not a diffusion process and, consequently, these algorithms are not part of the family of diffusion posterior sampling algorithms. *We will explicitly discuss this in the updated manuscript.*
>
> **Remarks and minor issues**
>
> > In background, ...
>
> Thank you for pointing this out. *We will change this formulation in the updated manuscript.*
>
> > The used abbreviation DPS ...
>
> We agree that this can cause confusion. *We will highlight the distinction between the umbrella term of DPS algorithms with the instance of Chung upon the introduction of the acronym.*
>
> > The authors may also consider ...
>
> Thank you for providing that reference. *The updated manuscript will contain a discussion about the differences of the proposed approach to [3] in the related work section.*
>
> > Line 288: ...
>
> Although the DiffPIR paper explicitly derives relations that make C-DPS a special case (Appendix A.2. of the arXiv version), we agree that this statement can be misleading. *We will remove this statement in the updated manuscript.
>
> [5] Wang et al., DMPlug: A Plug-in Method for Solving Inverse Problems with Diffusion Models, NeurIPS 2024

---

> ### Author Response · Authors · 2025-11-18
>
> **Questions**
>
> > I generally found the figures hard to understand and interpret, I'm referring namely to figure 1, it says that it shows reverse using the oracle denoiser, but it is not clear, similarly, for figure 3, it is hard to interpret namely to say wether the algorithms performs well or not
>
> The main purpose of Figures 1 and 2 is to show examples of signals that we consider in this work and that we can synthesize them with the (unconditional) reverse diffusion without any learned components (the expectation of the denoising posterior is estimated from Gibbs samples).
>
> The main purpose of Figure 3 is to support the argument that samples of the posterior distribution exhibit characteristics of the prior distribution, whereas the expectation of the posterior does not necessarily exhibit those. (In the particular example of Figure 3, the samples are piecewise constant while the expectation of the posterior is not.) We deliberately omitted the Gibbs reference (posterior expectation and marginal variance) since it is not important for that argument. The quality of the reconstructions for all considered forward operators, for all algorithms, and many prior distributions can be judged by comparing them to the Gibbs reference in Figures 8 through 15 in the appendix.
>
> *We will improve the placement of Figure 3 in the updated manuscript, such that it will appear in the paragraph that discusses it.* We would appreciate any additional clarification on what is unclear in Figures 1 and 3 such that it can be improved in an updated manuscript.
>
> > can the authors provides hints/explanation on the derivation of equation (13)
>
> To obtain (13) we first rewrite the likelihood term in (10) as
> $$\exp\bigl( -\tfrac{1}{2\sigma_{\mathrm{n}}^2}\|\|\mathbf{A}\mathbf{x} - \mathbf{y} \|\|^2 \bigr)  = \exp\Bigl( -\tfrac{1}{2\sigma_{\mathrm{n}}^2} \sum_{k=1}^m \bigl((\mathbf{A}\mathbf{x})\_k - \mathbf{y}\_k\bigr)^2 \Bigr) = \prod_{k=1}^m \exp\Bigl( -\tfrac{1}{2\sigma_{\mathrm{n}}^2} \bigl((\mathbf{A}\mathbf{x})\_{k} - \mathbf{y}\_k\bigr)^2 \Bigr) \propto \prod_{k=1}^m g_{\mathbf{y}\_k,\sigma_{\mathbf{n}}}\bigl( (\mathbf{A}\mathbf{x})\_k \bigr)$$
>  where $g_{\mu,\sigma^2}$ is the density of a one-dimensional Gaussian distribution with mean $\mu \in \mathbb{R}$ and variance $\sigma^2 \in \mathbb{R}$, which is given by $g_{\mu,\sigma^2}(x) = \tfrac{1}{\sqrt{2\pi\sigma^2}}\exp\bigl( -\tfrac{1}{2\sigma^2}(x-\mu)^2 \bigr)$. We get (13) by multiplying that formulation of the likelihood with the prior that is defined in (9) and appears in (10). The last equality in (13) is obtained by the appropriate definition of the matrix $\mathbf{K}$ and the distributions $\phi_1, \phi_2, \dotsc, \phi_{m+d}$ that is given in line 209. *We will make this clearer in the updated manuscript.*
>
> > the authors claim that introduced framework can also assess the approach where the conditional components is learned (Line: 168-170), but it is not clear how it can be achieved given that in some tasks the likelihood is not known ...
>
> Our framework can be used to evaluate any method that claims to sample the posteriors that can be expressed as in equation (10). This includes methods where the posterior score is learned directly. In this case, however, it is intrinsically impossible to disentangle learned components from algorithmic errors and, thus, our framework provides less utility.
>
> Our benchmark cannot be applied when the likelihood is unknown since, in that case, there does not exist a clear statistical target. *We will improve the formulation of lines 168–170 in the updated manuscript to reflect this better.*
>
>
> > A more broad question: did the authors think about how the benchmark can be extend to nonlinear inverse problems
>
> Fundamentally, the computation of the reference posterior samples in our framework relies on the existence of a bias-free and (reasonably) efficient sampler for those distributions. The inference stage of our framework relies on the ability to efficiently sample the denoising posteriors in the reverse diffusion; this is always a linear inverse problem and, consequently, does not change when the likelihood changes.
>
> Efficient sampling of general posteriors that arise from nonlinear measurement models and non-Gaussian noise models is highly nontrivial. There exist some approaches that are applicable to specialized settings, such as the one in [2] that considers a nonlinear measurement model with Gaussian noise and a Laplace prior. However, we believe that there currently exists no general-purpose method that covers all our distributions and, consequently, it is currently not possible to benchmark these settings comprehensively.
>
> In summary, it is possible to extend our framework to specialized settings for which semi-efficient sampling routines exist that can be used in the offline stage of the computation of the reference samples. *We will add a discussion about the extensions to nonlinear inverse problems to the updated manuscript.*

---

### Author Response · Authors · 2025-11-18
**Overview**

We thank the reviewers for the time and effort spent in reviewing our paper and for many suggestions on improvements that we will implement in the updated manuscript. We have identified three points that are shared by at least two reviewers. We address these points here on a high level and in more detail in the individual responses to the reviewers.
### 1. Evaluation desgin (tkeZ, TmEt)
The reviewers argue that the current evaluation does not isolate algorithmic errors from approximations errors of learned components, whereas parts of the manuscript suggest that it does. We fully agree *and we will update the manuscript to reflect that we currently benchmark the realistic setting where various learned objects are utilized (Table 1) and quantify the hyperparameter sensitivity with respect to the quality of the denoiser (Table 4). In addition, we will tune all algorithms on a selected subset of the problems (one forward operator and one increment distribution) using the Gibbs denoiser to showcase the possibility of isolating the algorithmic error.*
In any case, the current evaluation is valuable in and of itself. It is a compromise due to limited compute resources since the tuning of the hyperparameters of the algorithms with the reference Gibbs objects is time consuming. To address this, we will *provide a dedicated section on the github and a website where the community can report their results*. We hope that a comprehensive benchmark can be built with community effort. We will continuously update these and thereby build a comprehensive and fair evaluation.
### 2. Relevance of the setting and extensions to nonlinear and higher-dimensional problems (tkeZ, TmEt, pM9c)
In the paper, we did not clearly articulate the design principle behind our benchmark, which may have caused some confusion. To construct a statistical benchmark, one needs to be able to generate bias-free reference samples from the posterior in order to reliably assess the accuracy of approximate sampling methods. This requirement strongly constrains the class of admissible distributions and, to the best of our knowledge, existing state-of-the-art general-purpose samplers that can provide such reference posteriors are available precisely for the class we consider.

Importantly, the goal of the benchmark is not to mimic any single “realistic” application as closely as possible, but to provide a controlled yet nontrivial stress test for posterior sampling algorithms. In that sense, our setting plays a role analogous to Gaussian-mixture benchmarks, such as the one due to Crafts \& Villa (2025) that is referenced in the manuscript. Such benchmarks have been invaluable for diagnosing strengths and failure modes of inference methods. Our framework extends these simple Gaussian setups to a substantially richer family of heavy-tailed Lévy priors and non-Gaussian likelihoods, while still admitting computationally tractable reference posteriors. If an algorithm fails in this controlled, moderate-dimensional regime with well-specified ground truth, it is unlikely to perform robustly in higher-dimensional and more complex scenarios, so performance here is informative in and of itself.

---

> ### Author Response · Authors · 2025-11-18
> **Overview (continued)**
>
> **Nonlinear problems:** Fundamentally, the computation of the reference posterior samples in our framework relies on the existence of bias-free and (reasonably) efficient samplers for those distributions. Efficient sampling of general posteriors that arise from nonlinear measurement models and non-Gaussian noise models is highly nontrivial, and only some approaches that are applicable to specialized settings exist (for instance, the one presented in [1] that considers a nonlinear measurement model with Gaussian noise and a Laplace prior). However, we believe that there currently exists no general-purpose method that covers all our distributions and, consequently, it is not possible to benchmark these settings comprehensively. The benchmarking stage of our framework relies on the ability to efficiently sample the denoising posteriors in the reverse diffusion. This is always a linear inverse problem and, consequently, does not change when the likelihood changes.
>
> *We will discuss possible extensions to nonlinear problems in the updated manuscript.* In any case, given the popularity of linear measurement models throughout many scientific domains, a benchmark of these models is valuable in and of itself.
>
> **Higher dimensions:** The primary challenge when going to higher dimensions lies in the sampling of the Gaussian distributions that arise in the conditional sampling of $\mathbf{X} \mid \mathbf{Z} = \mathbf{z}$. As we discuss in Appendix D.2, sampling high-dimensional Gaussians is a well-studied problem and advances in that field can directly be used in our framework. In particular, the structure of the involved operators in our case is such that the Gaussians be efficiently sampled with perturb-and-MAP approaches with matrix-free conjugate gradient implementations. *We will add a discussion about the computational complexity to the updated manuscript along with an experiment that shows how the runtime depends on the dimensionality of the signals.* We believe that the relative ordering of the performance of the algorithms does not change with the dimensionality of the problems, and that the relative ordering of the algorithms on these moderate-dimensional problems is valuable in and of itself.
>
> ### Extensions to other posterior solvers (tkeZ, TmEt, pM9c)
> Samples of *any* sampling algorithm (even classical sampling algorithms) can be compared to the reference posterior samples obtained by the efficient parameter- and bias-free Gibbs methods. We primarily focus on diffusion posterior sampling algorithms because it is natural to sample the denoising posteriors that arise in those algorithms with the Gibbs methods. The fundamental idea of replacing learned components with reference objects is also applicable to other plug-and-play solvers, for instance those that utilize flow-matching priors. A unifying benchmarking framework that encompasses more posterior solvers is interesting future work but out of the scope of this paper. *We will emphasize that any algorithm can be benchmarked (though, possibly learned components can not be replaced) in the updated manuscript. In addition, we will add a discussion about extensions to a selection of alternative posterior solvers.*
>
> We will upload an updated manuscript as soon as possible but we would appreciate if the reviewers find the time to communicate if we have addressed their concerns based on the comments that we give below without access to the updated manuscript.
> That way we can make sure that the final version is up to standards.
>
> [1] Wang, Bardsley, et al. "Bayesian Inverse Problems with l1 Priors: A Randomize-Then-Optimize Approach", SIAM Journal on Scientific Computing, Vol. 39, Iss. 5 (2017).

---

### Author Response · Authors · 2025-12-02
**Summary of Revisions**

We thank the reviewers for their thoughtful and constructive feedback.
We have addressed the comments comprehensively, and the manuscript has improved substantially in clarity, scope, and technical precision.
To facilitate the next round of review, we provide two versions of the revised manuscript: (i) a marked-up version with changes highlighted and annotated to point to the corresponding reviewer feedback, and (ii) a clean version without markup that strictly adheres to the 10-page limit.
The two versions have been concatenated in that order and uploaded as a single file.

### **Clarified terminology and evaluation design**
We now describe the objects previously referred to as "oracle" more precisely as *arbitrary-precision Monte Carlo* objects.
This makes explicit that discrepancies between them and their corresponding target arise solely from Monte Carlo error and can be made arbitrarily small by increasing the sampling budget.

We also refined the description of our evaluation design in light of the reviewers' feedback:
We now clearly distinguish (i) isolating algorithmic errors from (ii) evaluating, *for fixed hyperparameters*, the robustness of replacing learned components (e.g., the MMSE denoiser and its Jacobian, or other statistics) by their corresponding Monte Carlo estimates computed from Gibbs samples at a specified precision.
We added a systematic assessment of the accuracy of the Monte Carlo denoiser to support the interpretation of these results.

### **Strengthened motivation and extended scope**
Following the reviewers' suggestions, we strengthen the motivation for Lévy processes as (1) models that are well-suited to real-world signals (now supported with examples from financial time series and image statistics) and (2) a broad class for which we can efficiently compute arbitrary-precision Monte Carlo objects.
We now report runtimes for our experiments and provide implementation-level speedup details.
We also discuss extensions to nonlinear and higher-dimensional problems and quantify runtime in terms of the signal dimension.

We also make the benchmark's broad applicability more explicit:
(1) *any* method that claims to sample from the posterior can be evaluated by comparison to our Gibbs reference samples; and
(2) *any* learned component that can be estimated from denoising-posterior samples can be replaced by its corresponding arbitrary-precision Monte Carlo estimate.
We illustrate this breadth by adding the update steps $\mathcal{S}$ for several additional well-known algorithms.

### **Improved exposition throughout**
We revised structure, notation, and several passages to improve soundness and remove potential sources of confusion.

---

### Meta-Review · Area_Chair_1roe · 2026-01-06

**Summary:**

This paper introduces a statistical benchmark based on discretized Lévy processes with tractable Gibbs posteriors, serving as ground truth for evaluating posterior sampling algorithms built on diffusion models. The reviewers generally find this to be a valuable contribution that can help benchmark, compare, and motivate developments in this active research area. While the first-round reviews raised several concerns regarding scope and evaluation design, the authors’ responses largely clarify these issues, leading to an overall inclination toward acceptance.

**Reviewer Concerns:**

Most of the reviewers’ concerns are adequately addressed in the rebuttal. The authors strengthen the motivation for using Lévy processes by connecting them to real-world signals such as financial time series and image statistics, demonstrate meaningful speedups enabled by various algorithmic techniques, and expand the proposed template to make it applicable to a much broader class of existing posterior sampling algorithms based on diffusion models.

One remaining issue noted by both reviewers tkeZ and pM9c, and agreed upon by the area chair, is that the abbreviation “DPS” conflicts with the name of an already well-known algorithm for this task. In addition, further results demonstrating scaling to higher-dimensional settings would strengthen the paper.

**Reviewer Scores:**

Reviewer tkeZ may increase their score from 4 to 6 or 8, as the misleading use of the term “oracle” and concerns about the evaluation design are clarified, and the updated algorithmic template now covers a much broader range of methods from the literature.

Reviewer bQ8j is likely to maintain their score of 8.

Reviewer TmEt may increase their score from 4 to 6, as the major weaknesses are addressed and the authors clarify that higher-dimensional experiments are feasible.

Reviewer pM9c has indicated they would maintain their score of 6.

---

### Decision · Program_Chairs · 2026-01-26

Accept (Poster)